# Does Representation Guarantee Welfare?

**Jakob de Raaij**
Harvard University
jderaaij@fas.harvard.edu

**Ariel D. Procaccia**
Harvard University
arielpro@seas.harvard.edu

**Alexandros Psomas**
Purdue University
apsomas@cs.purdue.edu

## Abstract

A panel satisfies *descriptive representation* when its composition reflects the population. We examine the role of descriptive representation in collective decision making through an optimization lens, asking whether representative panels make decisions that maximize social welfare for the underlying population. Our main results suggest that, in general, representation with respect to *intersections* of two or more features guarantees higher social welfare than that achieved by the status quo of proportionally representing individual features. Moreover, an analysis of real data suggests that representation with respect to pairs of features is feasible in practice. These results have significant implications for the design of *citizens' assemblies*, which are gaining prominence in AI governance.

## 1 Introduction

A *citizens' assembly* is a group of everyday people who come together to discuss a policy issue, with the goal of informing decision makers [23, 25, 8, 18]. This paradigm has been on the rise around the world, with some of the most prominent assemblies convened by European governments to tackle constitutional questions and climate change. The success of citizens' assemblies is driven, at least in part, by several characteristics.

- *Deliberation:* Assembly members engage in a deliberative process over a period of weeks or months.
- *Random selection:* The assembly is selected through a lottery.
- *Descriptive representation:* The lottery is designed to ensure that the assembly proportionally reflects the population according to multiple features, such as age, gender, ethnicity, level of education, geography, and even viewpoint.

In this paper, we are interested in better understanding the role of descriptive representation. Political theorists have long argued that descriptive representation contributes to the legitimacy of citizens' assemblies and promotes public trust in their recommendations. This thesis is largely supported by experimental evidence [15], but it is disputed by some studies [16].

We steer clear of this (partly normative) debate as we examine descriptive representation through a novel, quantitative lens. Our key research question is this:

> *If a panel satisfies descriptive representation, does it make decisions that optimize social welfare for the underlying population?*

39th Conference on Neural Information Processing Systems (NeurIPS 2025).

If the answer were a clear "yes," it would provide a new, objective way to justify the importance of descriptive representation. And if the answer were more nuanced (spoiler alert: it is), it would inform the design of future citizens' assemblies that make better decisions or recommendations.

**Our approach and results.** There are several things to unpack in our informal research question; we start with *social welfare*. We assume that each individual is identified by their feature values (e.g., state = Massachusetts), and that each *alternative* is associated with benefits for each feature value (e.g., a resident of Massachusetts would derive benefit 0.2). The *utility* of an individual for an alternative is simply these benefits summed over the individual's feature values. A social welfare function maps these individual utilities to an aggregate measure of welfare; for example, *utilitarian social welfare* returns the sum of individual utilities. We focus on the well-established class of *power mean functions* [6, 20], which is parameterized by $p \in [-\infty, 1]$. The class includes utilitarian social welfare ($p = 1$), as well as other prominent social welfare functions such as egalitarian and Nash.

The more subtle definition is that of *descriptive representation*. Ideally, one would want every intersection of features to be proportionally represented on the panel. In practice, citizens' assemblies ask for what we call *1-representation:* The proportion of each feature value on the panel (approximately) matches the population. One of our main insights is to extend this requirement to $m$-representation, which requires that each intersection of up to $m$ features be proportionally represented on the panel.

With this terminology in hand, we can refine our research question:

> *How well does the p-mean welfare of a m-representative panel estimate the p-mean welfare of the underlying population, for different values of p and m?*

The reader may have noticed an ostensible gap between the earlier goal of *making good decisions* and the current goal of *accurately estimating welfare*; as we show, the latter goal implies the former in our model. But if our goal is simply to estimate welfare, why do we even need a deliberative process — can we not just poll the population? The answer is that we view the utilities of individuals in the population as the utilities they would have under the same conditions as a citizens' assembly, after being informed about the topic and engaging in a deliberative process. These utilities are unknowable, of course, and indeed our guarantees do not require access to them.

In Section 3, we investigate the relation between the status quo of 1-representation and social welfare. Our first technical observation is that for utilitarian social welfare ($p = 1$), if a panel is 1-representative, its (mean) social welfare for any alternative is equal to the (mean of the) population's social welfare for the same alternative. Therefore, a 1-representative panel that seeks to maximize utilitarian social welfare will select an alternative that is optimal for the population (by the same measure). However, for $p < 1$, we prove that this is not the case: there are populations and sets of alternatives such that no 1-representative panel chooses the alternative that maximizes the $p$-mean welfare of the population. In Theorem 8 we determine the worst-case *accuracy* — that is, the worst-case difference between the $p$-mean welfare of the panel estimate and the $p$-mean welfare of the underlying population — of 1-representative panels.

In Section 4 we proceed to study 2-representative panels. In Theorem 9, we prove that 2-representation allows for significantly better worst-case guarantees for accuracy, compared to 1-representation. Our proof crucially relies on the fact that finding the minimum/maximum welfare a $m$-representative panel can have for an alternative (the difference between this maximum and minimum is the object we want to bound) boils down to solving a *moment problem* [22]. To get explicit bounds, we leverage the fact that extreme points of our moment problem are probability measures (corresponding to populations) supported on a discrete set of size at most 3 [17], for the case of 2-representative panels.

Finally, in Section 5 we experimentally show that it is possible to select 2-representative panels in real-world instances. We analyze data from four different citizens' assemblies: one from a Western European country and three from Australian states. Our experiments show that (approximate) 2-representative panels exist with the current panel sizes for two of the datasets, and would exist by picking a slightly smaller panel for a third dataset. In the final dataset, (approximate) 2-representation is infeasible; we attribute this fact to the very small and skewed pool of volunteers.

Combined, our theoretical and experimental results provide a clear, prescriptive message: It is practical to select panels that are 2-representative and, by doing so, the quality of the decision the panel makes (as measured by its $p$-mean welfare for the entire population) significantly increases.

**Related work.** There is a fast-growing body of work on algorithms for selecting and facilitating citizens' assemblies. The majority of papers develop and analyze algorithms for computing lotteries over assemblies [9, 10, 11, 12, 2, 14], while others focus on supporting deliberation [3, 4]. Our work is closer in spirit to papers that consider the connections and tradeoffs between representation and fairness [3, 7]; by contrast, we investigate the relation between representation and welfare (and discuss the addition of fairness in Section 6).

In terms of impact on AI, some of the most prominent technology companies have been experimenting with citizens' assemblies as a tool for AI governance. In particular, Meta has organized a large-scale deliberation on AI policy [5]. Google DeepMind has explored related ideas and their researchers have published prominent work on the use of large language models for mediating deliberation in citizens' assemblies [24].

## 2 Preliminaries

We consider the problem of a population of individuals selecting a representative panel to make a decision on its behalf. Every individual is represented using a set $\mathcal{F}$ of features (e.g., gender, age, geographical region, and so on), where each feature $f \in \mathcal{F}$ can take $d_f$ different values; we denote by $V_f$, $|V_f| = d_f$, the set of values that feature $f \in \mathcal{F}$ can take. Every individual is represented by a *type* $y$, an assignment of values to the features in $\mathcal{F}$. For any $f \in \mathcal{F}$, let $y(f, v)$ be the indicator function for $y$ having value $v$ for feature $f$. That is, $y(f, v) = 1$ if and only if the type $y$ has value $v$ for feature $f$, else $y(f, v) = 0$. We say that two types are distinct if there exists at least one feature $f$ for which they have a different value. We write $\mathbf{y} = (y(f, v))_{v \in V_f, f \in \mathcal{F}} \in \{0, 1\}^d$, where $d = \sum_{f \in \mathcal{F}} d_f$, to represent the vector containing $y(f, v)$ for all $v \in V_f$ and $f \in \mathcal{F}$.

A *population* $\mathcal{P} = \{(y_1, q_1), ..., (y_T, q_T)\}$ is a set of $T$ distinct types of individuals $\mathcal{I} = \{y_1, ..., y_T\}$, where individuals of type $y_i$ make up a fraction $q_i \in (0, 1]$ of the population. We call $q_i$ the *population share* of type $y_i$. It holds that $\sum_{i=1}^{T} q_i = 1$. We denote by $n$ the total number of individuals in the population. Thus, $q_i \cdot n \in \mathbb{N}$ for all $i \in [T]$.

**Example 1.** Consider a situation where $\mathcal{F} = \{age, gender\}$, with $V_{age} = \{15\text{-}29, 30+\}$ and $V_{gender} = \{Male, Female, Non\text{-}binary\}$. Therefore, $d_{age} = 2$ and $d_{gender} = 3$. Consider an individual who is *30+* and *Male*. Their type, $y_1$, is such that, for example, $y_1(age, 30+) = 1$ and $y_1(gender, Non\text{-}binary) = 0$. The corresponding vector is $\mathbf{y_1} = (0, 1; 1, 0, 0)$: they have the second value for the first feature and the first value for the second feature. Finally, consider a population $\mathcal{P} = \{(y_1, 0.4), (y_2, 0.4), (y_3, 0.2)\}$, meaning that 40% of the population has type $y_1$, 40% has type $y_2$, and 20% has type $y_3$.

**Panels and representation.** A *panel* $\mathcal{C} = \{(y_1, r_1), ..., (y_T, r_T)\}$ in a population $\mathcal{P}$ specifies, for each type $y_i \in \mathcal{I}$, its fraction $r_i \in [0, 1]$ in the panel. It holds that $\sum_{i=1}^{T} r_i = 1$. We denote by $k$ the total number of individuals in a panel (thus, $r_i \cdot k \in \mathbb{N}$ for all $i$). We assume that the panel is a subset of the population $\mathcal{P}$, and therefore $r_i \cdot k \leq q_i \cdot n$ for all $i \in [T]$.

We are interested in panels that are *representative* of the population. A panel $\mathcal{C}$ in a population $\mathcal{P}$ is *1-representative* if for every feature $f \in \mathcal{F}$, and every value $v \in V_f$, the fraction of individuals in $\mathcal{C}$ with value $v$ is equal to the fraction in $\mathcal{P}$. That is, for all $f \in \mathcal{F}$ and $v \in V_f$

$$\sum_{y_i \in \mathcal{I}} q_i \, y_i(f, v) = \sum_{y_i \in \mathcal{I}} r_i \, y_i(f, v) \,,$$

or, more concisely, $\sum_{y_i \in \mathcal{I}} q_i \mathbf{y_i} = \sum_{y_i \in \mathcal{I}} r_i \mathbf{y_i}$. This is essentially equivalent, up to weakening the equality constraint to allow for small deviations, to the notion of representation considered in prior work on panel selection.

More generally, we define an $m$-representative panel as follows:

**Definition 2.** For any $m \in \{1, ..., |\mathcal{F}|\}$, we say that a panel $\mathcal{C}$ in a population $\mathcal{P}$ is *m-representative* if for every set of $\ell \leq m$ features $F \subseteq \mathcal{F}$, $|F| = \ell$ and every possible $\ell$-tuple of values $(v_f)_{f \in F} \in \bigtimes_{f \in F} V_f$, the fraction of individuals in $\mathcal{C}$ with values $v_f$ is equal to the fraction in $\mathcal{P}$. That is, for all such $F$ and $(v_f)_{f \in F}$,

$$\sum_{y^i \in \mathcal{I}} q_i \prod_{f \in F} y_i(f, v_f) = \sum_{y^i \in \mathcal{I}} r_i \prod_{f \in F} y_i(f, v_f).$$

Note that $\prod_{f \in F} y_i(f, v_f) = 1$ if and only if type $y_i$ has value $v_f$ for all $f \in F$.

**Alternatives and utility.** Panels are asked to decide on an *alternative*, from a given set of alternatives $\mathcal{A}$. For an alternative $a \in \mathcal{A}$ and a feature $f \in \mathcal{F}$, let $a(f, v) \in \mathbb{R}_{\geq 0}$ be the benefit that an individual with value $v$ for feature $f$ gets from $a$. We define $\mathbf{a} = (a(f, v))_{v \in V_f, f \in \mathcal{F}}$ analogously to $\mathbf{y}$.

We assume that the *utility* of an individual for an alternative $a$ is the sum of all these feature benefits, i.e., the utility of an individual of type $y_i \in \mathcal{I}$ for alternative $a$ is

$$u_i(a) = \mathbf{y} \cdot \mathbf{a} = \sum_{f \in \mathcal{F}} \sum_{v \in V_f} y_i(f, v) a(f, v).$$

Without loss of generality, we assume throughout the paper that utilities are non-negative and that the utility of every individual for every alternative is at most 1. For a given set of alternatives $\mathcal{A}$ and population $\mathcal{P}$, we define $u_{\min} = \min_{y_i \in \mathcal{I}} \min_{a \in \mathcal{A}} u_i(a)$ to be the minimum utility of any individual in the population for any alternative in $\mathcal{A}$. We have that $u_{\min} \in [0, 1]$.

**Example 3** (Example 1 continued)**.** Consider a set of alternatives $\mathcal{A} = \{a_1, a_2\}$, where $a_1(age, 15\text{-}29) = 0$, $a_1(age, 30+) = 0.5$, $a_1(gender, Male) = 0.3$, $a_1(gender, Female) = 0.5$, $a_1(gender, Non\text{-}binary) = 0$, $a_2(age, 15\text{-}29) = 0.7$, $a_2(age, 30+) = 0.3$, $a_2(gender, Male) = 0$, $a_2(gender, Female) = 0$, and $a_2(gender, Non\text{-}binary) = 0.3$. In short, $\mathbf{a_1} = (0, 0.5; 0.3, 0.5, 0)$ and $\mathbf{a_2} = (0.7, 0.3; 0, 0, 0.3)$.

Then, since $y_1$ has values *30+* and *Male*, we have that

$$u_1(a_1) = \mathbf{y_1} \cdot \mathbf{a_1} = a_1(age, 30+) + a_1(gender, Male) = 0.5 + 0.3 = 0.8,$$

while $u_1(a_2) = 0.3 + 0 = 0.3$. Assume that $y_2$, $y_3$ are such that also $u_2(a_1)$, $u_2(a_2)$, $u_3(a_1)$, $u_3(a_2) \geq 0.3$; therefore, $u_{\min} = 0.3$.

**Panel and population welfare.** We assume that the welfare of a population $\mathcal{P}$ for an alternative $a$ is the $p$-mean of individuals' utilities, for some $p \in \mathbb{R}$, $p \leq 1$. That is, $u_{\mathcal{P}, p}(a) = (\sum_{y_i \in \mathcal{I}} q_i u_i(a)^p)^{1/p}$. For $p = 1$ this is the standard arithmetic mean ($u_{\mathcal{P}, 1}(a) = \sum_{y_i \in \mathcal{I}} q_i u_i(a)$), corresponding to utilitarian social welfare, for $p = 0$ it becomes the geometric mean ($u_{\mathcal{P}, 0}(a) = \prod_{y_i \in \mathcal{I}} u_i(a)^{q_i}$), corresponding to Nash social welfare, and as $p$ approaches $-\infty$ it becomes the minimum ($u_{\mathcal{P}, -\infty}(a) = \min_{y_i \in \mathcal{I}} u_i(a)$), corresponding to egalitarian social welfare. Analogously, we define the utility of a panel $\mathcal{C}$ for an alternative $a$, $u_{\mathcal{C}, p}$, to be the $p$-mean of the utilities of the individuals in the panel.

**Objectives.** There are several natural ways to measure whether a panel makes good decisions, in the sense of representing the opinion of the population well. A natural choice, for example, would be to ask that the social welfare of the panel is an accurate estimate of the social welfare of the population, for every alternative; then, naturally, optimizing for the panel approximately optimizes for the population. Another choice would be to ask that the panel orders alternatives in a similar way as the population would, i.e., if the panel prefers an alternative $a$ over an alternative $a'$, the population cannot have a high welfare for $a'$ and a low welfare for $a$.

Formally, given $\epsilon > 0$, we can define the two notions as follows.

- *Welfare based:* We say that a panel $\mathcal{C}$ is $\varepsilon$-accurate with respect to $p$ for welfare estimation if $|u_{\mathcal{C}, p}(a) - u_{\mathcal{P}, p}(a)| \leq \varepsilon$ for all $a \in \mathcal{A}$.
- *Pairwise comparisons:* We say $\mathcal{C}$ is $\varepsilon$-accurate with respect to $p$ for pairwise comparisons if $u_{\mathcal{C}, p}(a) > u_{\mathcal{C}, p}(a')$ implies that $u_{\mathcal{P}, p}(a) \geq u_{\mathcal{P}, p}(a') - \varepsilon$ for all $a, a' \in \mathcal{A}$.

These two metrics are closely related: the former implies the latter.

**Theorem 4.** *Let $\mathcal{A}$ be a set of alternatives, and let $\mathcal{C}$ be a panel in a population $\mathcal{P}$. Then, for all $\varepsilon \geq 0$ and $p \leq 1$, we have that if $\mathcal{C}$ is $\varepsilon$-accurate with respect to $p$ for welfare estimation, it is $2\varepsilon$-accurate with respect to $p$ for pairwise comparisons.*

The other direction approximately holds as well. If a panel is not $\varepsilon$-representative for welfare estimation for a set of alternatives $\mathcal{A}$, then it is not $\varepsilon$-representative for pairwise comparisons on $\mathcal{A} \cup \{a\}$ for an additional alternative $a$.

**Theorem 5.** *Let $\mathcal{A}$ be a set of alternatives, and let $\mathcal{C}$ be a panel in a population $\mathcal{P}$. Then, for all $\varepsilon \geq 0$ and $p \leq 1$, we have that if $\mathcal{C}$ is not $\varepsilon$-accurate with respect to $p$ and $\mathcal{A}$ for welfare estimation, there exists an alternative $a'$ such that $\mathcal{C}$ is not $\varepsilon$-accurate with respect to $p$ and $\mathcal{A} \cup \{a'\}$ for pairwise comparisons. Furthermore, adding $a'$ to $\mathcal{A}$ does not change $u_{\min}$.*

In light of the tight connection between the two notions implied by Theorems 4 and 5 (whose proofs are deferred to Appendix A.1), in the remainder of the paper, we focus on accuracy with respect to welfare estimation. We therefore refer to a panel that is $\varepsilon$-accurate for welfare estimation simply as $\varepsilon$-accurate.

## 3   Representation to Single Features

In this section, we prove tight upper and lower bounds on the worst-case accuracy of 1-representative panels, for different values of $p$.

We start by proving that, for $p = 1$, a 1-representative panel always has the same utility as the population for every alternative (i.e., it is $\varepsilon$-accurate for $\varepsilon = 0$).

**Theorem 6.** *Let $\mathcal{A}$ be a set of alternatives, $\mathcal{P}$ be population and $\mathcal{C}$ be a 1-representative panel in $\mathcal{P}$. Then, for all $a \in \mathcal{A}$, $u_{\mathcal{C},1}(a) = u_{\mathcal{P},1}(a)$. That is, $\mathcal{C}$ is 0-accurate with respect to $p = 1$.*

*Proof.* For $p = 1$, the welfare of the panel and population is a weighted sum of the utilities of the corresponding individuals. And, an individual's utility is the sum of the alternative's benefit $a(f, v)$ for their feature values $v \in V_f$, $f \in F$. By switching the order of summation, we have

$$u_{\mathcal{C},1}(a) = \sum_{y_i \in \mathcal{I}} r_i u_i(a) = \sum_{y_i \in \mathcal{I}} r_i(\mathbf{a} \cdot \mathbf{y_i}) = \mathbf{a} \cdot \sum_{y_i \in \mathcal{I}} r_i \mathbf{y_i} = \mathbf{a} \cdot \sum_{y_i \in \mathcal{I}} q_i \mathbf{y_i},$$

where the last equality follows from the fact that the panel is 1-representative. Continuing our derivation,

$$u_{\mathcal{C},1}(a) = \mathbf{a} \cdot \sum_{y_i \in \mathcal{I}} q_i \mathbf{y_i} = \sum_{y_i \in \mathcal{I}} q_i u_i(a) = u_{\mathcal{P},1}(a),$$

and the theorem follows. $\square$

Unfortunately, Theorem 6 does not hold for $p < 1$. When $p < 1$, the population and panel welfare functions become non-linear in $\mathbf{a}$, and therefore we cannot factor out $\mathbf{a}$ like we did in the proof of Theorem 6. In fact, as we show next, for any $p < 1$, it is possible that, in the worst case, all 1-representative panels (other than the entire population) perform poorly at welfare estimation for some alternative, and choose alternatives that are suboptimal for the population.

**Example 7.** In this example we will show that for all $u_{\min} \in [0, 1)$, there exists a population $\mathcal{P}$ and a set of alternatives $\mathcal{A}$, such that $\min_{y_i \in \mathcal{I}} \min_{a \in \mathcal{A}} u_i(a) = u_{\min}$, and such that for all $p < 1$, no 1-representative panel chooses the alternative that maximizes the $p$-mean welfare of $\mathcal{P}$.

Assume that there are two features, $f_v$ and $f_w$, that can take values $v_1, v_2$ and $w_1, w_2$ respectively. Thus, there are four possible types of individuals, represented in the left part of Table 1. Let $\mathcal{P}$ be the population where each type is present in exactly one individual. Therefore, the only two 1-representative panels are $\mathcal{C}_1 = \{((1, 0; 1, 0), 1/2), ((0, 1; 0, 1), 1/2)\}$ and $\mathcal{C}_2 = \{((0, 1; 1, 0), 1/2), ((1, 0; 0, 1), 1/2)\}$.

Consider the following set of 3 alternatives: $\mathcal{A} = \{a_1, a_2, a_3\}$, where $\mathbf{a_1} = (u_{\min}/2, 1/2, 1/2, u_{\min}/2)$, $\mathbf{a_2} = (u_{\min}/2, 1/2, u_{\min}/2, 1/2)$, and $\mathbf{a_3} = ((1+3u_{\min})/8, (5-u_{\min})/8, (1+3u_{\min})/8, (1+3u_{\min})/8)$. The utility of each type for each alternative are shown in Table 1.

For any $p < 1$, we can see that the first panel, $\mathcal{C}_1$, gets the highest utility from $a_1$, while the second panel, $\mathcal{C}_2$, gets the highest utility from $a_2$, since the $p$-mean of $x$ and $(1 + u_{\min}) - x$ for $p < 1$ is uniquely maximized when $x = (1 + u_{\min}) - x = (1+u_{\min})/2$. However, overall, the population gets the highest utility from alternative $a_3$. This fact is algebraically tedious to verify; we defer the calculations to Appendix A.2.

The main point of Example 7 is that any 1-representative panel in $\mathcal{P}$, $\mathcal{C}_1$ or $\mathcal{C}_2$, overestimates the welfare of some alternative, $a_1$ or $a_2$, respectively, quite badly. For example, for $u_{\min} = 0$ and any

Table 1: The population and alternatives in Example 7

| panels | values | y | $a_1$ | $a_2$ | $a_3$ |
|---|---|---|---|---|---|
| $C_1$ | $v_1, w_1$ | (1,0;1,0) | $(1+u_{\min})/2$ | $u_{\min}$ | $(1+3u_{\min})/4$ |
|  | $v_2, w_2$ | (0,1;0,1) | $(1+u_{\min})/2$ | 1 | $(3+u_{\min})/4$ |
| $C_2$ | $v_1, w_2$ | (1,0;0,1) | $u_{\min}$ | $(1+u_{\min})/2$ | $(1+3u_{\min})/4$ |
|  | $v_2, w_1$ | (0,1;1,0) | 1 | $(1+u_{\min})/2$ | $(3+u_{\min})/4$ |

$p \leq 0$, both $a_1$ and $a_2$ have a welfare of $0$ with respect to $\mathcal{P}$, but a welfare of $1/2$ for the panel that chooses them. However, alternative $a_3$ gives the highest welfare to the population (for $u_{\min} = 0$ and $p \leq 0$, the population's utility is a number between $1/4$ and $\sqrt{3}/4 \approx 0.43$).

We now state the main theorem of this section, giving a tight bound on the worst-case $\varepsilon$-accuracy of a 1-representative panel.

**Theorem 8.** *Let $\mathcal{A}$ be a set of alternatives, $\mathcal{P}$ be any population, and $\mathcal{C}$ be any 1-representative panel in $\mathcal{P}$, so that $u_{\min} \in [0,1)$. Then, for all $p < 1$, $\mathcal{C}$ is $\varepsilon$-accurate with respect to $p$, where*

$$
\varepsilon = \begin{cases}
\frac{1-p}{p}\left(p \cdot \frac{1-u_{\min}}{1-u_{\min}{}^p}\right)^{\frac{1}{1-p}} + \frac{u_{\min} - u_{\min}{}^p}{1-u_{\min}{}^p} & \text{for } p > 0 \vee (p < 0 \wedge u_{\min} > 0) \\[2mm]
1 + \frac{1-u_{\min}}{\ln(1/u_{\min})}\left(\ln\left(\frac{1-u_{\min}}{\ln(1/u_{\min})}\right) - 1\right) & \text{for } p = 0 \wedge u_{\min} > 0 \\[2mm]
1 & \text{for } p \leq 0 \wedge u_{\min} = 0
\end{cases}.
$$

*Furthermore, for all $p < 1$ and $u_{\min} \in [0,1)$, for any $\varepsilon' < \varepsilon$, there exists a 1-representative panel $\mathcal{C}$ such that $\mathcal{C}$ is not $\varepsilon'$-accurate with respect to $p$.*

The proof of Theorem 8 is deferred to Appendices A.3.3 and A.4.1. The first part of the theorem (the upper bounds on $\varepsilon$) follows as a special case of the analysis of the $m$-representative case in Appendix A.3 that also gives the main results of the following section for 2-representative panels.

## 4 Representation to Tuples of Features

In this section, we prove bounds on the accuracy of $m$-representative panels. We start by showing that 2-representation enables significantly better worst-case $\varepsilon$-accuracy compared to 1-representation.

**Theorem 9.** *Let $\mathcal{A}$ be a set of alternatives, $\mathcal{P}$ be any population, and $\mathcal{C}$ be any 2-representative panel in $\mathcal{P}$, so that $u_{\min} \in [0,1)$. Then, for all $p < 1$, $\mathcal{C}$ is $\varepsilon$-accurate with respect to $p$, where*

$$
\varepsilon = \max_{\kappa_1, \kappa_2} \left(\frac{\kappa_2 - \kappa_1^2}{\kappa_2 - 2\kappa_1 + 1} + \frac{(1-\kappa_1)^2}{\kappa_2 - 2\kappa_1 + 1}\left(\frac{\kappa_1 - \kappa_2}{1 - \kappa_1}\right)^p\right)^{\frac{1}{p}}
$$

$$
- \left(\frac{(\kappa_1 - u_{\min})^2}{\kappa_2 - 2u_{\min}\kappa_1 + u_{\min}{}^2}\left(\frac{\kappa_2 - u_{\min}\kappa_1}{\kappa_1 - u_{\min}}\right)^p + \frac{\kappa_2 - \kappa_1^2}{\kappa_2 - 2u_{\min}\kappa_1 + u_{\min}{}^2}u_{\min}{}^p\right)^{\frac{1}{p}}
$$

*s.t. $u_{\min}{}^2 < \kappa_1^2 < \kappa_2$ and $\frac{\kappa_2 + u_{\min}}{1 + u_{\min}} < \kappa_1 < 1$,*

*for $p > 0 \vee (p < 0 \wedge u_{\min} > 0)$,*

$$
\varepsilon = \max_{\kappa_1, \kappa_2} \left(\frac{\kappa_1 - \kappa_2}{1 - \kappa_1}\right)^{\left(\frac{(1-\kappa_1)^2}{\kappa_2 - 2\kappa_1 + 1}\right)}
$$

$$
- \left(\frac{\kappa_2 - u_{\min}\kappa_1}{\kappa_1 - u_{\min}}\right)^{\left(\frac{(\kappa_1 - u_{\min})^2}{\kappa_2 - 2u_{\min}\kappa_1 + u_{\min}{}^2}\right)} \cdot (u_{\min})^{\left(\frac{\kappa_2 - \kappa_1^2}{\kappa_2 - 2u_{\min}\kappa_1 + u_{\min}{}^2}\right)}
$$

*s.t. $u_{\min}{}^2 < \kappa_1^2 < \kappa_2$ and $\frac{\kappa_2 + u_{\min}}{1 + u_{\min}} < \kappa_1 < 1$*

*for $p = 0 \wedge u_{\min} > 0$, and*

$$
\varepsilon = 1 \text{ for } p \leq 0 \wedge u_{\min} = 0.
$$

*Furthermore, for all $p < 1$ and $u_{\min} \in [0,1)$, for any $\varepsilon' < \varepsilon$, there exists a 2-representative panel $\mathcal{C}$ such that $\mathcal{C}$ is not $\varepsilon'$-accurate with respect to $p$.*

The optimization program in Theorem 9 has two variables, $\kappa_1$ and $\kappa_2$. We call those the *moments*:

**Definition 10.** For $i = 0, ..., |\mathcal{F}|$, the *i-th moment* of a population $\mathcal{P}$ for an alternative $a$ is $\kappa_i = \sum_{y_j \in \mathcal{I}} q_j \cdot (u_j(a))^i$. The *i-th moment* of a panel $\mathcal{C}$ for an alternative $a$ is $\kappa_i = \sum_{y_j \in \mathcal{I}} r_j \cdot (u_j(a))^i$.

Our proof of $\varepsilon$-accuracy has the following high-level steps. First, we prove that the first $m$ moments of an $m$-representative panel $\mathcal{C}$ in $\mathcal{P}$ are the same as the moments of $\mathcal{P}$ for any alternative. That is, for all $a \in \mathcal{A}$ and any integer $1 \leq \ell \leq m$, $\sum_{y_i \in \mathcal{I}} q_i \cdot (u_i(a))^\ell = \sum_{y_i \in \mathcal{I}} r_i \cdot (u_i(a))^\ell$.

Second, given this fact, we can write a mathematical program for the minimum and maximum utility a panel or population with fixed first $m$ moments $\kappa_1, \ldots, \kappa_m$ (note that always $\kappa_0 = 1$) can have. Concretely, by having a variable $x_j$ for the utility of a type $y_j \in \mathcal{I}$, given $m$ moments, we can ensure feasibility by asking that $\sum_{y_j \in \mathcal{I}} q_j \cdot (x_j)^i = \kappa_i$, for all $i \in \{0, 1, ..., m\}$ (and $x_j \in [u_{\min}, 1], q_j \geq 0$). Then, by maximizing and minimizing $(\sum_{y_j \in \mathcal{I}} q_j \cdot (x_j)^p)^{1/p}$, we get bounds on the welfare of a panel or population with moments $\kappa_0, \ldots, \kappa_m$.

Third, the relaxation of this program to also allow $\mathcal{I}$ of infinite size is known as a *moment problem* [22, 21], hence our choice of terminology in Definition 10. In particular, the population is characterized as a probability measure (distribution) $\mu$ on $[u_{\min}, 1]$ where $\mu(x)$ corresponds to the share of individuals with utility $x$. The feasibility constraints become $\int_{u_{\min}}^1 \mu(x) \cdot x^i dx = \kappa_i$ for all $i \in \{0, 1, ..., m\}$, determining the first $m$ moments of $\mu$, and we maximize and minimize $(\int_{u_{\min}}^1 x^p d\mu(x))^{1/p}$.

It is known that the extreme points of moment problems are probability measures supported on a discrete set of size at most $m + 1$ [17]. This additional structure allows for explicit bounds on the maximum and minimum utility of a panel/population as a function of $\kappa_1, \ldots, \kappa_m$, which we obtain for $m = 2$. The difference of these quantities is our bound on the $\varepsilon$-accuracy of a 2-representative panel. We obtain worst-case bounds on $\varepsilon$, as we prove, by maximizing this difference over the set of all $\kappa_1$ and $\kappa_2$ that can be the first two moments of a population $\mathcal{P}$ for an alternative $a \in \mathcal{A}$.

We defer the details of the $\varepsilon$-accuracy proof to Appendix A.3. In Appendix A.4.2, we then prove the second part of Theorem 9: The bounds obtained with the proof technique described above are tight.

For greater $m$, solving the relaxed optimization program analytically to obtain explicit bounds on the maximum and minimum utility of a panel/population as a function of $\kappa_1, \ldots, \kappa_m$ becomes increasingly difficult. And, even if such bounds were obtained for $m > 2$, the overall bound on $\varepsilon$ in the style of Theorem 9 would be an even less tractable optimization program in $m$ variables; it is unclear if its value could even be numerically calculated.

In fact, we believe that already the solution to the optimization program in Theorem 9 does not admit a closed-form solution in terms of $p$ and $u_{\min}$. However, for a fixed $p$ and $u_{\min}$, we can numerically solve this optimization program. We plot these results in Figure 1. In particular, Figure 1a shows the tight worst-case $\varepsilon$-accuracy of 1-representative panels as dashed lines and the upper bound on worst-case $\varepsilon$-accuracy of 2-representative panels as solid lines, for 4 different values of $u_{\min}$, with $p$ varying along the horizontal axis. We find that 2-representative panels are $\varepsilon$-accurate in the worst case for significantly smaller $\varepsilon$. For example, for $u_{\min} = 1/5$ and Nash welfare ($p = 0$), 2-representation gives 0.039 accuracy, while 1-representation only gives 0.155 accuracy. Figure 1b shows the ratio of these bounds: the worst-case $\varepsilon$ for 1-representative panels divided by the bound on the worst-case $\varepsilon$ for 2-representative panels, for different values of $p$, with $u_{\min}$ varying along the horizontal axis. We can see that the relative accuracy increases quite drastically, especially when $p$ is not far into the negatives and $u_{\min}$ is far enough away from 0. If utilities are generally high, this ratio is the largest: for $u_{\min} = 4/5$ (corresponding, after scaling, to utilities in $[8, 10]$) and Nash welfare, accuracy increases by 2700% for 2-representative panels over 1-representative panels.

Given the positive result in Theorem 9, we turn our attention to the only case in which 2-representative panels don't improve on 1-representative panels: In the case $p \leq 0$ and $u_{\min} = 0$, we proved that 1- and 2-representative panels are no better than 1-accurate in the worst case. It is natural to ask if we can do better with $m$-representative panels for larger $m$.

The answer is negative. One could imagine a very popular alternative with utility close to 1 for everyone but a single person $y^*$, who has utility 0. The population's utility (for $p \leq 0$) for this alternative is 0, but any panel that does not include $y^*$ has utility close to 1 for this alternative (and therefore chooses it). We formalize this idea in Theorem 11.

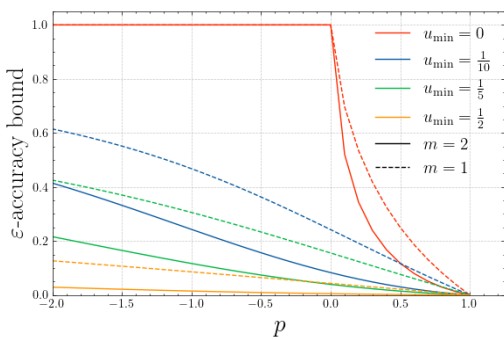 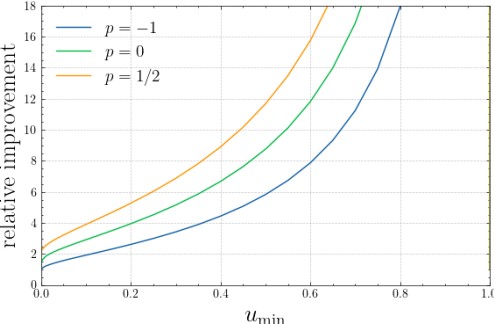

(a) Worst-case $\varepsilon$-accuracy of $m$-representative panels for $m \in \{1, 2\}$.

(b) Relative improvement of $\varepsilon$-accuracy from $m = 1$ to $m = 2$, i.e., ($\varepsilon$-ac. at $m = 1$)/($\varepsilon$-ac. at $m = 2$).

Figure 1: Numerical comparisons of $\varepsilon$-accuracy bounds for $m \in \{1, 2\}$.

**Theorem 11.** *For any positive integer $m$ and number of features $|\mathcal{F}|$, there exists a population $\mathcal{P}$ and a set of alternatives $\mathcal{A}$, such that $\min_{y_i \in \mathcal{I}} \min_{a \in \mathcal{A}} u_i(a) = u_{\min} = 0$, and a $m$-representative panel $\mathcal{C}$ in $\mathcal{P}$, such that $\mathcal{C}$ is not $\varepsilon$-accurate with respect to any $p \leq 0$ for any $\varepsilon < \frac{|\mathcal{F}| - m}{|\mathcal{F}|}$.*

As $|\mathcal{F}|$ becomes large, this lower bound goes to 1, for any fixed $m$. Since $u_{\mathcal{C},p}(a), u_{\mathcal{P},p}(a) \in [0, 1]$ for all $a \in \mathcal{A}$, 1 is also a trivial upper bound. We prove Theorem 11 in Appendix A.5.

## 5 Empirical Analysis

We have shown that 2-representative panels give significantly better worst-case guarantees than 1-representative panels. However, this result only has practical impact if it is possible to select 2-representative panels in real-world instances. Our next goal is to show that this is indeed the case.

**Selecting a panel.** Selecting a citizens' assembly in practice commonly has three stages. First, a large number of individuals receive an invitation to participate in the panel. Second, some fraction of these individuals indicate their willingness to participate. We call this group the *pool of volunteers*. Finally, a panel of the desired size is chosen from the pool of volunteers.

It is a common occurrence in practice that individuals with certain features are more likely to accept the invitation to participate than others. Therefore, the composition of the pool of volunteers often is fairly different from the underlying population. If the size of the pool is too small, it may not be possible to select a panel of a desired size that is 2-representative (or even 1-representative).

**Data.** We analyze data from four citizens' assemblies: a nationwide panel from a Western European country (which we denote EUR1) and three state-wide panels from Australian states (which we denote AUS1, AUS2, and AUS3). Detailed information about the datasets can be found in Appendix A.6. All actual panels were chosen to be approximately 1-representative of the underlying population.

For all four panels, we obtained data on the intersections between feature-values of the underlying population. For EUR1, we used the European Social Survey [19], which collected data on a wide variety of features and their intersections, including those used in EUR1. Responses are re-weighted in the data to account for sampling bias. For AUS1, AUS2, and AUS3, we used data from the 2021 Australian Census [1]. Through their online data tools, we obtained data on the intersections of individuals' features. We excluded a feature capturing the level of climate concern from EUR1 since the number of volunteers with no concern at all was too low to even form a 1-representative panel of the desired size. In AUS1 and AUS2, we had to exclude a feature capturing an individual's ownership of their residence since these statistics are stored in a separate dataset on dwellings, so we weren't able to obtain this feature's intersection with other features. For more details on the datasets and data cleaning, we again refer to Appendix A.6.

**$m$-representation in practice.** Exact 1-representation or $m$-representation for any $m$ is (essentially) never attainable for a fixed desired panel size $k$, as the desired number of individuals with a certain

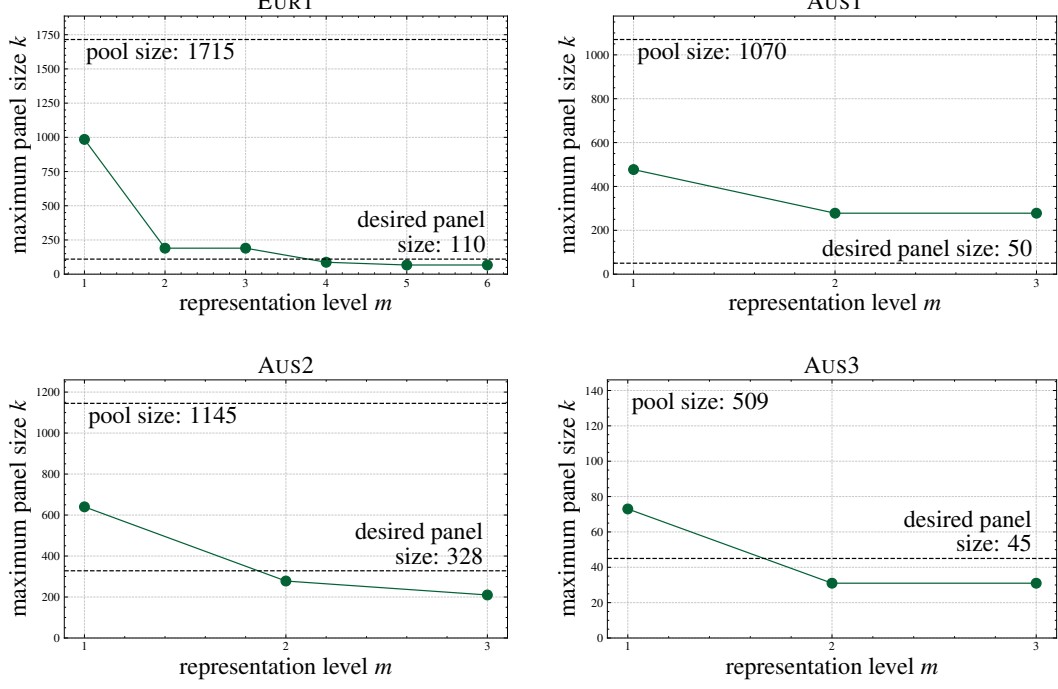

Figure 2: The maximum size of $m$-representative panels up to rounding for $m \in [|\mathcal{F}|]$, shown in teal. The dashed lines show the size of the pool of volunteers and the panel size used in practice.

tuple of feature-values in the panel will almost never be an integer. We consider two weakenings of $m$-representation for practice: A panel is *$m$-representation up to rounding* if for any $\ell$-tuple of feature-values, $\ell \in [m]$, the number of people with these feature-values on the panel is equal to the (rational) desired number, up to rounding it up or down. A panel is *$\varepsilon$-approximately $m$-representative* if for any $\ell$-tuple of feature-values, $\ell \in [m]$, the number of people with these feature-values on the panel is within a multiplicative factor of $(1 \pm \varepsilon)$ of the desired number. Since the quotas from the underlying populations are only ever estimates with some error, these small "wiggle rooms" around the quotas are justified.

**Experiments.** For each $m$ between 1 and the number of features considered in the panel, $|\mathcal{F}|$, we found the largest size of an $m$-representative panel up to rounding using individuals from the pool of volunteers. Furthermore, we found for each $m$ between 1 and $|\mathcal{F}|$ and panel size $k$ between 1 and the pool size the smallest $\varepsilon$ for which an $\varepsilon$-approximately $m$-representative panel exists. For a given $m$ and desired panel size $k$, we checked for the existence of an $m$-representative panel of size $k$ using an integer linear program. Each person in the pool of volunteers corresponds to one binary variable, encoding whether this person is in the panel or not. The constraints on the variables are the size of the panel, $k$, and having to be $m$-representative (up to rounding or $\varepsilon$-approximately) to the underlying population data. We used Gurobi, run on a 14-inch MacBook Pro (2023) with Apple M3 Pro chip, to check whether the program is feasible, i.e., whether an $m$-representative panel exists.

**Results.** Our results for $m$-representation up to rounding are shown in Section 5. The main takeaway is that 2-representation up to rounding is generally feasible in practice.

In the EUR1 and AUS1 datasets, the bottom dashed lines — showing the maximum size of an $m$-representative panel — lie above the "desired panel size" used in practice. For AUS2, the largest 2-representative panel is of size 278 — slightly below the desired panel size of 328. Since the number 328 does not have special significance,[1] our results suggest that slightly contracting the panel in order to boost its degree of representation would be a good tradeoff. By contrast, in AUS3, the

---

[1] It likely started out as a "round" number and decreased when selected panelists dropped out.

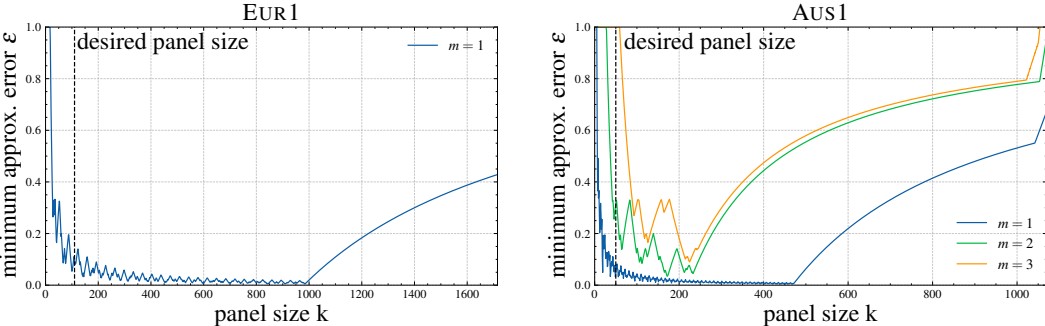

Figure 3: The smallest $\varepsilon$ for which $\varepsilon$-approximate $m$-representative panels exist as a function of the panel size $k$ for selected $m \in [|\mathcal{F}|]$. The dashed lines show the desired panel sizes.

largest 2-representative panel is of size 31, which is 69% of the desired panel size of 45. The pool of volunteers for AUS3 only contains 509 individuals — significantly fewer than the other assemblies, which all have more than 1000 volunteers. We conclude that an investment in the recruitment of volunteers may make a greater degree of representation possible.

Our results for $\varepsilon$-approximate $m$-representation give more nuanced insights on the tradeoff between accurate representation and panel size. The results for EUR1 and AUS1 are shown in Figure 3; the plots for the other instances can be found in Appendix A.6.4. First, we note that $\varepsilon$-approximate $m$-representation for $\varepsilon < 1$ is not possible if the total number of possible $m$-tuples of feature-values in the population is greater than $k$, or a feature-value $m$-tuple exists in the population but does not exist in the pool of volunteers, since in either case for some feature-value $m$-tuple the number of individuals on the panel is 0 while the quota is greater than 0. This is the case for $m \geq 2$ in EUR1 and AUS3. In all other instances, we find that the best achievable $\varepsilon$ decreases with the panel size (approximately inversely linearly) for panel sizes less than a *critical size* $k^*$; this is the panel size at which the pool is "too skewed" since for some feature-value $m$-tuple the quota is higher than the number of candidates in the pool with this feature-value $m$-tuple. For panels larger than this size $k^*$, $\varepsilon$ increases with the panel size (approximately proportional to $1 - \frac{k^*}{k}$). This critical size is the largest panel size up to which adding people to the panel comes with improved $\varepsilon$-approximate $m$-representation guarantees, and is slightly below the maximum size of an $m$-representative panel up to rounding in all instances where $\varepsilon < 1$ is possible. We conjecture that aiming for a mix of the two definitions of approximate $m$-representation will do best in practice.

## 6 Discussion

We conclude by discussing two points that may contribute to a fuller understanding of our approach and its limitations.

**Throwing fairness into the mix.** As mentioned in Section 1, citizens' assemblies are randomly selected. In practice, this is done by computing a lottery such that each panel in its support is 1-representative. A commonly used algorithm optimizes the lottery to maximize a fairness objective: the minimum probability of selecting any volunteer [10]. The same algorithm can be directly applied when the underlying notion of representation is $m$-representation for $m > 1$ instead of $m = 1$. However, this may come at a cost to the fairness objective, because the feasible set shrinks as the constraints become stronger. While we focused on the relation between representation and welfare, future work could explore the tradeoffs between representation, welfare and fairness.

**Empirically estimating welfare gains?** The reader may have noticed that, despite access to real data from citizens' assemblies, our empirical analysis does not evaluate our main theoretical claim: a greater degree of representation contributes to social welfare. The reason is that we do not have access to real utilities, and synthetic methods for generating them may seem contrived. Let us reiterate, however, that the lack of access to utilities is not a barrier to applying our results, as we argue for a greater degree of representation with respect to known features, which leads to improved social welfare with respect to *unknown* utilities.

## Acknowledgments

We would like to express our gratitude to Bailey Flanigan for her invaluable assistance in obtaining the datasets on the pools of volunteers in the four citizens' assemblies we analyze, by suggesting for which citizens' assemblies sufficient data on the underlying population is available, contacting the organizers of the citizens' assemblies for permission to use their data, and sharing the datasets with us. We also thank her for providing the dataset on the underlying population data for the European citizens' assembly. We furthermore thank the organizers of the citizens' assemblies for granting us permission to use their data.

Alexandros Psomas is supported in part by an NSF CAREER award CCF-2144208, and a research award from the Herbert Simon Family Foundation.

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

# A  Technical Appendices and Supplementary Material

## A.1  Proofs missing from Section 2

*Proof of Theorem 4.* Let $a, a' \in \mathcal{A}$ be any two alternatives such that $u_{\mathcal{C},p}(a) > u_{\mathcal{C},p}(a')$. Since the panel is $\varepsilon$-accurate for welfare estimation, we have that

$$u_{\mathcal{P},p}(a) \geq u_{\mathcal{C},p}(a) - \varepsilon > u_{\mathcal{C},p}(a') - \varepsilon \geq u_{\mathcal{P},p}(a') - 2\varepsilon,$$

which implies that $\mathcal{C}$ is $2\varepsilon$-accurate for pairwise comparisons. $\qquad\square$

*Proof of Theorem 5.* Since $\mathcal{C}$ is not $\varepsilon$-accurate, with respect to $p$ and $A$, for welfare estimation, there exists an alternative $a \in \mathcal{A}$ such that $|u_{\mathcal{C},p}(a) - u_{\mathcal{P},p}(a)| > \varepsilon$. Let $\delta > 0$ be any real number such that $\delta < |u_{\mathcal{C},p}(a) - u_{\mathcal{P},p}(a)| - \varepsilon$.

If $u_{\mathcal{C},p}(a) > u_{\mathcal{P},p}(a)$, consider an alternative $a'$ such that $a'(f, v) = (u_{\mathcal{P},p}(a) + \varepsilon + \delta)/|\mathcal{F}|$ for all $f \in \mathcal{F}$, $v \in V_f$. Then, $u_y(a') = u_{\mathcal{P},p}(a) + \varepsilon + \delta$ for all $y \in \mathcal{I}$, and therefore also $u_{\mathcal{P},p}(a') = u_{\mathcal{C},p}(a') = u_{\mathcal{P},p}(a) + \varepsilon + \delta$ (where we used the fact that, if all individuals have the same utility $x$ for an alternative, the $p$-mean of their utilities is also $x$). We have that $u_{\mathcal{C},p}(a) > u_{\mathcal{P},p}(a) + \varepsilon + \delta = u_{\mathcal{C},p}(a')$ but $u_{\mathcal{P},p}(a) < u_{\mathcal{P},p}(a) + \delta = u_{\mathcal{P},p}(a') - \varepsilon$, which contradicts the fact that $\mathcal{C}$ is $\varepsilon$-accurate for pairwise comparisons. Similarly, if $u_{\mathcal{C},p}(a) < u_{\mathcal{P},p}(a)$, consider an alternative $a'$ such that $a'(f, v) = (u_{\mathcal{P},p}(a) - \varepsilon - \delta)/|\mathcal{F}|$ for all $f \in \mathcal{F}$, $v \in V_f$. Then, $u_y(a') = u_{\mathcal{P},p}(a) - \varepsilon - \delta$ for all $y \in \mathcal{I}$, and therefore $u_{\mathcal{P},p}(a') = u_{\mathcal{C},p}(a') = u_{\mathcal{P},p}(a) - \varepsilon - \delta$. We have that $u_{\mathcal{C},p}(a) < u_{\mathcal{P},p}(a) - \varepsilon - \delta = u_{\mathcal{C},p}(a')$ but $u_{\mathcal{P},p}(a) - \varepsilon > u_{\mathcal{P},p}(a) - \varepsilon - \delta = u_{\mathcal{P},p}(a')$, a contradiction to $\mathcal{C}$ being $\varepsilon$-accurate for pairwise comparisons.

Lastly, we note that the alternatives $a'$ defined above don't make any individual's utility be more than 1 or less then $u_{\min}$. We know that $u_{\mathcal{C},p}(a), u_{\mathcal{P},p}(a) \in [u_{\min}, 1]$, so since $\min(u_{\mathcal{C},p}(a), u_{\mathcal{P},p}(a)) \leq u_y(a') \leq \max(u_{\mathcal{C},p}(a), u_{\mathcal{P},p}(a))$, we also have $u_y(a') \in [u_{\min}, 1]$ for all $y \in \mathcal{I}$. $\qquad\square$

## A.2  Calculations for Example 7

We first prove a short technical lemma to simplify the calculations.

**Lemma 12.** *Let $p < 1$ and $c_1, c_2, x \in \mathbb{R}$ so that $c_2 > c_1 \geq x \geq 0$ if $p \in (0, 1)$ and $c_2 > c_1 > x \geq 0$ if $p \leq 0$. Then for $p \neq 0$,*

$$\frac{d}{dx}\left(\frac{1}{4}((c_1 - x)^p + (c_1 + x)^p + (c_2 - x)^p + (c_2 + x)^p)\right)^{1/p} \leq 0,$$

*and equality holds only if $x = 0$. Furthermore,*

$$\frac{d}{dx}\left(((c_1 - x)(c_1 + x)(c_2 - x)(c_2 + x))^{1/4}\right) \leq 0,$$

*and again equality holds only if $x = 0$.*

*Proof.* We'll first denote $f(x) = \frac{1}{4}((c_1 - x)^p + (c_1 + x)^p + (c_2 - x)^p + (c_2 + x)^p)$ and consider the case $p \neq 0$. We can see that

$$\frac{d}{dx}(f(x))^{1/p} = \frac{1}{p}(f(x))^{1/p-1} \cdot \frac{1}{4}p\left((c_1 + x)^{p-1} - (c_1 - x)^{p-1} + (c_2 + x)^{p-1} - (c_2 - x)^{p-1}\right)$$

The two $p$'s cancel. Since all of the summand in $f(x)$ are non-negative (positive if $p \leq 0$) and $(c_2 + x)$ is positive, we have that $f(x) > 0$, so $(f(x))^{1/p-1} > 0$. It follows that the sign of $\frac{d}{dx}(f(x))^{1/p}$ is the same as the sign of $(c_1 + x)^{p-1} - (c_1 - x)^{p-1} + (c_2 + x)^{p-1} - (c_2 - x)^{p-1}$. Since $p < 1$, we know that the function $g(x) = x^{p-1}$ is strictly decreasing in $x$, so $(c_1 + x)^{p-1} \leq (c_1 - x)^{p-1}$ and $(c_2 + x)^{p-1} \leq (c_2 - x)^{p-1}$, with equality only if $x = 0$. The first part of the lemma follows.

For the second part of the lemma, we denote

$$f(x) = (c_1 - x)(c_1 + x)(c_2 - x)(c_2 + x) = (c_1^2 - x^2)(c_2^2 - x^2)$$

and get that

$$\frac{d}{dx}(f(x))^{1/4} = \frac{1}{4}(f(x))^{1/4-1}\left(4x^3 - 2x(c_1^2 + c_2^2)\right).$$

All the terms in the fourth root are positive, so we have that $f(x) > 0$ and thus $(f(x))^{1/4-1} > 0$. It follows that the sign of $\frac{d}{dx}(f(x))^{1/4}$ is the same as the sign of $\left(4x^3 - 2x(c_1^2 + c_2^2)\right)$. If $x = 0$, we are done. Else, since $c_2 \geq c_1 > x > 0$, we have that $4x^3 - 2x(c_1^2 + c_2^2) < 4x^3 - 2x(x^2 + x^2) = 0$. The second part of the lemma follows. $\qquad\square$

To show that the population indeed gets the strictly highest utility from $a_3$ for any $u_{\min} \in [0, 1)$ as claimed in Example 7, we let

$$h(x) = \begin{cases} \left(\frac{1}{4}((c_1 - x)^p + (c_1 + x)^p + (c_2 - x)^p + (c_2 + x)^p)\right)^{1/p} & \text{if } p \neq 0 \\ ((c_1 - x)(c_1 + x)(c_2 - x)(c_2 + x))^{1/4} & \text{if } p = 0 \end{cases},$$

where $c_1 = \frac{1+3u_{\min}}{4}$ and $c_2 = \frac{3+u_{\min}}{4}$. Note that we can express the welfare of the population for the alternatives in $\mathcal{A}$ using $h$:

$$u_{\mathcal{P},p}(a_1) = u_{\mathcal{P},p}(a_2) = h\left(\frac{1 - u_{\min}}{4}\right), \quad u_{\mathcal{P},p}(a_3) = h(0).$$

By Lemma 12, we know that $h$ is strictly decreasing on $[0, c_1)$ for $p \leq 0$ and on $[0, c_1]$ for $p \in (0, 1)$. For $p \in (0, 1)$ and any $u_{\min} \in [0, 1)$ we have that $c_1 \geq \frac{1-u_{\min}}{4}$, and for $p \leq 0$ and $u_{\min} \in (0, 1)$ we have that $c_1 > \frac{1-u_{\min}}{4}$, so we get that $h(0) > h\left(\frac{1-u_{\min}}{4}\right)$. Thus, $u_{\mathcal{P},p}(a_3) > u_{\mathcal{P},p}(a_1), u_{\mathcal{P},p}(a_2)$.

There remains the case $u_{\min} = 0$ and $p \leq 0$ that we already treat in Example 7 but repeat here for the sake of completeness. In that case, both $a_1$ and $a_2$ have a welfare of 0 for $\mathcal{P}$ since one individual in $\mathcal{P}$ has utility 0. Alternative $a_3$ has a welfare of at least $\frac{1}{4} > 0$ for $\mathcal{P}$ (using that the $p$-mean of a list of numbers is at least as big as the smallest number).

### A.3 Upper-bounding the $\varepsilon$-accuracy of $m$-representative panels

In order to prove the upper bounds in Theorems 8 and 9, we first establish our general framework for bounding the $\varepsilon$-accuracy of $m$-representative panels in Appendix A.3.1. We then turn to the cases $m = 1$ and $m = 2$ in Appendices A.3.3 and A.3.4, respectively, where we analytically solve the optimization problems bounding the $\varepsilon$-accuracy from the general framework for $m$-representative panel.

#### A.3.1 Properties of $m$-representative panels

We start by proving Lemma 13, a property of $m$-representative panels that is crucial for the proofs of Theorems 8 and 9.

**Lemma 13.** *Let $\mathcal{A}$ be a set of alternatives, $\mathcal{P}$ be a population, and $\mathcal{C}$ be an $m$-representative panel in $\mathcal{P}$. Then, for any $a \in \mathcal{A}$, the first $m$ moments of $\mathcal{C}$ and $\mathcal{P}$ for $a$ are the same. That is, for any integer $1 \leq \ell \leq m$,*

$$\sum_{y_i \in \mathcal{I}} q_i(u_i(a))^\ell = \sum_{y_i \in \mathcal{I}} r_i(u_i(a))^\ell.$$

*Proof of Lemma 13.* First, note that by definition a panel that is $m$-representative is also $\ell$-representative, for $\ell \leq m$. Thus, it suffices to show that the equation in Lemma 13 holds for $\ell = m$.

The key insight is that $\sum_{y_j \in \mathcal{I}} q_j(u_j(a))^m$ is a polynomial in $\{y_i(f, v)\}_{f \in \mathcal{F}, v \in V_f}$ of degree $m$. This allows us to use the definition of being $m$-representative to "switch" the $q_j$ in any term of the

polynomial to $r_j$. Formally, we get that

$$\sum_{y_j \in \mathcal{I}} q_j(u_j(a))^m = \sum_{y_j \in \mathcal{I}} q_j \left( \sum_{f \in \mathcal{F}} \sum_{v \in V_f} a(f,v) y_j(f,v) \right)^m$$

$$= \sum_{y_j \in \mathcal{I}} q_j \sum_{\substack{(f_1,...,f_m) \\ \in \mathcal{F}^m}} \sum_{\substack{(v_1,...,v_m) \\ \in V_{f_1} \times ... \times V_{f_m}}} \prod_{i=1}^{m} a(f_i, v_i) y_j(f_i, v_i)$$

$$= \sum_{\substack{(f_1,...,f_m) \\ \in \mathcal{F}^m}} \sum_{\substack{(v_1,...,v_m) \\ \in V_{f_1} \times ... \times V_{f_m}}} \left( \prod_{i=1}^{m} a(f_i, v_i) \right) \left( \sum_{y_j \in \mathcal{I}} q_j \prod_{i=1}^{m} y_j(f_i, v_i) \right).$$

and analogously for the panel

$$\sum_{y_j \in \mathcal{I}} r_j(u_j(a))^m = \sum_{\substack{(f_1,...,f_m) \\ \in \mathcal{F}^m}} \sum_{\substack{(v_1,...,v_m) \\ \in V_{f_1} \times ... \times V_{f_m}}} \left( \prod_{i=1}^{m} a(f_i, v_i) \right) \left( \sum_{y_j \in \mathcal{I}} r_j \prod_{i=1}^{m} y_j(f_i, v_i) \right). \quad (1)$$

We conclude the proof by arguing that for any $(f_1, ..., f_m) \in \mathcal{F}^m$ and $v_1, ..., v_m \in V_{f_1} \times ... \times V_{f_m}$, it holds that

$$\sum_{y_j \in \mathcal{I}} q_j \prod_{i=1}^{m} y_j(f_i, v_i) = \sum_{y_j \in \mathcal{I}} r_j \prod_{i=1}^{m} y_j(f_i, v_i).$$

Let's first assume there exist two indices $i \neq i'$ such that $f_i = f_{i'}$, but $v_i \neq v_{i'}$. In this case, we know both sides of Equation (1) are zero since no type $y_j \in \mathcal{I}$ can have $y_j(f, v_i) = y_j(f, v_{i'}) = 1$ for $v_i \neq v_{i'}$ (i.e., each feature takes exactly one value per type), where $f = f_i = f_{i'}$.

The case where whenever $f_i = f_{i'}$, also $v_i = v_{i'}$ remains. We know that $y_j(f_i, v_i) = y_j(f_{i'}, v_{i'}) \in \{0, 1\}$ for all $y_j \in \mathcal{I}$, and thus $y_j(f_i, v_i) = y_j(f_i, v_i) y_j(f_{i'}, v_{i'})$. Let $Z = \{f \in (f_1, ..., f_m)\}$ be the set of the unique features in $(f_1, ..., f_m)$, $|Z| \leq m$. We get that for any $(v_f)_{f \in Z} \in \bigtimes_{f \in Z} V_f$, it holds that

$$\sum_{y_j \in \mathcal{I}} q_j \prod_{i=1}^{m} y_j(f_i, v_i) = \sum_{y_j \in \mathcal{I}} q_j \prod_{f \in Z} y_j(f, v_f) = \sum_{y_j \in \mathcal{I}} r_j \prod_{f \in Z} y_j(f, v_f) = \sum_{y_j \in \mathcal{I}} r_j \prod_{i=1}^{m} y_j(f_i, v_i),$$

where the second equality follows from the definition of $m$-representation (see Definition 2), and the fact that $|Z| \leq m$. $\square$

From Lemma 13, we have that the first $m$ moments of an $m$-representative panel $\mathcal{C}$ in $\mathcal{P}$ are the same as the moments of $\mathcal{P}$. Also note that the 0-th moment is always identical, $\kappa_0 = \sum_{y_j \in \mathcal{I}} q_j = 1$. We'll now use moments to bound the utility an $m$-representative panel can have.

**Lemma 14.** *Let $u_{\min} \in [0, 1]$ and $p < 1$. Let $\mathcal{P}$ be a population (resp., let $\mathcal{C}$ be a panel) with moments $\kappa_0 = 1, \kappa_1, ..., \kappa_m$. Then for any alternative $a$ such that $u_i(a) \in [u_{\min}, 1]$ for all $y_i \in \mathcal{I}$, it holds that*

$$\text{UMIN}(\kappa_1, ..., \kappa_m; p, u_{\min}) \leq u_{\mathcal{P},p}(a), u_{\mathcal{C},p}(a) \leq \text{UMAX}(\kappa_1, ..., \kappa_m; p, u_{\min}),$$

*where*[2] $\text{UMIN}(\kappa_1, ..., \kappa_m; p, u_{\min})$ *and* $\text{UMAX}(\kappa_1, ..., \kappa_m; p, u_{\min})$ *are,* *respectively,*

$$\min \quad \left( \sum_{i=0}^{m} q_i x_i^p \right)^{\frac{1}{p}} \qquad\qquad \max \quad \left( \sum_{i=0}^{m} q_i x_i^p \right)^{\frac{1}{p}}$$

$$\text{s.t.} \quad \sum_{i=0}^{m} q_i x_i^j = \kappa_j \quad \text{for all } j \in \{0, ..., m\} \qquad \text{s.t.} \quad \sum_{i=0}^{m} q_i x_i^j = \kappa_j \quad \text{for all } j \in \{0, ..., m\}$$

$$q_i \geq 0 \quad \text{for all } i \in \{0, ..., m\} \qquad\qquad q_i \geq 0 \quad \text{for all } i \in \{0, ..., m\}$$

$$x_i \in [u_{\min}, 1] \quad \text{for all } i \in \{0, ..., m\}. \qquad x_i \in [u_{\min}, 1] \quad \text{for all } i \in \{0, ..., m\}.$$

*Proof.* We'll prove the theorem for any population $\mathcal{P}$. The proof for a panel $\mathcal{C}$ is identical.

We know that for any population $\mathcal{P}$ with moments $\kappa_0, \kappa_1, ..., \kappa_m$ it holds that the utility $u_{\mathcal{P},p}(a)$ is upper and lower bounded by maximizing and minimizing, respectively, the following optimization problem

$$\max / \min \quad \left( \sum_{y_j \in \mathcal{I}} q_j \cdot (u_j)^p \right)^{\frac{1}{p}}$$

$$\text{s.t.} \quad \sum_{y_j \in \mathcal{I}} q_j \cdot (u_j)^i = \kappa_i \quad \text{for all } i \in \{0, 1, ..., m\}$$

$$u_j \in [u_{\min}, 1] \quad \text{for all } y_j \in \mathcal{I}$$

since the feasible region is exactly the region of all populations with moments as given, and the utilities are allowed to take any value within $[u_{\min}, 1]$.

Since $\sum_{y_j \in \mathcal{I}} q_j = 1$ (this is enforced above by the moment constraint for $i = 0$), we can interpret the $q_j$ and $u_j$ in the above program as a probability measure over $[u_{\min}, 1]$, placing probability mass $q_j$ on $u_j$. We can use this observation to relax the constraints of the above program by replacing the $q_i$ and $u_i$ with any probability measure on $[u_{\min}, 1]$. Note that this is essentially equivalent to allowing $n$, the individuals in the population, go to infinity. Thus, we get that $u_{\mathcal{P},p}(a)$ is upper and lower bounded by maximizing and minimizing, respectively, the following optimization problem

$$\max / \min \quad \left( \int_{u_{\min}}^{1} x^p d\mu(x) \right)^{\frac{1}{p}}$$

$$\text{s.t.} \quad \int_{u_{\min}}^{1} x^i d\mu(x) = \kappa_i \quad \text{for all } i \in \{0, 1, ..., m\}$$

$$\mu(x) \geq 0 \quad \text{for all } x \in [u_{\min}, 1].$$

Let $M$ be the feasible region of the optimization program(s) above (noting that continuous, discrete, and mixed probability measures are all in $M$). If we take the $\mu$ corresponding to some population, we see that the moments of the population are identical to the moments of the probability distribution $\mu \in M$.

Problems of the above type, where a space of probability measures over a bounded interval is defined by their moments, are called moment problems [22]. It is known that the space of such probability measures is convex, with the extreme points being distributions supported on a discrete set of size at most $m + 1$, where $m$ is the number of moments [17, Theorem 2.1 for a general $u_{\min}$, Proposition 3.2 for $u_{\min} = 0$].

It remains to show that the objective function of the above program takes its extreme values at the extreme points of the constraint set. To see this, let $\mu_1, \mu_2 \in M$ be any two distributions, and for

---

[2]To keep the notation concise, we'll write the following proofs for a general $p$-mean and don't explicitly state the special case $p = 0$ whenever the corresponding mathematical statements can immediately be obtained from replacing the $p$-mean with the geometric mean.

some $t \in [0, 1]$, let $\mu = t\mu_1 + (1-t)\mu_2 \in M$ be a convex combination of $\mu_1, \mu_2$. Then, the objective function (i.e. $p$-mean) of $\mu$ is

$$\left(\int_{u_{\min}}^1 x^p d\mu\right)^{\frac{1}{p}} = \left(t\int_{u_{\min}}^1 x^p d\mu_1 + (1-t)\int_{u_{\min}}^1 x^p \mu_2\right)^{\frac{1}{p}}$$

$$= \left(t\left(\int_{u_{\min}}^1 x^p d\mu_1\right)^{\frac{1}{p}p} + \left((1-t)\int_{u_{\min}}^1 x^p \mu_2\right)^{\frac{1}{p}p}\right)^{\frac{1}{p}}.$$

In particular, the objective function for $\mu$ is the $p$-mean of the objective functions for $\mu_1$ and $\mu_2$, weighted with $t$ and $1 - t$, respectively, and as such, always between the two.

Thus, we conclude that the above optimization program takes its maximum and minimum values at extreme points of the constraint set, which is discrete distributions with support size at most $m + 1$. Applying this yields the optimization programs in the theorem statement. $\qquad\square$

Lemma 14 gives us an upper and lower bound on the utility of a population or panel with given first $m$-moments. We can use Lemma 14 to prove the following lemma.

**Lemma 15.** *Let $u_{\min} \in [0, 1]$ and $p < 1$. Let $\mathcal{P}$ be a population and let $\mathcal{C}$ be a $m$-representative panel in $\mathcal{P}$. Let $a$ be any alternative such that $u_j(a) \in [u_{\min}, 1]$ for all $y_j \in \mathcal{I}$. Then $\mathcal{C}$ is $\varepsilon$-accurate with*

$$\varepsilon = \max \quad \text{UMAX}(\kappa_1, ..., \kappa_m; p, u_{\min}) - \text{UMIN}(\kappa_1, ..., \kappa_m; p, u_{\min})$$

$$s.t. \quad [\kappa_{i+j}]_{i,j=0}^{m/2}, [(1+u_{\min})\kappa_{i+j+1} - \kappa_{i+j+2} - u_{\min}\kappa_{i+j}]_{i,j=0}^{m/2-1} \succeq 0 \quad \textit{if } m \equiv 0 \mod 2$$

$$[\kappa_{i+j+1} - u_{\min}\kappa_{i+j}]_{i,j=0}^{(m-1)/2}, [\kappa_{i+j} - \kappa_{i+j+1}]_{i,j=0}^{(m-1)/2} \succeq 0 \qquad \textit{if } m \equiv 1 \mod 2$$

*where $[M_{i,j}]_{i,j=0}^{d-1}$ is a $d \times d$ matrix with entry $M_{i,j}$ at position $(i, j)$ and $M \succeq 0$ means that $M$ is positive-semidefinite.*

*Proof.* Lemma 14 implies that for a given sequence of moments $\kappa_1, ..., \kappa_m$ it holds that

$$|u_{\mathcal{C},p}(a) - u_{\mathcal{P},p}(a)| \le \text{UMAX}(\kappa_1, ..., \kappa_m; p, u_{\min}) - \text{UMIN}(\kappa_1, ..., \kappa_m; p, u_{\min}).$$

Thus, $\mathcal{C}$ is guaranteed to be $\varepsilon$-accurate with

$$\varepsilon = \max \quad \text{UMAX}(\kappa_1, ..., \kappa_m; p, u_{\min}) - \text{UMIN}(\kappa_1, ..., \kappa_m; p, u_{\min})$$

$$s.t. \quad \kappa_1, ..., \kappa_m \text{ are the moments of a population (or panel) and alternative, given } u_{\min}$$

We can relax the constraints on $\kappa_1, ..., \kappa_m$, similarly to the proof of Lemma 14, by only requiring $\kappa_1, ..., \kappa_m$ to be the moments of a probability measure $\mu$ on $[u_{\min}, 1]$. Determining whether such a probability measure $\mu$ exists given moments and a bounded interval is known as the one-dimensional truncated moment problem on a bounded interval; we refer to [21] for a concise and contemporary proof that the constraints in the lemma statement are necessary and sufficient for $\kappa_1, ..., \kappa_m$ to be the moments of a probability measure $\mu$ on $[u_{\min}, 1]$. $\qquad\square$

### A.3.2 Solving for UMIN and UMAX

The optimization program in Lemma 15 is by itself not overly insightful since there are two further optimization programs nested in it, from the definitions of $\text{UMAX}(\kappa_1, ..., \kappa_m; p, u_{\min})$ and $\text{UMIN}(\kappa_1, ..., \kappa_m; p, u_{\min})$. In this section, we describe a general framework for getting closed form solutions for UMAX and UMIN, which we apply in subsequent sections for $m = 1$ and $m = 2$.

We start by noting that the feasible regions of $\text{UMAX}(\kappa_1, ..., \kappa_m; p, u_{\min})$ and $\text{UMIN}(\kappa_1, ..., \kappa_m; p, u_{\min})$ are identical. There are $m + 1$ moment conditions, in the form of $m + 1$ equality constraints, all linear in the $m + 1$ $q_i$'s. W.l.o.g. we assume that $x_0 > x_1 > ... > x_m$ (we don't need to consider equality since it is equivalent to setting a $q_i$ to 0), so we get that the $m + 1$ moment equality constraints for fixed $x_i$ are linearly independent since the coefficients are $x_i^j$ for $j = 0, ..., m$ and the $x_i$ are distinct. We can thus solve uniquely for the $q_i$ to get

$$q_i = \frac{1}{\prod_{j \neq i}(x_i - x_j)} \sum_{j=0}^m (-1)^j \kappa_{m-j} e_j(x_{-i}), \tag{2}$$

where $e_j$ is the $j$-th elementary symmetric polynomial and $x_{-i}$ are all $x_0, \ldots, x_m$ except $x_i$:

$$e_j(x_{-i}) = e_j(x_0, ..., x_{i-1}, x_{i+1}, ..., x_m) = \sum_{\substack{0 \leq i_1 < ... < i_j \leq m \\ i_1, ..., i_j \neq i}} x_{i_1}...x_{i_j}.$$

Defining the $q_i$ as a function of $x = (x_0, ..., x_m)$, we can rewrite the optimization problems with only $m + 1$ instead of $2m + 2$ variables so that

$$\text{UMIN}(\kappa_1, ..., \kappa_m; p, u_{\min}) =$$
$$\min \quad \left( \sum_{i=0}^{m} q_i(x) x_i^p \right)^{\frac{1}{p}}$$
$$\text{s.t.} \quad q_i(x) \geq 0 \quad \text{for all } i \in \{0, ..., m\}$$
$$1 \geq x_0 > x_1 > \cdots > x_m \geq u_{\min},$$

$$\text{UMAX}(\kappa_1, ..., \kappa_m; p, u_{\min}) =$$
$$\max \quad \left( \sum_{i=0}^{m} q_i(x) x_i^p \right)^{\frac{1}{p}}$$
$$\text{s.t.} \quad q_i(x) \geq 0 \quad \text{for all } i \in \{0, ..., m\}$$
$$1 \geq x_0 > x_1 > \cdots > x_m \geq u_{\min}.$$

To get an analytical solution for this optimization problem, we note that for $m = 1$ and $m = 2$, the derivative of the objective with respect to each variable is either non-negative or non-positive on the entire feasible region.

**Lemma 16.** *Let $p \leq 1$ and $u_{\min} \in (0, 1)$, or $p \in (0, 1]$ and $u_{\min} = 0$. Let $m \in \{1, 2\}$ and $1 \geq x_0 > x_1 > \cdots > x_m \geq u_{\min}$. Then for all $x = (x_0, \ldots, x_m)$ so that $q_j(x) \geq 0$ for all $j \in \{0, ..., m\}$, it holds that*

$$\frac{\partial}{\partial x_i} \left( \sum_{i=0}^{m} q_i(x) x_i^p \right)^{\frac{1}{p}} \begin{cases} \geq 0 & \text{if } i \equiv m \pmod 2 \\ \leq 0 & \text{if } i \not\equiv m \pmod 2 \end{cases},$$

*for all $i = 0, ..., m$, where the $q_i$ are a function of $x_0, ..., x_m$ as defined in Equation (2).*

We conjecture that Lemma 16 also holds for $m > 2$ and can be proven almost analogously to our proof for $m = 2$. For our results, we only investigate 1-representative and 2-representative panels and prove Lemma 16 for the respective $m$ in the corresponding sections. We then apply it to solve explicitly for UMIN and UMAX.

### A.3.3 1-representative panels

We'll first consider the case $m = 1$. We prove Lemma 16 for $m = 1$ and then use it to solve for UMIN and UMAX. By Lemma 15, the proof of the first, "upper bound", part of Theorem 8 will follow (the proof of the second, "lower bound", part can be found in Appendix A.4).

*Proof of Lemma 16, $m = 1$.* Recall that w.l.o.g. we assume $x_0 > x_1$. We can solve for $q_i$ as in Equation (2) to get

$$q_0 = \frac{\kappa_1 - x_1}{x_0 - x_1} \qquad\qquad q_1 = 1 - q_0 = \frac{x_0 - \kappa_1}{x_0 - x_1}.$$

We'll now first treat the case $p \neq 0$ and return to $p = 0$ at the end.

Let's calculate the derivatives of the objective without the outer $p$-th root:

$$\frac{\partial}{\partial x_0}(q_0 x_0^p + q_1 x_1^p) = p x_0^{p-1} \frac{\kappa_1 - x_1}{x_0 - x_1} + x_0^p \frac{x_1 - \kappa_1}{(x_0 - x_1)^2} + x_1^p \frac{\kappa_1 - x_1}{(x_0 - x_1)^2}$$

$$= \frac{\kappa_1 - x_1}{x_0 - x_1} \left( p x_0^{p-1} - \frac{x_0^p - x_1^p}{x_0 - x_1} \right)$$

$$= q_0 \left( p x_0^{p-1} - \frac{x_0^p - x_1^p}{x_0 - x_1} \right),$$

$$\frac{\partial}{\partial x_1}(q_0 x_0^p + q_1 x_1^p) = q_1 \left( p x_1^{p-1} - \frac{x_0^p - x_1^p}{x_0 - x_1} \right).$$

To determine the sign of the two expressions in parenthesis, we let $f(x) = x^p$ and note that $\frac{df}{dx}(x_0) = px_0^{p-1}$ and $\frac{df}{dx}(x_1) = px_1^{p-1}$ while $\frac{x_0^p - x_1^p}{x_0 - x_1}$ is the difference quotient of $f$ for $[x_1, x_0]$. For $p \in (0, 1)$, $f$ is concave, so we get $px_0^{p-1} < \frac{x_0^p - x_1^p}{x_0 - x_1} < px_1^{p-1}$; for $p < 0$, $f$ is convex, so $px_0^{p-1} > \frac{x_0^p - x_1^p}{x_0 - x_1} > px_1^{p-1}$. Since $q_0, q_1 \geq 0$, we get that

$$\frac{\partial}{\partial x_i}(q_0 x_0^p + q_1 x_1^p) \begin{cases} \leq 0 & \text{if } (i = 0 \wedge p \in (0,1)) \vee (i = 1 \wedge p < 0) \\ \geq 0 & \text{if } (i = 1 \wedge p \in (0,1)) \vee (i = 0 \wedge p < 0) \end{cases}.$$

To finish the proof, we note that

$$\frac{\partial}{\partial x_0}(q_0 x_0^p + q_1 x_1^p)^{\frac{1}{p}} = \frac{1}{p}(q_0 x_0^p + q_1 x_1^p)^{\frac{1}{p} - 1} \cdot \frac{\partial}{\partial x_0}(q_0 x_0^p + q_1 x_1^p),$$

so if $p < 0$ the sign switches when considering the objective function instead of the objective function without the $p$-th root. This implies the lemma for $p \neq 0$.

Let's now turn to the case $p = 0$. We note that the sign of the objective doesn't change when taking the logarithm, so it suffices to show that

$$\frac{\partial}{\partial x_i}(q_0 \log(x_0) + q_1 \log(x_1)) \begin{cases} \leq 0 & \text{if } i = 0 \\ \geq 0 & \text{if } i = 1 \end{cases}.$$

Taking the derivatives, we get

$$\frac{\partial}{\partial x_0}(q_0 \log(x_0) + q_1 \log(x_1)) = q_0 \left( \frac{1}{x_0} - \frac{\log(x_0) - \log(x_1)}{x_0 - x_1} \right),$$

$$\frac{\partial}{\partial x_1}(q_0 \log(x_0) + q_1 \log(x_1)) = q_1 \left( \frac{1}{x_1} - \frac{\log(x_0) - \log(x_1)}{x_0 - x_1} \right).$$

If we let $f(x) = \log(x)$, again, the expressions in the parenthesis is the difference between the derivative of $f$ at $x_0$ and $x_1$, respectively, minus the difference quotient of $f$ for $[x_1, x_0]$. Since $f$ is concave, the lemma for $p = 0$ follows. $\square$

We now apply Lemma 16 to solve for UMIN and UMAX.

**Lemma 17.** *Let $p \leq 1$ and $u_{\min} \in (0, 1)$, or $p \in (0, 1]$ and $u_{\min} = 0$. Let $\kappa_1 \in (u_{\min}, 1)$. Then $\text{UMAX}(\kappa_1; p, u_{\min}) = \kappa_1$ and*

$$\text{UMIN}(\kappa_1; p, u_{\min}) = \begin{cases} \left( \frac{\kappa_1 - u_{\min}}{1 - u_{\min}} + \frac{1 - \kappa_1}{1 - u_{\min}} \cdot u_{\min}^p \right)^{\frac{1}{p}} & \text{if } p \neq 0 \\ (u_{\min})^{\frac{1 - \kappa_1}{1 - u_{\min}}} & \text{if } p = 0 \end{cases}$$

*Proof.* W.l.o.g. we assume $x_0 > x_1$. Thus, the constraints $q_i \geq 0$ are equivalent to $\kappa_1 \geq x_1$ and $x_0 \geq \kappa_1$. We solve for UMIN and UMAX by finding the minimum and maximum of their defining optimization program from Lemma 14.

*Finding the maximum:* By Lemma 16, we know that the objective (non-strictly) increases as $x_0$ decreases. The only lower bounds on $x_0$ are $x_0 \geq \kappa_1$ and $x_0 > x_1$, so we know there is a maximum either at $x_0 = \kappa_1$ (where the first lower bound is tight) or for $x_1 \geq \kappa_1$ (the only case in which $x_0$ can't be tight to the first lower bound since $x_0 > x_1$). Since $\kappa_1$ is the weighted mean of $x_0$ and $x_1$, we know that $x_1 \leq \kappa_1$, so the second case becomes $x_1 = \kappa_1$. These two cases, $x_0 = \kappa_1$ and $x_1 = \kappa_1$, imply, respectively, $q_1 = 0$ and $q_0 = 0$, so the other variable doesn't affect the objective. We thus get that a maximum is attained at

$$x_0 = \kappa_1, \quad q_0 = 1, \quad x_1 \in [u_{\min}, \kappa_1), \quad q_1 = 0 \quad \text{or} \quad x_0 \in (\kappa_1, 1], \quad q_0 = 0, \quad x_1 = \kappa_1, \quad q_1 = 1.$$

In both cases, the objective value is $\kappa_1$.

*Finding the minimum:* By Lemma 16, we know that the objective (non-strictly) decreases as $x_0$ increases. The only upper bound on $x_0$ is $x_0 \leq 1$. Similarly, we know that the objective (non-strictly) decreases as $x_1$ decreases. The only lower bound on $x_1$ is $x_1 \geq u_{\min}$. Thus, we know that the objective takes its minimum value at

$$x_0 = 1, \qquad q_0 = \frac{\kappa_1 - u_{\min}}{1 - u_{\min}}, \qquad x_1 = u_{\min}, \qquad q_1 = \frac{1 - \kappa_1}{1 - u_{\min}}.$$

Plugging these values into the objective gives the lemma statement. $\square$

We know have the tools to prove the first part of the main theorem, Theorem 8, the upper bound on $\varepsilon$-accuracy of a 1-representative panel.

*Proof of Theorem 8, upper bound.* We first consider the case $p \leq 0$ and $u_{\min} = 0$. Since $u_{\mathcal{C},p}(a), u_{\mathcal{P},p}(a) \in [0,1]$ for all $a \in \mathcal{A}$, we know that $|u_{\mathcal{C},p}(a) - u_{\mathcal{P},p}(a)| \leq 1$ for all $a \in \mathcal{A}$. Thus, $\mathcal{C}$ is 1-accurate.

Let's now consider the case $p < 1$ and $u_{\min} \in (0,1)$ or $p \in (0,1)$ and $u_{\min} = 0$. By Lemma 15, we know that any 1-representative panel $\mathcal{C}$ is $\varepsilon$-accurate with

$$\varepsilon = \max \quad \text{UMAX}(\kappa_1; p, u_{\min}) - \text{UMIN}(\kappa_1, \kappa_2; p, u_{\min})$$
$$\text{s.t.} \quad [\kappa_{i+j+1} - u_{\min}\kappa_{i+j}]_{i,j=0}^0, [\kappa_{i+j} - \kappa_{i+j+1}]_{i,j=0}^0 \succeq 0$$

where $[M_{i,j}]_{i,j=0}^{d-1}$ is a $d \times d$ matrix with entry $M_{i,j}$ at position $(i,j)$ and $M \succeq 0$ means that $M$ is positive-semi-definite. Since both matrices are $1 \times 1$ and $\kappa_0 = 1$, the constraints are equivalent to $\kappa_1 \geq u_{\min}$ and $1 \geq \kappa_1$.

If both constraints are strict, we know the values of UMIN and UMAX by Lemma 17. There remains the case when at least one of the two inequalities is not strict (recall Lemma 14 for the definitions of UMAX and UMIN in this case). Let's again assume w.l.o.g. that $x_0 > x_1$. Since $\kappa_1$ is the weighted mean of $x_0$ and $x_1$ (with weights $q_0$ and $q_1$, respectively), and both $x_0, x_1 \in [u_{\min}, 1]$, we know that $\kappa_1 = u_{\min}$ implies $x_1 = u_{\min}, q_0 = 0$, so $\text{UMIN}(u_{\min}; p, u_{\min}) = \text{UMAX}(u_{\min}; p, u_{\min}) = u_{\min}$. Similarly, $\kappa_1 = 1$ implies $x_0 = 1, q_1 = 0$, so $\text{UMIN}(1; p, u_{\min}) = \text{UMAX}(1; p, u_{\min}) = 1$.

In either case, if one of the two constraints is not tight, $\text{UMAX}(\kappa_1, \kappa_2; p, u_{\min}) - \text{UMIN}(\kappa_1, \kappa_2; p, u_{\min}) = 0$. Since there always exists a point at which all constraints are strictly followed, and $\text{UMAX}(\kappa_1, \kappa_2; p, u_{\min}) - \text{UMIN}(\kappa_1, \kappa_2; p, u_{\min}) \geq 0$ at any such point, as follows immediately from their definition, we can restrict the constraints to being strict, to get that any 1-representative panel is $\varepsilon$-accurate for $p \neq 0$ with

$$\max_{\kappa_1} \quad \kappa_1 - \left( \frac{\kappa_1 - u_{\min}}{1 - u_{\min}} + \frac{1 - \kappa_1}{1 - u_{\min}} \cdot u_{\min}{}^p \right)^{\frac{1}{p}}$$
$$\text{s.t.} \quad u_{\min} < \kappa_1 < 1,$$

and for $p = 0$ with

$$\max_{\kappa_1} \quad \kappa_1 - u_{\min}{}^{\frac{1 - \kappa_1}{1 - u_{\min}}}$$
$$\text{s.t.} \quad u_{\min} < \kappa_1 < 1.$$

We'll now solve these two optimization programs, starting with the case $p \neq 0$. We let $f(\kappa_1) = \kappa_1 - \left( \frac{\kappa_1 - u_{\min}}{1 - u_{\min}} + \frac{1 - \kappa_1}{1 - u_{\min}} \cdot u_{\min}{}^p \right)^{\frac{1}{p}}$ be the objective of the optimization program. We take the first derivative of $f$, to get

$$\frac{df}{d\kappa_1} = 1 - \frac{1}{p} \left( \frac{\kappa_1 - u_{\min}}{1 - u_{\min}} + \frac{1 - \kappa_1}{1 - u_{\min}} \cdot u_{\min}{}^p \right)^{\frac{1}{p} - 1} \frac{1 - u_{\min}{}^p}{1 - u_{\min}}.$$

Setting this equal to zero, we get that

$$p \frac{1 - u_{\min}}{1 - u_{\min}{}^p} = \left( \frac{\kappa_1 - u_{\min}}{1 - u_{\min}} + \frac{1 - \kappa_1}{1 - u_{\min}} \cdot u_{\min}{}^p \right)^{\frac{1}{p} - 1}.$$

We know that both the left hand side as well as everything within the parenthesis on the right hand side is positive for $\kappa_1 \in (u_{\min}, 1)$, so we can raise the above equality to the $(1 - p)/p$-th power without introducing or loosing solutions for $\kappa_1 \in (u_{\min}, 1)$. We do so and solve for $\kappa_1$ to get

$$\kappa_1 = \frac{1 - u_{\min}}{1 - u_{\min}{}^p} \left( p \cdot \frac{1 - u_{\min}}{1 - u_{\min}{}^p} \right)^{\frac{p}{1 - p}} + \frac{u_{\min} - u_{\min}{}^p}{1 - u_{\min}{}^p}. \tag{3}$$

Taking the second derivative of $f$, we see that for $\kappa_1 \in [u_{\min}, 1]$,

$$\frac{d^2 f}{(d\kappa_1)^2} = -\frac{1}{p}\left(\frac{1}{p} - 1\right)\left(\frac{\kappa_1 - u_{\min}}{1 - u_{\min}} + \frac{1 - \kappa_1}{1 - u_{\min}} \cdot u_{\min}{}^p\right)^{\frac{1}{p} - 2}\left(\frac{1 - u_{\min}{}^p}{1 - u_{\min}}\right)^2 \leq 0,$$

so $f$ is concave on $[u_{\min}, 1]$. We can see that $f(u_{\min}) = f(1) = 0$, and we know from above that $f(x) > 0$ for some $x \in (u_{\min}, 1)$. Thus, we know that $f$ has an extreme point on $[u_{\min}, 1]$, which we know has to be at $\kappa_1$ as specified in Equation (3), and $f$ is maximized on $(u_{\min}, 1)$ for this $\kappa_1$.

Plugging Equation (3) into $f$, we get that

$$\max_{u_{\min} < \kappa_1 < 1} f(\kappa_1) = \frac{1 - p}{p}\left(p \cdot \frac{1 - u_{\min}}{1 - u_{\min}{}^p}\right)^{\frac{1}{1-p}} + \frac{u_{\min} - u_{\min}{}^p}{1 - u_{\min}{}^p},$$

which gives $\varepsilon$ from the theorem statement, for $p < 0$ and $u_{\min} \in (0, 1)$ or $p \in (0, 1)$ and $u_{\min} \in [0, 1)$.

Let's now solve the optimization program in the remaining case $p = 0$ and $u_{\min} \in (0, 1)$. Again, we let $f(\kappa_1) = \kappa_1 - u_{\min}{}^{\frac{1 - \kappa_1}{1 - u_{\min}}}$ be the objective of corresponding optimization program. We take the first derivative of $f$, to get

$$\frac{df}{d\kappa_1} = 1 + \frac{1}{1 - u_{\min}} u_{\min}{}^{\frac{1 - \kappa_1}{1 - u_{\min}}} \ln(u_{\min}).$$

Setting this equal to zero and solving for $\kappa_1$, we get

$$\kappa_1 = 1 - (1 - u_{\min})\frac{\ln\left(\frac{1}{1 - u_{\min}}\right) + \ln\ln\left(\frac{1}{u_{\min}}\right)}{\ln\left(\frac{1}{u_{\min}}\right)} \tag{4}$$

Taking the second derivative of $f$, we see that for $\kappa_1 \in [u_{\min}, 1]$,

$$\frac{d^2 f}{(d\kappa_1)^2} = -\left(\frac{1}{1 - u_{\min}}\right)^2 u_{\min}{}^{\frac{1 - \kappa_1}{1 - u_{\min}}} \ln^2(u_{\min}) \leq 0,$$

so $f$ is concave on $[u_{\min}, 1]$. Similar to before, we can see that $f(u_{\min}) = f(1) = 0$, and we know from above that $f(x) > 0$ for some $x \in (u_{\min}, 1)$. Thus, we know that $f$ has an extreme points on $[u_{\min}, 1]$, which we know has to be at $\kappa_1$ as specified in Equation (4), and $f$ is maximized on $(u_{\min}, 1)$ for this $\kappa_1$.

Plugging Equation (4) into $f$, we get that

$$\max_{u_{\min} < \kappa_1 < 1} f(\kappa_1) = 1 + \frac{1 - u_{\min}}{\ln\left(\frac{1}{u_{\min}}\right)}\left(\ln\left(\frac{1 - u_{\min}}{\ln\left(\frac{1}{u_{\min}}\right)}\right) - 1\right),$$

which gives $\varepsilon$ from the theorem statement for $p = 0$ and $u_{\min} \in (0, 1)$. □

### A.3.4  2-representative panels

Let's consider $m = 2$. We will first prove Lemma 16 for $m = 2$, which we'll then apply to solve for UMIN and UMAX. Together with Lemma 15 the proof of the first, "upper bound", part of Theorem 9 will follow (the proof of the second, "lower bound", part can be found in Appendix A.4).

*Proof of Lemma 16, $m = 2$.* From Equation (2) we obtain that

$$q_0 = \frac{1}{(x_0 - x_1)(x_0 - x_2)}(\kappa_2 - \kappa_1(x_1 + x_2) + x_1 x_2)$$

$$q_1 = \frac{1}{(x_1 - x_0)(x_1 - x_2)}(\kappa_2 - \kappa_1(x_0 + x_2) + x_0 x_2)$$

$$q_2 = \frac{1}{(x_2 - x_0)(x_2 - x_1)}(\kappa_2 - \kappa_1(x_0 + x_1) + x_0 x_1).$$

We now calculate the derivative of the objective without the outer $p$-th root. We'll treat the case $p = 0$ separately at the end.

$$\frac{\partial}{\partial x_0}\left(\sum_{i=0}^{2} q_i x_i^p\right) = q_0 p x_0^{p-1} + x_0^p q_0 \frac{(x_1 - x_0) + (x_2 - x_0)}{(x_0 - x_1)(x_0 - x_2)}$$

$$+ q_1 \frac{x_1^p}{(x_1 - x_0)} + \frac{x_1^p (x_2 - \kappa_1)}{(x_1 - x_0)(x_1 - x_2)}$$

$$+ q_2 \frac{x_2^p}{(x_2 - x_0)} + \frac{x_2^p (x_1 - \kappa_1)}{(x_2 - x_0)(x_2 - x_1)}$$

$$= q_0 \left( p x_0^{p-1} - \frac{x_0^p}{x_0 - x_1} - \frac{x_0^p}{x_0 - x_2}\right)$$

$$+ \frac{x_1^p}{(x_1 - x_0)(x_1 - x_2)}\left(\frac{q_1(x_1 - x_2) + (x_2 - \kappa_1)}{q_0}\right)$$

$$+ \frac{x_2^p}{(x_2 - x_0)(x_2 - x_1)}\left(\frac{q_2(x_2 - x_1) + (x_1 - \kappa_1)}{q_0}\right)\Bigg)$$

$$= q_0 \left( p x_0^{p-1} - \frac{x_0^p}{x_0 - x_1} - \frac{x_0^p}{x_0 - x_2}\right.$$

$$+ \frac{x_1^p(x_2 - x_0)}{(x_1 - x_0)(x_1 - x_2)}\left(\frac{(\kappa_2 - \kappa_1(x_0 + x_2) + x_0 x_2) + (x_2 - \kappa_1)(x_1 - x_0)}{\kappa_2 - \kappa_1(x_1 + x_2) + x_1 x_2}\right)$$

$$+ \frac{x_2^p(x_1 - x_0)}{(x_2 - x_0)(x_2 - x_1)}\left(\frac{(\kappa_2 - \kappa_1(x_0 + x_1) + x_0 x_1) + (x_1 - \kappa_1)(x_2 - x_0)}{\kappa_2 - \kappa_1(x_1 + x_2) + x_1 x_2}\right)\Bigg)$$

$$= q_0 \left( p x_0^{p-1} - \frac{x_0^p - x_1^p \frac{x_0 - x_2}{x_1 - x_2}}{x_0 - x_1} - \frac{x_0^p - x_2^p \frac{x_0 - x_1}{x_2 - x_1}}{x_0 - x_2}\right).$$

To determine the sign of the last line we consider the function $f : \mathbb{R}_{\geq 0} \to \mathbb{R}_{\geq 0}$ where $f(x) = x^p$. We let $P$ be the unique degree-2 polynomial for which $P(x_0) = f(x_0)$, $P(x_1) = f(x_1)$, $P(x_2) = f(x_2)$ using the Lagrange basis polynomials:

$$P(x) = x_0^p \frac{(x - x_1)(x - x_2)}{(x_0 - x_1)(x_0 - x_2)} + x_1^p \frac{(x - x_0)(x - x_2)}{(x_1 - x_0)(x_1 - x_2)} + x_2^p \frac{(x - x_0)(x - x_1)}{(x_2 - x_0)(x_2 - x_1)}.$$

We can now note that

$$\frac{dP}{dx}(x_0) = \frac{x_0^p - x_1^p \frac{x_0 - x_2}{x_1 - x_2}}{x_0 - x_1} + \frac{x_0^p - x_2^p \frac{x_0 - x_1}{x_2 - x_1}}{x_0 - x_2} \qquad\qquad \frac{df}{dx}(x_0) = p x_0^{p-1},$$

so that

$$\frac{\partial}{\partial x_0}\left(\sum_{i=0}^{2} q_i x_i^p\right) = q_0 \left(\frac{df}{dx}(x_0) - \frac{dP}{dx}(x_0)\right).$$

Analogously, for $x_1$ and $x_2$ we obtain

$$\frac{\partial}{\partial x_1}\left(\sum_{i=0}^{2} q_i x_i^p\right) = q_1 \left( p x_1^{p-1} - \frac{x_1^p - x_0^p \frac{x_1 - x_2}{x_0 - x_2}}{x_1 - x_0} - \frac{x_1^p - x_2^p \frac{x_1 - x_0}{x_2 - x_0}}{x_1 - x_2}\right) = q_1 \left(\frac{df}{dx}(x_1) - \frac{dP}{dx}(x_1)\right),$$

$$\frac{\partial}{\partial x_2}\left(\sum_{i=0}^{2} q_i x_i^p\right) = q_2 \left( p x_2^{p-1} - \frac{x_2^p - x_0^p \frac{x_2 - x_1}{x_0 - x_1}}{x_2 - x_0} - \frac{x_2^p - x_1^p \frac{x_2 - x_0}{x_1 - x_0}}{x_2 - x_1}\right) = q_2 \left(\frac{df}{dx}(x_2) - \frac{dP}{dx}(x_2)\right).$$

We now prove that $f$ can intersect in at most 3 points with a degree-2 polynomial. If we let $P(x) = a_2 x^2 + a_1 x^1 + a_0$ for some $a_0, a_1, a_2$, we get that $\frac{d^2}{dx^2}(f - P) = p(p-1)x^{p-2} - 2a_2$. Since

$x^{p-2}$ is monotone and non-negative (and we know $a_2 \neq 0$ since the three points $P$ is interpolating through are not collinear) $\frac{d^2}{dx^2}(f - P)$ has at most one (single) root if $a_2$ and $p(p-1)$ are either both positive or both negative, and no root otherwise. We conclude that $f - P$ has at most 3 roots, and that $a_2 < 0$ when $p \in (0, 1)$ while $a_2 > 0$ when $p < 0$.

Since we know $x_0, x_1, x_2$ are 3 distinct roots of $f - P$, we know that all 3 roots are all single, and therefore that which one of $f$ and $P$ is greater 'switches' at each root. In particular, if $p \in (0, 1)$, we know that $a_2 < 0$, so $P(x) < f(x)$ for $x > x_0$ and $x \in (x_2, x_1)$, while $P(x) > f(x)$ for $x < x_2$ and $x \in (x_1, x_0)$. Therefore

$$\frac{df}{dx}(x_0) - \frac{dP}{dx}(x_0) > 0, \qquad \frac{df}{dx}(x_1) - \frac{dP}{dx}(x_1) < 0, \qquad \frac{df}{dx}(x_2) - \frac{dP}{dx}(x_2) > 0.$$

Similarly, for $p < 0$, we know that $a_2 > 0$, so $P(x) > f(x)$ for $x > x_0$ and $x \in (x_2, x_1)$, while $P(x) < f(x)$ for $x < x_2$ and $x \in (x_1, x_0)$. Therefore

$$\frac{df}{dx}(x_0) - \frac{dP}{dx}(x_0) < 0, \qquad \frac{df}{dx}(x_1) - \frac{dP}{dx}(x_1) > 0, \qquad \frac{df}{dx}(x_2) - \frac{dP}{dx}(x_2) < 0.$$

Since $q_0, q_1, q_2 \geq 0$, we get that

$$\frac{\partial}{\partial x_i} \sum_{i=0}^{2} q_i x_i^p \begin{cases} \geq 0 & \text{if } i = 0, 2 \\ \leq 0 & \text{if } i = 1 \end{cases} \text{ if } p \in (0, 1), \quad \frac{\partial}{\partial x_i} \sum_{i=0}^{2} q_i x_i^p \begin{cases} \leq 0 & \text{if } i = 0, 2 \\ \geq 0 & \text{if } i = 1 \end{cases} \text{ if } p < 0.$$

To finish the proof, we note that

$$\frac{\partial}{\partial x_0} \left( \sum_{i=0}^{2} q_i x_i^p \right)^{\frac{1}{p}} = \frac{1}{p} \left( \sum_{i=0}^{2} q_i x_i^p \right)^{\frac{1}{p} - 1} \frac{\partial}{\partial x_0} \left( \sum_{i=0}^{2} q_i x_i^p \right),$$

so if $p < 0$ the sign switches when considering the objective function instead of the objective function without the $p$-th root. This implies the lemma for $p \neq 0$.

We now turn to the case $p = 0$. Taking the logarithm of objective doesn't change the sign of its derivative, so it suffices to show that

$$\frac{\partial}{\partial x_i} \left( \sum_{i=0}^{2} q_i \ln(x_i) \right) \begin{cases} \geq 0 & \text{if } i \equiv m \pmod 2 \\ \leq 0 & \text{if } i \not\equiv m \pmod 2 \end{cases}.$$

Taking the derivative of the objective with respect to $x_0$, we obtain similarly to the case before that

$$\frac{\partial}{\partial x_0} \left( \sum_{i=0}^{2} q_i \ln(x_i) \right) = q_0 \frac{1}{x_0} + \ln(x_0) q_0 \frac{(x_1 - x_0) + (x_2 - x_0)}{(x_0 - x_1)(x_0 - x_2)}$$

$$+ q_1 \frac{\ln(x_1)}{(x_1 - x_0)} + \frac{\ln(x_1)(x_2 - \kappa_1)}{(x_1 - x_0)(x_1 - x_2)}$$

$$+ q_2 \frac{\ln(x_2)}{(x_2 - x_0)} + \frac{\ln(x_2)(x_1 - \kappa_1)}{(x_2 - x_0)(x_2 - x_1)}$$

$$= q_0 \left( \frac{1}{x_0} - \frac{\ln(x_0) - \ln(x_1) \frac{x_0 - x_2}{x_1 - x_2}}{x_0 - x_1} - \frac{\ln(x_0) - \ln(x_2) \frac{x_0 - x_1}{x_2 - x_1}}{x_0 - x_2} \right).$$

To determine the sign of the last line we consider the function $f : \mathbb{R}_{\geq 0} \to \mathbb{R}_{\geq 0}$ where $f(x) = \ln(x)$. We let $P$ be the unique degree-2 polynomial for which $P(x_0) = f(x_0), P(x_1) = f(x_1), P(x_2) = f(x_2)$ using the Lagrange basis polynomials:

$$P(x) = \ln(x_0) \frac{(x - x_1)(x - x_2)}{(x_0 - x_1)(x_0 - x_2)} + \ln(x_1) \frac{(x - x_0)(x - x_2)}{(x_1 - x_0)(x_1 - x_2)} + \ln(x_2) \frac{(x - x_0)(x - x_1)}{(x_2 - x_0)(x_2 - x_1)}.$$

We can now note that

$$\frac{dP}{dx}(x_0) = \frac{\ln(x_0) - \ln(x_1) \frac{x_0 - x_2}{x_1 - x_2}}{x_0 - x_1} - \frac{\ln(x_0) - \ln(x_2) \frac{x_0 - x_1}{x_2 - x_1}}{x_0 - x_2} \qquad \frac{df}{dx}(x_0) = \frac{1}{x_0},$$

so that

$$\frac{\partial}{\partial x_0}\left(\sum_{i=0}^{2} q_i x_i^p\right) = q_0\left(\frac{df}{dx}(x_0) - \frac{dP}{dx}(x_0)\right).$$

Analogously, for $x_1$ and $x_2$, we obtain

$$\frac{\partial}{\partial x_1}\left(\sum_{i=0}^{2} q_i x_i^p\right) = q_1\left(\frac{df}{dx}(x_1) - \frac{dP}{dx}(x_1)\right),$$

$$\frac{\partial}{\partial x_2}\left(\sum_{i=0}^{2} q_i x_i^p\right) = q_2\left(\frac{df}{dx}(x_2) - \frac{dP}{dx}(x_2)\right).$$

We now prove that $f$ can intersect in at most 3 points with a degree-2 polynomial. If we let $P(x) = a_2 x^2 + a_1 x^1 + a_0$ for some $a_0, a_1, a_2$, we get that $\frac{d^2}{dx^2}(f - P) = -1/x^2 - 2a_2$. Since $1/x^2$ is monotone and non-negative, $\frac{d^2}{dx^2}(f - P)$ has at most one (single) root if $a_2$ is non-positive and no root otherwise. We conclude that $f - P$ has at most 3 roots.

We know $x_0, x_1, x_2$ are 3 distinct roots of $f - P$, so we know that $a_2 < 0$ ($a_2$ cannot be zero since the three points $P$ is interpolating through are not collinear) and that all 3 roots are all single. Therefore, which one of $f$ and $P$ is greater 'switches' at each root. Since $a_2 < 0$, $P(x) < f(x)$ for $x > x_0$, so it follows that also $P(x) < f(x)$ for $x \in (x_2, x_1)$, while $P(x) > f(x)$ for $x < x_2$ and $x \in (x_1, x_0)$. Therefore

$$\frac{df}{dx}(x_0) - \frac{dP}{dx}(x_0) > 0, \qquad \frac{df}{dx}(x_1) - \frac{dP}{dx}(x_1) < 0, \qquad \frac{df}{dx}(x_2) - \frac{dP}{dx}(x_2) > 0;$$

the lemma for $p = 0$ follows. $\qquad\square$

We now apply Lemma 16 to solve for UMIN and UMAX. Note that in the lemma statement, the case $p = 0$ is the same as $p \neq 0$ when replacing the weighted $p$-mean by a weighted geometric mean.

**Lemma 18.** *Let $p < 1$ and $1 > \kappa_1 > \frac{\kappa_2 + u_{\min}}{1 + u_{\min}}$ and $\kappa_2 > \kappa_1^2 > u_{\min}^2$. Then, if $p \neq 0$,*

$$\mathrm{UMIN}(\kappa_1, \kappa_2; p, u_{\min}) =$$

$$\left(\frac{(\kappa_1 - u_{\min})^2}{\kappa_2 - 2u_{\min}\kappa_1 + u_{\min}^2}\left(\frac{\kappa_2 - u_{\min}\kappa_1}{\kappa_1 - u_{\min}}\right)^p + \frac{\kappa_2 - \kappa_1^2}{\kappa_2 - 2u_{\min}\kappa_1 + u_{\min}^2}u_{\min}^p\right)^{\frac{1}{p}},$$

*and*

$$\mathrm{UMAX}(\kappa_1, \kappa_2; p, u_{\min}) = \left(\frac{\kappa_2 - \kappa_1^2}{\kappa_2 - 2\kappa_1 + 1} + \frac{(1 - \kappa_1)^2}{\kappa_2 - 2\kappa_1 + 1}\left(\frac{\kappa_1 - \kappa_2}{1 - \kappa_1}\right)^p\right)^{\frac{1}{p}}.$$

*If $p = 0$,*

$$\mathrm{UMAX}(\kappa_1, \kappa_2; 0, u_{\min}) = \left(\frac{\kappa_1 - \kappa_2}{1 - \kappa_1}\right)^{\left(\frac{(1 - \kappa_1)^2}{\kappa_2 - 2\kappa_1 + 1}\right)}$$

*and*

$$\mathrm{UMIN}(\kappa_1, \kappa_2; 0, u_{\min}) = \left(\frac{\kappa_2 - u_{\min}\kappa_1}{\kappa_1 - u_{\min}}\right)^{\left(\frac{(\kappa_1 - u_{\min})^2}{\kappa_2 - 2u_{\min}\kappa_1 + u_{\min}^2}\right)} \cdot (u_{\min})^{\left(\frac{\kappa_2 - \kappa_1^2}{\kappa_2 - 2u_{\min}\kappa_1 + u_{\min}^2}\right)}$$

*Proof.* W.l.o.g. we assume $x_0 > x_1 > x_2$. Since the weighted mean of $x_0, x_1, x_2$ is $\kappa_1$, we can furthermore assume w.l.o.g. that $x_0 > \kappa_1 > x_2$ (If $x_0 > \kappa_1$, then also $x_2 < \kappa_1$; if $x_0 = \kappa_1$, then $q_1 = q_2 = 0$, which is still captured by $x_0 > x_1 = \kappa_1 > x_2$ and $q_0 = q_2 = 0$; $x_0 < \kappa_1$ is not possible).

We now use this to rewrite the constraints $q_i \geq 0$ in terms of $x_0, x_1, x_2$. For $q_0$, we obtain

$$q_0 \geq 0 \Leftrightarrow \frac{1}{(x_0 - x_1)(x_0 - x_2)}(\kappa_2 - \kappa_1(x_1 + x_2) + x_1 x_2) \geq 0$$

$$\Leftrightarrow \kappa_2 - \kappa_1 x_2 + x_1(x_2 - \kappa_1) \geq 0$$

$$\Leftrightarrow x_1 \leq \frac{\kappa_2 - x_2 \kappa_1}{\kappa_1 - x_2}$$

$$\Leftrightarrow x_1 \leq \kappa_1 + \frac{\kappa_2 - \kappa_1^2}{\kappa_1 - x_2}. \tag{5}$$

Similarly, for $q_1$ and $q_2$, we obtain

$$q_1 \geq 0 \Leftrightarrow x_0 \geq \kappa_1 + \frac{\kappa_2 - \kappa_1^2}{\kappa_1 - x_2} \tag{6}$$

$$\Leftrightarrow x_2 \leq \kappa_1 - \frac{\kappa_2 - \kappa_1^2}{x_0 - \kappa_1}, \tag{7}$$

$$q_2 \geq 0 \Leftrightarrow x_1 \geq \kappa_1 - \frac{\kappa_2 - \kappa_1^2}{x_0 - \kappa_1}. \tag{8}$$

We now solve for UMIN and UMAX by finding the minimum and maximum of their defining optimization program from Lemma 14.

*Finding the minimum:* We'll first assume $p \neq 0$. By Lemma 16, we know that increasing $x_1$ will (non-strictly) decrease the objective. The upper bounds on $x_1$ are Constraint (5) and $x_1 < x_0 \leq 1$. Thus, we know that there is a minimum either for $x_1 = \kappa_1 + \frac{\kappa_2 - \kappa_1^2}{\kappa_1 - x_2}$ (tight to the upper bound given by Constraint (5)) or for $x_0 \leq \kappa_1 + \frac{\kappa_2 - \kappa_1^2}{\kappa_1 - x_2}$ (the only case in which $x_1$ can't take this value since $x_1 < x_0$). By Constraint (6), we know $x_0 \geq \kappa_1 + \frac{\kappa_2 - \kappa_1^2}{\kappa_1 - x_2}$, so the second case becomes $x_0 = \kappa_1 + \frac{\kappa_2 - \kappa_1^2}{\kappa_1 - x_2}$.

We first consider the case $x_1 = \kappa_1 + \frac{\kappa_2 - \kappa_1^2}{\kappa_1 - x_2} = \frac{\kappa_2 - x_2 \kappa_1}{\kappa_1 - x_2}$. We get that

$$q_0 = \frac{\kappa_2 - \kappa_1\left(\frac{\kappa_2 - x_2 \kappa_1}{\kappa_1 - x_2} + x_2\right) + \frac{\kappa_2 - x_2 \kappa_1}{\kappa_1 - x_2} x_2}{(x_0 - x_1)(x_0 - x_2)}$$

$$= \frac{\kappa_2 - \left(\frac{\kappa_1 \kappa_2 - x_2 \kappa_1^2 + x_2 \kappa_1^2 - x_2^2 \kappa_1 - \kappa_2 x_2 + x_2^2 \kappa_1}{\kappa_1 - x_2}\right)}{(x_0 - x_1)(x_0 - x_2)}$$

$$= \frac{\kappa_2 - \left(\frac{\kappa_2(\kappa_1 - x_2)}{\kappa_1 - x_2}\right)}{(x_0 - x_1)(x_0 - x_2)} = 0,$$

so the value of $x_0$ won't affect the objective. Furthermore,

$$q_1 = \frac{\kappa_2 - \kappa_1(x_0 + x_2) + x_0 x_2}{\left(\frac{\kappa_2 - x_2 \kappa_1}{\kappa_1 - x_2} - x_0\right)\left(\frac{\kappa_2 - x_2 \kappa_1}{\kappa_1 - x_2} - x_2\right)}$$

$$= \frac{(\kappa_2 - \kappa_1(x_0 + x_2) + x_0 x_2)(\kappa_1 - x_2)^2}{(\kappa_2 - x_2 \kappa_1 - x_0 \kappa_1 + x_0 x_2)(\kappa_2 - x_2 \kappa_1 - x_2 \kappa_1 + x_2^2)}$$

$$= \frac{(\kappa_1 - x_2)^2}{\kappa_2 - 2x_2 \kappa_1 + x_2^2},$$

$$q_2 = 1 - q_0 - q_1 = \frac{\kappa_2 - \kappa_1^2}{\kappa_2 - 2x_2 \kappa_1 + x_2^2},$$

so the objective becomes

$$\left(\frac{(\kappa_1 - x_2)^2}{\kappa_2 - 2x_2 \kappa_1 + x_2^2}\left(\frac{\kappa_2 - x_2 \kappa_1}{\kappa_1 - x_2}\right)^p + \frac{\kappa_2 - \kappa_1^2}{\kappa_2 - 2x_2 \kappa_1 + x_2^2} x_2^p\right)^{1/p} \quad \text{if } p \neq 0 \tag{9}$$

$$\left(\frac{\kappa_2 - x_2 \kappa_1}{\kappa_1 - x_2}\right)^{\left(\frac{(\kappa_1 - x_2)^2}{\kappa_2 - 2x_2 \kappa_1 + x_2^2}\right)} \cdot (x_2)^{\left(\frac{\kappa_2 - \kappa_1^2}{\kappa_2 - 2x_2 \kappa_1 + x_2^2}\right)} \quad \text{if } p = 0. \tag{10}$$

In both cases, the objective is a univariate function in $x_2$. We will show that it is monotonically increasing on the interval $[0, \kappa_1)$, which implies that the minimum is achieved at $x_2 = u_{\min}$, which implies $x_1 = \frac{\kappa_2 - u_{\min}\kappa_1}{\kappa_1 - u_{\min}}$. Since $\kappa_1(1 + u_{\min}) > \kappa_2 + u_{\min}$, by assumption in the lemma statement, if follows that $x_1 < 1$. We know that $x_0$ doesn't affect the objective, so we can just set $x_0$ to be any value in $(x_1, 1]$. Plugging this in, we get

$$
\begin{aligned}
&x_0 \in (x_1, 1] &&q_0 = 0 \\
&x_1 = \frac{\kappa_2 - u_{\min}\kappa_1}{\kappa_1 - u_{\min}} &&q_1 = \frac{(\kappa_1 - u_{\min})^2}{\kappa_2 - 2u_{\min}\kappa_1 + u_{\min}{}^2} \\
&x_2 = u_{\min} &&q_2 = \frac{\kappa_2 - \kappa_1^2}{\kappa_2 - 2u_{\min}\kappa_1 + u_{\min}{}^2}.
\end{aligned}
\tag{11}
$$

We'll now show for the case $p \neq 0$ that Equation (9) is monotonically increasing on $[0, \kappa_1)$. To do that, we first drop the $p$-th root and take the derivative of the expression without the root (with respect to $x_2$), to calculate its sign. We get

$$
\frac{(\kappa_2 - \kappa_1^2)\Big(x_1^{p-1}((p-2)\kappa_2 - 2(p-1)\kappa_1 x_2 + p x_2^2) + x_2^{p-1}(p\kappa_2 - 2(p-1)\kappa_1 x_2 + (p-2)x_2^2)\Big)}{(\kappa_2 - 2\kappa_1 x_2 + x_2^2)^2},
$$

where we write $x_1$ for $\frac{\kappa_2 - x_2\kappa_1}{\kappa_1 - x_2}$ for brevity. Since $\kappa_2 - \kappa_1^2 > 0$ and $\kappa_2 - 2\kappa_1 x_2 + x_2^2 = \kappa_2 - \kappa_1^2 + (\kappa_1 - x_2)^2 > 0$, we can drop those terms without changing the sign of the expression. We simplify further to get

$$
\begin{aligned}
&x_1^{p-1}((p-2)\kappa_2 - 2(p-1)\kappa_1 x_2 + p x_2^2) + x_2^{p-1}(p\kappa_2 - 2(p-1)\kappa_1 x_2 + (p-2)x_2^2) \\
&= (x_1^{p-1} + x_2^{p-1})p(\kappa_2 - 2\kappa_1 x_2 + x_2^2) - 2x_1^{p-1}(\kappa_2 - \kappa_1 x_2) + 2x_2^{p-1}(\kappa_1 x_2 - x_2^2) \\
&= (x_1^{p-1} + x_2^{p-1})p(x_1 - x_2)(\kappa_1 - x_2) - 2x_1^p(\kappa_1 - x_2) + 2x_2^p(\kappa_1 - x_2).
\end{aligned}
$$

Since $\kappa_1 > x_2$ and $x_1 > x_2$, we can divide by $2(\kappa_1 - x_2)(x_1 - x_2)$ without changing the sign, to get:

$$
\frac{p x_1^{p-1} + p x_2^{p-1}}{2} - \frac{x_1^p - x_2^p}{x_1 - x_2}.
$$

If we let $f(x) = p x^{p-1}$, it follows immediately from the Hermite–Hadamard inequality that the above expression is non-negative for $p \in (0, 1)$ and non-positive for $p < 0$.

By the chain rule, taking the $p$-th root switches the sign of the derivative if $p < 0$ and does not if $p \in (0, 1)$. Thus, we can conclude that the derivative of Equation (9) on $[0, \kappa_1)$ is non-negative for all $p < 0$ and $p \in (0, 1)$.

Let's next show for the case $p = 0$ that Equation (10) is also monotonically increasing on $[0, \kappa_1)$. The proof is very similar to the proof right above for Equation (9). First, we note that taking the logarithm of Equation (10) doesn't change the sign of its derivative, so we just need to show that the derivative of the logarithm is non-negative. This derivative is:

$$
\frac{(\kappa_2 - \kappa_1^2)\Big(2(\kappa_1 - x_2)(\ln(x_2) - \ln(x_1)) + \frac{(\kappa_2 - x_2^2)(\kappa_2 - 2x_2\kappa_1 + x_2^2)}{x(\kappa_2 - \kappa_1 x_2)}\Big)}{(\kappa_2 - 2\kappa_1 x_2 + x_2^2)^2},
$$

where again we write $x_1$ for $\frac{\kappa_2 - x_2\kappa_1}{\kappa_1 - x_2}$ for brevity. Since $\kappa_2 - \kappa_1^2 > 0$ and $\kappa_2 - 2\kappa_1 x_2 + x_2^2 = \kappa_2 - \kappa_1^2 + (\kappa_1 - x_2)^2 > 0$, we can drop those terms without changing the sign of the expression. We simplify further to get

$$
\begin{aligned}
&2(\kappa_1 - x_2)(\ln(x_2) - \ln(x_1)) + \frac{(\kappa_2 - x_2^2)(\kappa_2 - 2x_2\kappa_1 + x_2^2)}{x(\kappa_2 - \kappa_1 x_2)} \\
&= 2(\kappa_1 - x_2)(\ln(x_2) - \ln(x_1)) + \left(\frac{1}{x_2} + \frac{\kappa_1 - x_2}{\kappa_2 - x_2\kappa_1}\right)(\kappa_2 - 2x_2\kappa_1 + x_2^2) \\
&= 2(\kappa_1 - x_2)(\ln(x_2) - \ln(x_1)) + \left(\frac{1}{x_2} + \frac{1}{x_1}\right)(x_1 - x_2)(\kappa_1 - x_2).
\end{aligned}
$$

Since $\kappa_1 > x_2$ and $x_1 > x_2$, we can divide by $2(\kappa_1 - x_2)(x_1 - x_2)$ without changing the sign to get

$$\frac{\frac{1}{x_1} + \frac{1}{x_2}}{2} - \frac{\ln(x_1) - \ln(x_2)}{x_1 - x_2}.$$

If we let $f(x) = 1/x$, it follows immediately from the Hermite–Hadamard inequality that the above expression is non-negative. Thus, we can conclude that Equation (10) on $[0, \kappa_1)$ is non-negative.

It remains to consider the case $x_0 = \kappa_1 + \frac{\kappa_2 - \kappa_1^2}{\kappa_1 - x_2} = \frac{\kappa_2 - x_2 \kappa_1}{\kappa_1 - x_2}$. We get analogous equations for $q_0, q_1, q_2$ as in the earlier case (with $q_0$ and $q_1$ switched, naturally), so we get the same objective as in Equations (9) and (10), which we know is minimized for $x_2 = u_{\min}$. Analogously to above, this implies $x_0 = \frac{\kappa_2 - u_{\min} \kappa_1}{\kappa_1 - u_{\min}}$. We know that $x_0 > u_{\min}$ since $\kappa_2 - u_{\min} \kappa_1 - u_{\min}(\kappa_1 - u_{\min}) = \kappa_2 - \kappa_1^2 + (\kappa_1 - u_{\min})^2 > 0$, so since $x_1$ doesn't affect the objective, we can just set it to be any value in $(x_2, x_0)$. We get

$$
\begin{aligned}
x_0 &= \frac{\kappa_2 - u_{\min}\kappa_1}{\kappa_1 - u_{\min}} & q_0 &= \frac{(\kappa_1 - u_{\min})^2}{\kappa_2 - 2u_{\min}\kappa_1 + u_{\min}^2} \\
x_1 &\in (x_2, x_0) & q_1 &= 0 \\
x_2 &= u_{\min} & q_2 &= \frac{\kappa_2 - \kappa_1^2}{\kappa_2 - 2u_{\min}\kappa_1 + u_{\min}^2}.
\end{aligned}
\tag{12}
$$

We know that the objective is minimized for either the variable assignments from Solution (11) or from Solution (12). We note that in fact both assignments give the same objective value, thus, the global minimum. This value is exactly $\text{UMIN}(\kappa_1, \kappa_2; p, u_{\min})$ in the lemma statement.

*Finding the maximum:* By Lemma 16, we know that decreasing $x_1$ will (non-strictly) increase the objective. The lower bounds on $x_1$ that we have are Constraint (8) and $x_1 > x_2 \geq u_{\min}$. Thus, we know that there is a maximum either for $x_1 = \kappa_1 - \frac{\kappa_2 - \kappa_1^2}{x_0 - \kappa_1}$ (tight to the lower bound given by Constraint (8)) or for $x_2 \geq \kappa_1 - \frac{\kappa_2 - \kappa_1^2}{x_0 - \kappa_1}$ (the only case in which $x_1$ can't take this value since $x_1 > x_2$). By Constraint (7), we know that $x_2 \leq \kappa_1 - \frac{\kappa_2 - \kappa_1^2}{x_0 - \kappa_1}$, so the second case becomes $x_2 = \kappa_1 - \frac{\kappa_2 - \kappa_1^2}{x_0 - \kappa_1}$.

We first consider the case $x_1 = \kappa_1 - \frac{\kappa_2 - \kappa_1^2}{x_0 - \kappa_1} = \frac{\kappa_1 x_0 - \kappa_2}{x_0 - \kappa_1}$. We get that

$$
\begin{aligned}
q_2 &= \frac{\kappa_2 - \kappa_1\left(x_0 + \frac{\kappa_1 x_0 - \kappa_2}{x_0 - \kappa_1}\right) + x_0 \frac{\kappa_1 x_0 - \kappa_2}{x_0 - \kappa_1}}{(x_2 - x_0)(x_2 - x_1)} \\
&= \frac{\kappa_2 - \left(\frac{\kappa_1 x_0^2 - \kappa_1^2 x_0 + \kappa_1^2 x_0 - \kappa_1 \kappa_2 - \kappa_1 x_0^2 + \kappa_2 x_0}{x_0 - \kappa_1}\right)}{(x_2 - x_0)(x_2 - x_1)} \\
&= \frac{\kappa_2 - \left(\frac{\kappa_2(x_0 - \kappa_1)}{x_0 - \kappa_1}\right)}{(x_2 - x_0)(x_2 - x_1)} = 0,
\end{aligned}
$$

so the value of $x_2$ won't affect the objective. Furthermore,

$$
\begin{aligned}
q_1 &= \frac{\kappa_2 - \kappa_1(x_0 + x_2) + x_0 x_2}{\left(\frac{\kappa_1 x_0 - \kappa_2}{x_0 - \kappa_1} - x_0\right)\left(\frac{\kappa_1 x_0 - \kappa_2}{x_0 - \kappa_1} - x_2\right)} \\
&= \frac{(\kappa_2 - \kappa_1(x_0 + x_2) + x_0 x_2)(x_0 - \kappa_1)^2}{(\kappa_1 x_0 - \kappa_2 - x_0^2 + \kappa_1 x_0)(\kappa_1 x_0 - \kappa_2 - x_0 x_2 + \kappa_1 x_2)} \\
&= \frac{(x_0 - \kappa_1)^2}{\kappa_2 - 2\kappa_1 x_0 + x_0^2}, \\
q_0 &= 1 - q_1 - q_2 = \frac{\kappa_2 - \kappa_1^2}{\kappa_2 - 2\kappa_1 x_0 + x_0^2},
\end{aligned}
$$

so the objective becomes

$$\left( \frac{\kappa_2 - \kappa_1^2}{\kappa_2 - 2\kappa_1 x_0 + x_0^2} x_0^p + \frac{(x_0 - \kappa_1)^2}{\kappa_2 - 2\kappa_1 x_0 + x_0^2} \left( \frac{\kappa_1 x_0 - \kappa_2}{x_0 - \kappa_1} \right)^p \right)^{1/p} \qquad \text{if } p \neq 0 \qquad (13)$$

$$(x_0)^{\left( \frac{\kappa_2 - \kappa_1^2}{\kappa_2 - 2\kappa_1 x_0 + x_0^2} \right)} \cdot \left( \frac{\kappa_1 x_0 - \kappa_2}{x_0 - \kappa_1} \right)^{\left( \frac{(x_0 - \kappa_1)^2}{\kappa_2 - 2\kappa_1 x_0 + x_0^2} \right)} \qquad \text{if } p = 0 \qquad (14)$$

In both cases, this is a univariate function in $x_0$. We know that $\kappa_1 x_0 = \sum_{i=0}^{2} q_i x_i x_0 \geq \sum_{i=0}^{2} q_i x_i^2 = \kappa_2$, so we have $x_0 \in [\frac{\kappa_2}{\kappa_1}, 1]$. In fact, note that if $p \leq 0$, the inequality is strict since $x_2 > 0$ and at least 2 of the $q_i$ are positive (else $\kappa_1^2 = \kappa_2$), so we have $x_0 \in (\frac{\kappa_2}{\kappa_1}, 1]$ and Equations (13) and (14) are well-defined for all valid $x_0$ and $p$. We'll show that Equations (13) and (14) are monotonically increasing on the interval $(\frac{\kappa_2}{\kappa_1}, 1]$ ($[\frac{\kappa_2}{\kappa_1}, 1]$ for $p \in (0, 1)$). This implies that the maximum is achieved at $x_0 = 1$, which gives $x_1 = \frac{\kappa_1 - \kappa_2}{1 - \kappa_1}$. Since $\kappa_1(1 + u_{\min}) > \kappa_2 + u_{\min}$, by assumption in the lemma statement, it follows that $x_1 > u_{\min}$. We know that $x_2$ doesn't affect the objective, so we can just set $x_2$ to be any value in $[u_{\min}, x_1)$. Plugging this in, we get

$$
\begin{aligned}
x_0 &= 1 & q_0 &= \frac{\kappa_2 - \kappa_1^2}{\kappa_2 - 2\kappa_1 + 1} \\
x_1 &= \frac{\kappa_1 - \kappa_2}{1 - \kappa_1} & q_1 &= \frac{(1 - \kappa_1)^2}{\kappa_2 - 2\kappa_1 + 1} \\
x_2 &\in [u_{\min}, x_1] & q_2 &= 0.
\end{aligned}
\qquad (15)
$$

We now show that for $p \neq 0$, Equation (13) is monotonically increasing on $(\frac{\kappa_2}{\kappa_1}, 1]$ ($[\frac{\kappa_2}{\kappa_1}, 1]$ for $p \in (0, 1)$). This proof is almost analogous to the minimization part. First, we note that Equation (13) is identical to Equation (9) with $x_2$ replaced by $x_0$. We can do the same algebraic steps, now noting that since $\kappa_1 < \kappa_2/\kappa_1 \leq x_0$ and $x_1 < x_0$, diving by $2(\kappa_1 - x_0)(x_1 - x_0)$ doesn't change the sign. We thus get, analogously to the minimization part, that the sign of the derivative of Equation (13) without the outer $p$-th root is the same as the sign of

$$\frac{p x_0^{p-1} + p x_1^{p-1}}{2} - \frac{x_0^p - x_1^p}{x_0 - x_1}.$$

Again, by the Hermite–Hadamard inequality with $f(x) = p x^{p-1}$, it follows immediately that the above expression is non-negative for $p \in (0, 1)$ and non-positive for $p < 0$. By the chain rule, taking the $p$-th root switches the sign of the derivative if $p < 0$ and does not if $p \in (0, 1)$. Thus, we can conclude that the derivative of Equation (13) on $(\frac{\kappa_2}{\kappa_1}, 1]$ ($[\frac{\kappa_2}{\kappa_1}, 1]$ for $p \in (0, 1)$) is non-negative for all $p < 0$ and $p \in (0, 1)$.

Let's next show for the case $p = 0$ that Equation (14) is monotonically increasing on $(\frac{\kappa_2}{\kappa_1}, 1]$. Also this proof is almost analogous to the minimization part. We note that Equation (14) is identical to Equation (10) with $x_2$ replaced by $x_0$. We can do the same algebraic steps, now noting that since $\kappa_1 < \kappa_2/\kappa_1 \leq x_0$ and $x_1 < x_0$, diving by $2(\kappa_1 - x_0)(x_1 - x_0)$ doesn't change the sign. We thus get, analogously to the minimization part, that the sign of the derivative of Equation (13) without the outer $p$-th root is the same as the sign of

$$\frac{\frac{1}{x_0} + \frac{1}{x_1}}{2} - \frac{\ln(x_0) - \ln(x_1)}{x_0 - x_1}.$$

If we let $f(x) = 1/x$, it follows immediately from the Hermite–Hadamard inequality that the above expression is non-negative. Thus, we can conclude that Equation (14) on $(\frac{\kappa_2}{\kappa_1}, 1]$ is non-negative.

It remains to consider the case $x_2 = \kappa_1 - \frac{\kappa_2 - \kappa_1^2}{x_0 - \kappa_1} = \frac{\kappa_1 x_0 - \kappa_2}{x_0 - \kappa_1}$. We get analogous equations for $q_0, q_1, q_2$ as in the earlier case (with $q_1$ and $q_2$ switched, naturally), so we get the same objective as in Equations (9) and (10), which we know is maximized for $x_0 = 1$. Analogously to above, this implies $x_2 = \frac{\kappa_1 - \kappa_2}{1 - \kappa_1}$. We know that $x_2 < 1$ since $(1 - \kappa_1) - (\kappa_1 - \kappa_2) = \kappa_2 - \kappa_1^2 + (1 - \kappa_1)^2 > 0$,

so since $x_1$ doesn't affect the objective, we can just set it to be any value in $(x_2, x_0)$. We get

$$
\begin{aligned}
x_0 &= 1 & q_0 &= \frac{\kappa_2 - \kappa_1^2}{\kappa_2 - 2\kappa_1 + 1} \\
x_1 &\in (x_2, x_0) & q_1 &= 0 \\
x_2 &= \frac{\kappa_1 - \kappa_2}{1 - \kappa_1} & q_2 &= \frac{(1 - \kappa_1)^2}{\kappa_2 - 2\kappa_1 + 1}.
\end{aligned}
\tag{16}
$$

We know that the objective is maximized for either the variable assignments from Solution (15) or from Solution (16). We note that in fact both assignments give the same objective value, thus, the global maximum. This value is exactly $\text{UMAX}(\kappa_1, \kappa_2; p, u_{\min})$ in the lemma statement. $\qquad\square$

We now have the tools to prove the main theorem.

*Proof of Theorem 9, upper bound.* We first consider the case $p \leq 0$ and $u_{\min} = 0$. Since $u_{\mathcal{C},p}(a), u_{\mathcal{P},p}(a) \in [0,1]$ for all $a \in \mathcal{A}$, we know that $|u_{\mathcal{C},p}(a) - u_{\mathcal{P},p}(a)| \leq 1$ for all $a \in \mathcal{A}$. Thus, $\mathcal{C}$ is 1-accurate.

Let's now consider the remaining cases $p < 1$ and $u_{\min} \in (0, 1)$ or $p \in (0, 1)$ and $u_{\min} = 0$. By Lemma 15, we know that any 2-representative panel $\mathcal{C}$ is $\varepsilon$-accurate with

$$
\begin{aligned}
\varepsilon = \max \quad & \text{UMAX}(\kappa_1, \kappa_2; p, u_{\min}) - \text{UMIN}(\kappa_1, \kappa_2; p, u_{\min}) \\
\text{s.t.} \quad & [\kappa_{i+j}]_{i,j=0}^{1}, [(1 + u_{\min})\kappa_{i+j+1} - \kappa_{i+j+2} - u_{\min}\kappa_{i+j}]_{i,j=0}^{0} \succeq 0
\end{aligned}
$$

where $[M_{i,j}]_{i,j=0}^{d-1}$ is a $d \times d$ matrix with entry $M_{i,j}$ at position $(i, j)$ and $M \succeq 0$ means that $M$ is positive-semidefinite. The first matrix $[\kappa_{i+j}]_{i,j=0}^{1}$ thus is $2 \times 2$ and hence positive semi-definite if and only if its trace and determinant are non-negative, i.e. $\kappa_0 + \kappa_2 = 1 + \kappa_2 \geq 0$ and $\det([\kappa_{i+j}]_{i,j=0}^{1}) = \kappa_2 - \kappa_1^2 \geq 0$. The second matrix is $1 \times 1$ and thus positive semi-definite if and only if $(1 + u_{\min})\kappa_1 - \kappa_2 - u_{\min} \geq 0$. These conditions are equivalent to $\kappa_2 \geq \kappa_1^2$ and $\kappa_1 \geq \frac{\kappa_2 + u_{\min}}{1 + u_{\min}}$.

We can combine the two inequalities to get $\kappa_1 \geq \frac{\kappa_1^2 + u_{\min}}{1 + u_{\min}}$ which implies $0 \geq (\kappa_1 - 1)(\kappa_1 - u_{\min})$, or equivalently $\kappa_1 \in [u_{\min}, 1]$. This gives us the constraints $\kappa_2 \geq \kappa_1^2 \geq u_{\min}^2$ and $1 \geq \kappa_1 \geq \frac{\kappa_2 + u_{\min}}{1 + u_{\min}}$ in the theorem statement, except with weak instead of strict inequalities.

Thus, we'll next show that if any of the 4 inequalities is not strict, then $\text{UMAX}(\kappa_1, \kappa_2; p, u_{\min}) - \text{UMIN}(\kappa_1, \kappa_2; p, u_{\min}) = 0$ (see Lemma 14 for the definition of UMAX and UMIN).

First, if $\kappa_1^2 = \kappa_2$, we get that $\sqrt{\sum_{i=0}^{2} q_i x_i^2} = \sum_{i=0}^{2} q_i x_i$ which by the arithmetic-quadratic inequality only holds if $x_0 = x_1 = x_2$ (unless $q_i = 0$ for some $i$, in which case the corresponding $x_i$ can take any value). Thus, $\text{UMAX}(\kappa_1, \kappa_2; p, u_{\min}) = \text{UMIN}(\kappa_1, \kappa_2; p, u_{\min}) = \kappa_1$.

Next, if $\kappa_1 = \frac{\kappa_2 + u_{\min}}{1 + u_{\min}}$, we get that

$$
\begin{aligned}
0 &= \kappa_1(1 + u_{\min}) - \kappa_2 - u_{\min} \\
&= \sum_{i=0}^{2} q_i(x_i + x_i u_{\min} - x_i^2) - u_{\min} \\
&= \sum_{i=0}^{2} q_i(x_i + x_i u_{\min} - x_i^2 - u_{\min}) + \sum_{i=0}^{2} q_i u_{\min} - u_{\min} \\
&= \sum_{i=0}^{2} q_i(1 - x_i)(x_i - u_{\min}).
\end{aligned}
$$

Since $q_i(1 - x_i)(x_i - u_{\min}) \geq 0$ for $x_i \in [u_{\min}, 1]$, it follows that $x_i = u_{\min}$ or $x_i = 1$ for all $i = 0, 1, 2$. To obtain first moment $\kappa_1$, the weights $(q_i)$ need to be $\frac{1 - \kappa_1}{1 - u_{\min}}$ for $u_{\min}$ and $\frac{\kappa_1 - u_{\min}}{1 - u_{\min}}$ for 1, which also gives second moment $\kappa_2 = \kappa_1(1 + u_{\min}) - u_{\min}$ as desired. It follows that $\text{UMAX}(\kappa_1, \kappa_2; p, u_{\min}) = \text{UMIN}(\kappa_1, \kappa_2; p, u_{\min}) = \left( \frac{1 - \kappa_1}{1 - u_{\min}} u_{\min}^p + \frac{\kappa_1 - u_{\min}}{1 - u_{\min}} \right)^{1/p}$.

Lastly, both $\kappa_1 = 1$ and $\kappa_1 = u_{\min}$ imply $\kappa_1^2 = \kappa_2$, so the first case applies.

Since there always exists a point at which all constraints are strictly followed, and $\mathrm{UMAX}(\kappa_1, \kappa_2; p, u_{\min}) - \mathrm{UMIN}(\kappa_1, \kappa_2; p, u_{\min}) \geq 0$ at any such point, as follows immediately from their definition, we can restrict the constraints to being strict without changing the value of the optimization program. This allows us to plug in the equations for UMAX and UMIN from Lemma 18 to obtain the upper bounds from the theorem statement. $\qquad\square$

### A.4 Lower-bounding the $\varepsilon$-accuracy of $m$-representative panels

The proofs of Theorems 8 and 9 consist of two parts: showing that every 1- and 2-representative panel is $\varepsilon$-accurate for the $\varepsilon$ in the statement of the respective theorem, and then showing that this $\varepsilon$ cannot be made smaller. We proved the first part in Appendix A.3. Here, we show the second part: For any $\varepsilon' < \varepsilon$ (and $p < 1$, $u_{\min} \in [0,1)$), for $\varepsilon$ as specified in the respective theorem statements, there exists a 1- or 2-representative panel that is not $\varepsilon'$-accurate.

The two lower bound proofs are similar: We first show a helper lemma (Lemmas 19 and 20) that, intuitively, says that if the worst-case utilities giving UMAX and UMIN in the upper bound are integers divided by $|\mathcal{F}|$, there are two panels in which each feature-value (or intersection of two feature-values) appears in the same share of individuals and the two panels have the highest and lowest welfare possible for panels with first moment $\kappa_1$ (and second moment $\kappa_2$) and individual utilities in $[u_{\min}, 1]$, as specified by UMAX and UMIN in Lemmas 17 and 18.

#### A.4.1 1-representative panels

**Lemma 19.** *Let $p < 1$ and $u_{\min} \in [0,1)$, or $p \in (0,1)$ and $u_{\min} = 0$. Let $\ell$ be an integer, and $\kappa_1, u_{\min} \in \mathbb{Q}$ such that $1 > \kappa_1 > u_{\min}$ and $\ell u_{\min}, \ell \kappa_1 \in \mathbb{Z}$. Then, there exist two panels $\mathcal{C} = \{(r_1, y_1), \ldots, (r_T, y_T)\}$, $\mathcal{C}' = \{(r'_1, y_1), \ldots, (r'_T, y_T)\}$ with individuals of types in $\mathcal{I} = \{y_1, \ldots, y_T\}$ with $|\mathcal{F}| = \ell$ features and an alternative $a$ s.t. $u_i(a) \in [u_{\min}, 1]$ for all $y_i \in \mathcal{I}$, so that each feature value appears in the same share in $\mathcal{C}$ and $\mathcal{C}'$, i.e. $\sum_{y_i \in \mathcal{I}} r_i \mathbf{y_i} = \sum_{y_i \in \mathcal{I}} r'_i \mathbf{y_i}$ and*

$$
u_{\mathcal{C},p}(a) = \mathrm{UMIN}(\kappa_1; p, u_{\min}) = \begin{cases} \left( \frac{\kappa_1 - u_{\min}}{1 - u_{\min}} + \frac{1 - \kappa_1}{1 - u_{\min}} u_{\min}^p \right)^{1/p} & \text{if } p \neq 0 \\ (u_{\min})^{\frac{1 - \kappa_1}{1 - u_{\min}}} & \text{if } p = 0 \end{cases},
$$

*and $u_{\mathcal{C}',p}(a) = \mathrm{UMAX}(\kappa_1; p, u_{\min}) = \kappa_1$.*

*Proof.* Let $\mathcal{F} = \{f_1, ..., f_\ell\}$ and $V_f = \{0,1\}$ for all $f \in \mathcal{F}$. We let $y_1$ be the type that has value 1 for all features, i.e. $y_1(f,i) = i$ for all $f \in \mathcal{F}, i = 0,1$. We let

$$
Y_{\kappa_1} = \{y : |\{f \in \mathcal{F} : y(f,1) = 1\}| = \ell \kappa_1\} \quad Y_{u_{\min}} = \{y : |\{f \in \mathcal{F} : y(f,1) = 1\}| = \ell u_{\min}\}
$$

be the set of all possible types that have value 1 for exactly $\ell \kappa_1$ and $\ell u_{\min}$ of their features, respectively. We let $\mathcal{I} = \{y_1\} \cup Y_{\kappa_1} \cup Y_{u_{\min}}$.

Let's now define the panels. We let $\mathcal{C}$ be the panel consisting of $(\ell \kappa_1 - \ell u_{\min})\binom{\ell}{\ell u_{\min}} \in \mathbb{Z}$ individuals of type $y_1$ and of $(\ell - \ell \kappa_1) \in \mathbb{Z}$ individuals of each of the $\binom{\ell}{\ell u_{\min}}$ types in $Y_{u_{\min}}$. We let $\mathcal{C}'$ be the panel consisting of 1 individual for each type in $Y_{\kappa_1}$.

We need to confirm that for every feature $f \in \mathcal{F}$, each value $v \in V_f$ appears in the same share of $\mathcal{C}$ and $\mathcal{C}'$. Since there are only two values, 0 and 1, for each feature, it suffices to show that value 1 appears in the same share of individuals in $\mathcal{C}$ and $\mathcal{C}'$ for any feature.

To keep the math concise moving forward, we'll define binomial coefficients with a negative bottom entry as zero. This precisely handles the cases $u_{\min} < 1/\ell$, i.e. $u_{\min} = 0$, and $\kappa_1 < 1/\ell$, i.e. $\kappa_1 = 0$, below.

Consider any feature $f \in \mathcal{F}$. First, we can see that a fraction of $\binom{\ell-1}{\ell\kappa_1-1}/\binom{\ell}{\ell\kappa_1} = \kappa_1$ of the individuals in $\mathcal{C}'$ have value 1 for feature $f$. Also for $\mathcal{C}$ we get that a fraction of

$$
\frac{(\ell\kappa_1 - \ell u_{\min})\binom{\ell}{\ell u_{\min}} + (\ell - \ell\kappa_1)\binom{\ell-1}{\ell u_{\min}-1}}{(\ell\kappa_1 - \ell u_{\min})\binom{\ell}{\ell u_{\min}} + (\ell - \ell\kappa_1)\binom{\ell}{\ell u_{\min}}} = \frac{(\kappa_1 - u_{\min}) + (1 - \kappa_1)u_{\min}}{(\kappa_1 - u_{\min}) + (1 - \kappa_1)} = \kappa_1
$$

of the individuals in $\mathcal{C}'$ have value 1 for feature $f$.

We let $a$ be an alternative such that $a(f, i) = i/\ell$ for all $f \in \mathcal{F}$ and $i = 0, 1$; i.e., an individual's utility is the number of features for which they have value 1, divided by the total number of features. Thus, all individuals in $\mathcal{C}'$ get utility $\kappa_1$ from $a$, so $u_{\mathcal{C}',p}(a) = \kappa_1$. Furthermore, the individuals of type $y_1$ get utility 1 and the individuals of a type in $Y_{u_{\min}}$ get utility $u_{\min}$. Note that thus all individuals $y \in \mathcal{I}$ get a utility in $[u_{\min}, 1]$ from $a$, as needed. The share of individuals with utility 1 in $\mathcal{C}$ is

$$\frac{(\ell\kappa_1 - \ell u_{\min})\binom{\ell}{\ell u_{\min}}}{(\ell\kappa_1 - \ell u_{\min})\binom{\ell}{\ell u_{\min}} + (\ell - \ell\kappa_1)\binom{\ell}{\ell u_{\min}}} = \frac{\kappa_1 - u_{\min}}{1 - u_{\min}}$$

and the share of with utility $u_{\min}$ in $\mathcal{C}$ is $1 - \frac{\kappa_1 - u_{\min}}{1 - u_{\min}} = \frac{1 - \kappa_1}{1 - u_{\min}}$. Applying the definition of panel welfare, we find that $u_{\mathcal{C},p}(a)$ is as specified in the lemma. $\qquad\square$

*Proof of Theorem 8, lower bound.* Given any $u_{\min} \in [0, 1)$ and $p < 1$, we'll show that we can construct a population $\mathcal{P}$, an alternative $a$ so that the utility of every individual in the population for $a$ is in $[u_{\min}, 1]$, and a 1-representative panel $\mathcal{C}$ in $\mathcal{P}$, such that $|u_{\mathcal{C},p}(a) - u_{\mathcal{P},p}(a)|$ gets as close to $\varepsilon$ as desired (in particular, greater than any fixed $\varepsilon' < \varepsilon$) if the size of the population and the number of features is sufficiently large. This implies that there are 1-representative panels $\mathcal{C}$ that are not $\varepsilon'$-accurate for any $\varepsilon' < \varepsilon$. To make it clear that $\varepsilon$ depends on $p$ and $u_{\min}$, we'll write

$$\varepsilon(p, u_{\min}) = \begin{cases} \frac{1-p}{p}\left(p \cdot \frac{1-u_{\min}}{1-u_{\min}{}^p}\right)^{\frac{1}{1-p}} + \frac{u_{\min} - u_{\min}{}^p}{1 - u_{\min}{}^p} & \text{for } p > 0 \lor (p < 0 \land u_{\min} > 0) \\ 1 + \frac{1-u_{\min}}{\ln(1/u_{\min})}\left(\ln\left(\frac{1-u_{\min}}{\ln(1/u_{\min})}\right) - 1\right) & \text{for } p = 0 \land u_{\min} > 0 \\ 1 & \text{for } p \leq 0 \land u_{\min} = 0 \end{cases}$$

for $\varepsilon$ as defined in the theorem statement.

Let's first consider the case that $p \leq 0$, $u_{\min} = 0$, where we want to show that for any $\varepsilon' < 1 = \varepsilon(p, u_{\min})$ there exists a 1-representative panel that is not $\varepsilon'$-accurate. By Theorem 11, we know that for any $|\mathcal{F}|$, there exists a 1-representative panel that is not $\varepsilon$-accurate for any $\varepsilon < (|\mathcal{F}| - 1)/|\mathcal{F}|$. If we set $|\mathcal{F}|$ large enough in terms of $\varepsilon'$, for example $|\mathcal{F}| > 1 + \lceil 1/(1 - \varepsilon') \rceil$, we get that

$$\frac{|\mathcal{F}| - 1}{|\mathcal{F}|} = \frac{\lceil 1/(1 - \varepsilon') \rceil}{1 + \lceil 1/(1 - \varepsilon') \rceil} > \varepsilon',$$

so there exists a 1-representative panel that is not $\varepsilon'$-accurate.

We now turn to the case $p < 1$ and $u_{\min} \in (0, 1)$ or $p \in (0, 1)$ and $u_{\min} = 0$, noting that these cases cover all of "$p > 0$", "$p < 0 \land u_{\min} > 0$," and "$p = 0 \land u_{\min} > 0$," which are all remaining cases. We let

$$\kappa_1 = \begin{cases} \frac{1-u_{\min}}{1-u_{\min}{}^p}\left(p \cdot \frac{1-u_{\min}}{1-u_{\min}{}^p}\right)^{\frac{p}{1-p}} + \frac{u_{\min} - u_{\min}{}^p}{1 - u_{\min}{}^p} & \text{if } p \neq 0 \\ 1 - (1 - u_{\min})\frac{\ln\left(\frac{1}{1-u_{\min}}\right) + \ln\ln\left(\frac{1}{u_{\min}}\right)}{\ln\left(\frac{1}{u_{\min}}\right)} & \text{if } p = 0 \end{cases},$$

to be the value in $(u_{\min}, 1)$ maximizing the difference between $\text{UMIN}(\kappa_1; p, u_{\min})$ and $\text{UMAX}(\kappa_1; p, u_{\min})$, as found in Equations (3) and (4), so that

$$\text{UMAX}(\kappa_1; p, u_{\min}) - \text{UMIN}(\kappa_1; p, u_{\min}) = \varepsilon(p, u_{\min})$$

as shown in the proof of the first part of this theorem.

We now consider a fixed $|\mathcal{F}| = \ell$, and let $\hat{u}_{\min} = \lceil \ell u_{\min} \rceil / \ell$ and $\hat{\kappa}_1 = \lceil \ell\kappa_1 \rceil / \ell$ be the smallest rational numbers with $\ell$ as denominator that are greater than $u_{\min}$ and $\kappa_1$, respectively. We assume $\ell$ is large enough so that $1 > \hat{\kappa}_1 > \hat{u}_{\min}$, which is always possible since $\kappa_1 \in (u_{\min}, 1)$. Then, Lemma 19 tells us that for any $p < 1, u_{\min} \in (0, 1)$ or $p \in (0, 1), u_{\min} = 0$, there exist two panels $\mathcal{C} = \{(r_1, y_1), \ldots, (r_T, y_T)\}$, $\mathcal{C}' = \{(r_1', y_1), \ldots, (r_T', y_T)\}$ with $\ell$ features, so that each feature value appears in the same frequency in $\mathcal{C}$ and $\mathcal{C}'$, and that there exists an alternative $a$ that has the maximum and minimum social welfare possible for a panel with first moment $\hat{\kappa}_1$ and minimum population utility $\hat{u}_{\min}$, i.e., $\text{UMAX}(\hat{\kappa}_1; p, \hat{u}_{\min})$ and $\text{UMIN}(\hat{\kappa}_1; p, \hat{u}_{\min})$, respectively.

Now, note that as $\ell \to \infty$, we get that $\hat{\kappa}_1 \to \kappa_1$ and $\hat{u}_{\min} \to u_{\min}$. Since $\text{UMAX}(\kappa_1; p, u_{\min})$ and $\text{UMIN}(\kappa_1; p, u_{\min})$ are continuous for $p < 1, u_{\min} \in (0, 1)$ or $p \in (0, 1), u_{\min} = 0$, and $\kappa_1 \in [u_{\min}, 1]$, we get that as $\ell \to \infty$,

$$\text{UMAX}(\hat{\kappa}_1; p, \hat{u}_{\min}) \to \text{UMAX}(\kappa_1; p, u_{\min}), \qquad \text{UMIN}(\hat{\kappa}_1; p, \hat{u}_{\min}) \to \text{UMIN}(\kappa_1; p, u_{\min}),$$

and therefore also

$$|u_{\mathcal{C}',p}(a) - u_{\mathcal{C},p}(a)| = \text{UMAX}(\hat{\kappa}_1; p, \hat{u}_{\min}) - \text{UMIN}(\hat{\kappa}_1; p, \hat{u}_{\min}) \to \varepsilon(p, u_{\min}).$$

We now construct a population $\mathcal{P}$ that consists of the individuals in $\mathcal{C}$ once and $n'$ times each individual in $\mathcal{C}'$. Since each feature value appears in the same share of $\mathcal{C}$ and $\mathcal{C}'$, it also appears in the same share of $\mathcal{P}$, so $\mathcal{C}$ and $\mathcal{C}'$ are 1-representative panels in $\mathcal{P}$. If we let $n' \to \infty$, we see that

$$u_{\mathcal{P},p}(a) = \sum_{y_i \in \mathcal{I}} \frac{r_i + n'r'_i}{n' + 1}(u_i(a))^p \to \sum_{y_i \in \mathcal{I}} r'_i(u_i(a))^p = u_{\mathcal{C}',p}(a),$$

so $|u_{\mathcal{P},p}(a) - u_{\mathcal{C},p}(a)| \to |u_{\mathcal{C}',p}(a) - u_{\mathcal{C},p}(a)|$ as $n' \to \infty$.

Putting it all together, we get that as $\ell, n' \to \infty$, $|u_{\mathcal{P},p}(a) - u_{\mathcal{C},p}(a)| \to \varepsilon(p, u_{\min})$ for the 1-representative panel $\mathcal{C}$ in $\mathcal{P}$. In particular, this implies that for any $\delta > 0$, for all $\ell$ and $n'$ large enough, $|\varepsilon(p, u_{\min}) - |u_{\mathcal{P},p}(a) - u_{\mathcal{C},p}(a)|| < \delta$. If we set $\delta = \varepsilon(p, u_{\min}) - \varepsilon' > 0$ (and notice that always $|u_{\mathcal{P},p}(a) - u_{\mathcal{C},p}(a)| \leq \varepsilon(p, u_{\min})$ as shown in the first, upper bound, part of the proof), it follows that for any $\varepsilon' < \varepsilon(p, u_{\min})$, for all $\ell$ and $n'$ large enough, $|u_{\mathcal{P},p}(a) - u_{\mathcal{C},p}(a)| \in (\varepsilon', \varepsilon(p, u_{\min})]$, and in particular, $|u_{\mathcal{P},p}(a) - u_{\mathcal{C},p}(a)| > \varepsilon'$. Thus, $\mathcal{C}$ is a 1-representative panel that is not $\varepsilon'$ accurate. $\qquad \square$

### A.4.2    2-representative panels

To simplify notation moving forward, we define $\lambda = \frac{\kappa_2 - u_{\min}\kappa_1}{\kappa_1 - u_{\min}}$ and $\lambda' = \frac{\kappa_1 - \kappa_2}{1 - \kappa_1}$. These are the values of the $x$'s in the optimization programs for $\text{UMIN}(\kappa_1, \kappa_2; p, u_{\min})$ and $\text{UMAX}(\kappa_1, \kappa_2; p, u_{\min})$, respectively, that are not equal to $u_{\min}$ and 1, respectively, as obtained in Lemma 18.

**Lemma 20.** *Let $p < 1$ and $u_{\min} \in [0, 1)$, or $p \in (0, 1)$ and $u_{\min} = 0$. Let $\ell$ be an integer, $\kappa_1, \kappa_2 \in \mathbb{Q}$ so that $1 > \kappa_1 > \frac{\kappa_2 + u_{\min}}{1 + u_{\min}}$ and $\kappa_2 > \kappa_1^2 > u_{\min}$, as well as $\ell u_{\min}, \ell\lambda, \ell\lambda' \in \mathbb{Z}$. Then, there exist two panels $\mathcal{C} = \{(r_1, y_1), \dots, (r_T, y_T)\}$, $\mathcal{C}' = \{(r'_1, y_1), \dots, (r'_T, y_T)\}$ with individuals of types in $\mathcal{I} = \{y_1, \dots, y_T\}$ with $|\mathcal{F}| = \ell$ features and an alternative $a$ s.t. $u_i(a) \in [u_{\min}, 1]$ for all $y_i \in \mathcal{I}$, so that each tuple of up to two feature values appears in the same share in $\mathcal{C}$ and $\mathcal{C}'$, i.e., for every $F \subseteq \mathcal{F}$ such that $|F| \leq 2$ and $(v_f)_{f \in F}$, it holds that $\sum_{y^i \in \mathcal{I}} r_i \prod_{f \in F} y_i(f, v_f) = \sum_{y^i \in \mathcal{I}} r'_i \prod_{f \in F} y_i(f, v_f)$, and $u_{\mathcal{C},p}(a) = \text{UMIN}(\kappa_1, \kappa_2; p, u_{\min})$ and $u_{\mathcal{C}',p}(a) = \text{UMAX}(\kappa_1, \kappa_2; p, u_{\min})$.*

*Proof.* Let $\mathcal{F} = \{f_1, \dots, f_\ell\}$ and $V_f = \{0, 1\}$ for all $f \in \mathcal{F}$. We let $y_1$ be the type that has value 1 for all features, i.e. $y_1(f, i) = i$ for all $f \in \mathcal{F}, i = 0, 1$. We let

$$Y_\lambda = \{y : |\{f \in \mathcal{F} : y(f, 1) = 1\}| = \ell\lambda\}, \quad Y_{\lambda'} = \{y : |\{f \in \mathcal{F} : y(f, 1) = 1\}| = \ell\lambda'\}, \text{ and}$$

$$Y_{u_{\min}} = \{y : |\{f \in \mathcal{F} : y(f, 1) = 1\}| = \ell u_{\min}\}$$

be the set of all possible types that have value 1 for exactly $\ell\lambda$, $\ell\lambda'$, and $\ell u_{\min}$ of their features, respectively. We let $\mathcal{I} = \{y_1\} \cup Y_\lambda \cup Y_{\lambda'} \cup Y_{u_{\min}}$.

Let's now define the panels. We let $\mathcal{C}$ be the panel consisting of $c(\kappa_1 - u_{\min})^2 \binom{\ell}{\ell u_{\min}}$ individuals of each of the $\binom{\ell}{\ell\lambda}$ types in $Y_\lambda$ and of $c(\kappa_2 - \kappa_1^2)\binom{\ell}{\ell\lambda} \in \mathbb{Z}$ individuals of each of the $\binom{\ell}{\ell u_{\min}}$ types in $Y_{u_{\min}}$, where $c$ is any number so that $c(\kappa_1 - u_{\min})^2 \binom{\ell}{\ell u_{\min}}, c(\kappa_2 - \kappa_1^2)\binom{\ell}{\ell\lambda} \in \mathbb{Z}$. We let $\mathcal{C}'$ be the panel consisting of $c'(1 - \kappa_1)^2$ individuals of each of the $\binom{\ell}{\ell\lambda'}$ types in $Y_{\lambda'}$ and of $c'(\kappa_2 - \kappa_1^2)\binom{\ell}{\ell\lambda'} \in \mathbb{Z}$ individuals of type $y_1$, where $c'$ is any number so that $c'(1 - \kappa_1)^2 \binom{\ell}{\ell u_{\min}}, c'(\kappa_2 - \kappa_1^2)\binom{\ell}{\ell\lambda} \in \mathbb{Z}$.

We now confirm that for every feature $f \in \mathcal{F}$, each value $v \in V_f$ appears in the same share of $\mathcal{C}$ and $\mathcal{C}'$. Since there are only two values, 0 and 1, for each feature, it suffices to show that value 1 appears in the same share of individuals in $\mathcal{C}$ and $\mathcal{C}'$ for any feature.

To keep the math concise moving forward, we'll define binomial coefficients with a negative bottom entry as zero. Here, this precisely handles the cases $u_{\min} < 1/\ell$, i.e. $u_{\min} = 0$, $\lambda < 1/\ell$, i.e. $\lambda = 0$, and $\lambda' < 1/\ell$, i.e. $\lambda' = 0$.

Consider any feature $f \in \mathcal{F}$. For $\mathcal{C}$, we get that a fraction of

$$\frac{c(\kappa_1 - u_{\min})^2 \binom{\ell}{\ell u_{\min}}\binom{\ell-1}{\ell\lambda-1} + c(\kappa_2 - \kappa_1^2)\binom{\ell}{\ell\lambda}\binom{\ell-1}{\ell u_{\min}-1}}{c(\kappa_1 - u_{\min})^2 \binom{\ell}{\ell u_{\min}}\binom{\ell}{\ell\lambda} + c(\kappa_2 - \kappa_1^2)\binom{\ell}{\ell\lambda}\binom{\ell}{\ell u_{\min}}} = \frac{(\kappa_1 - u_{\min})^2\lambda + (\kappa_2 - \kappa_1^2)u_{\min}}{(\kappa_1 - u_{\min})^2 + (\kappa_2 - \kappa_1^2)} =$$

$$= \frac{(\kappa_1 - u_{\min})(\kappa_2 - u_{\min}\kappa_1) + (\kappa_2 - \kappa_1^2)u_{\min}}{(\kappa_1 - u_{\min})^2 + (\kappa_2 - \kappa_1^2)} = \kappa_1$$

of the individuals in $\mathcal{C}$ have value 1 for feature $f$. For $\mathcal{C}'$, we get that a fraction of

$$\frac{c'(1 - \kappa_1)^2 \binom{\ell-1}{\ell\lambda'-1} + c'(\kappa_2 - \kappa_1^2)\binom{\ell}{\ell\lambda'}}{c'(1 - \kappa_1)^2 \binom{\ell}{\ell\lambda'} + c'(\kappa_2 - \kappa_1^2)\binom{\ell}{\ell\lambda'}} = \frac{(1 - \kappa_1)^2\lambda' + (\kappa_2 - \kappa_1^2)}{(1 - \kappa_1)^2 + (\kappa_2 - \kappa_1^2)} =$$

$$= \frac{(1 - \kappa_1)(\kappa_1 - \kappa_2) + (\kappa_2 - \kappa_1^2)}{(1 - \kappa_1)^2 + (\kappa_2 - \kappa_1^2)} = \kappa_1$$

of the individuals in $\mathcal{C}'$ have value 1 for feature $f$.

Let's now confirm that for every pair of features $f_1, f_2 \in \mathcal{F}$, each pair of values $(v_1, v_2) \in V_{f_1} \times V_{f_2}$ appears in the same share of $\mathcal{C}$ and $\mathcal{C}'$. Since there are only two values, 0 and 1, for each feature, by symmetry it suffices to show that values $(1,1)$ and $(0,0)$ appear in the same share of individuals in $\mathcal{C}$ and $\mathcal{C}'$ for any pair of features. Consider any features $f_1, f_2 \in \mathcal{F}$. The case $f_1 = f_2$ follows from the single-feature case above, so we assume $f_1 \neq f_2$. We get that the fraction of individuals in $C$ that have value 1 for $f_1$ and $f_2$ are

$$\frac{c(\kappa_1 - u_{\min})^2 \binom{\ell}{\ell u_{\min}}\binom{\ell-2}{\ell\lambda-2} + c(\kappa_2 - \kappa_1^2)\binom{\ell}{\ell\lambda}\binom{\ell-2}{\ell u_{\min}-2}}{c(\kappa_1 - u_{\min})^2 \binom{\ell}{\ell u_{\min}}\binom{\ell}{\ell\lambda} + c(\kappa_2 - \kappa_1^2)\binom{\ell}{\ell\lambda}\binom{\ell}{\ell u_{\min}}} =$$

$$= \frac{(\kappa_1 - u_{\min})^2 \frac{\lambda(\ell\lambda-1)}{(\ell-1)} + (\kappa_2 - \kappa_1^2)\frac{u_{\min}(\ell u_{\min}-1)}{(\ell-1)}}{(\kappa_1 - u_{\min})^2 + (\kappa_2 - \kappa_1^2)} = \kappa_1 - \frac{l(\kappa_1 - \kappa_2)}{\ell - 1}.$$

We get that the fraction of individuals in $C'$ that have value 1 for $f_1$ and $f_2$ is

$$\frac{c'(1 - \kappa_1)^2 \binom{\ell-2}{\ell\lambda'-2} + c'(\kappa_2 - \kappa_1^2)\binom{\ell}{\ell\lambda'}}{c'(1 - \kappa_1)^2 \binom{\ell}{\ell\lambda'} + c'(\kappa_2 - \kappa_1^2)\binom{\ell}{\ell\lambda'}} = \frac{(1 - \kappa_1)^2 \frac{\lambda'(\ell\lambda'-1)}{(l-1)} + (\kappa_2 - \kappa_1^2)}{(1 - \kappa_1)^2 + (\kappa_2 - \kappa_1^2)} = \kappa_1 - \frac{l(\kappa_1 - \kappa_2)}{\ell - 1}.$$

Let's now turn to the share of individuals with values $(0,0)$ for some pair of features. We get that the fraction of individuals in $C$ that have value 0 for $f_1$ and $f_2$ are

$$\frac{c(\kappa_1 - u_{\min})^2 \binom{\ell}{\ell u_{\min}}\binom{\ell-2}{\ell\lambda} + c(\kappa_2 - \kappa_1^2)\binom{\ell}{\ell\lambda}\binom{\ell-2}{\ell u_{\min}}}{c(\kappa_1 - u_{\min})^2 \binom{\ell}{\ell u_{\min}}\binom{\ell}{\ell\lambda} + c(\kappa_2 - \kappa_1^2)\binom{\ell}{\ell\lambda}\binom{\ell}{\ell u_{\min}}} =$$

$$= \frac{(\kappa_1 - u_{\min})^2 \frac{(1-\lambda)(\ell-\ell\lambda-1)}{(\ell-1)} + (\kappa_2 - \kappa_1^2)\frac{(1-u_{\min})(\ell-\ell u_{\min}-1)}{(\ell-1)}}{(\kappa_1 - u_{\min})^2 + (\kappa_2 - \kappa_1^2)} = 1 - \kappa_1 - \frac{l(\kappa_1 - \kappa_2)}{\ell - 1}.$$

We get that the fraction of individuals in $C'$ that have value 0 for $f_1$ and $f_2$ is

$$\frac{c'(1 - \kappa_1)^2 \binom{\ell-2}{\ell\lambda'}}{c'(1 - \kappa_1)^2 \binom{\ell}{\ell\lambda'} + c'(\kappa_2 - \kappa_1^2)\binom{\ell}{\ell\lambda'}} = \frac{(1 - \kappa_1)^2 \frac{(1-\lambda')(\ell-\ell\lambda'-1)}{(l-1)}}{(1 - \kappa_1)^2 + (\kappa_2 - \kappa_1^2)} = 1 - \kappa_1 - \frac{l(\kappa_1 - \kappa_2)}{\ell - 1}.$$

We note that in the math above, setting binomial coefficients with a negative bottom entry as zero handles precisely the cases $u_{\min} < 2/\ell$, i.e. $\ell u_{\min} \in \{0, 1\}$, $\lambda < 2/\ell$, i.e. $\ell\lambda \in \{0, 1\}$, and $\lambda' < 2/\ell$, i.e. $\ell\lambda' \in \{0, 1\}$.

We now let $a$ be an alternative such that $a(f, i) = i/\ell$ for all $f \in \mathcal{F}$ and $i = 0, 1$; i.e., an individual's utility is the number of features for which they have value 1, divided by the total number of features. Thus, the individuals of type $y_1$ get utility 1, the individuals of a type in $Y_\lambda$ get utility $\lambda$, the individuals of a type in $Y_{\lambda'}$ get utility $\lambda'$, and the individuals of a type in $Y_{u_{\min}}$ get utility $u_{\min}$. Note that thus all individuals $y \in \mathcal{I}$ get a utility in $[u_{\min}, 1]$ from $a$, as needed.

The share of individuals with utility $\lambda$ in $\mathcal{C}$ is

$$\frac{c(\kappa_1 - u_{\min})^2 \binom{\ell}{\ell u_{\min}}\binom{\ell}{\ell\lambda}}{c(\kappa_1 - u_{\min})^2 \binom{\ell}{\ell u_{\min}}\binom{\ell}{\ell\lambda} + c(\kappa_2 - \kappa_1^2)\binom{\ell}{\ell\lambda}\binom{\ell}{\ell u_{\min}}} = \frac{(\kappa_1 - u_{\min})^2}{\kappa_2 - 2\kappa_1 u_{\min} + u_{\min}^2}$$

and the share of individuals with utility $u_{\min}$ in $\mathcal{C}$ is

$$\frac{c(\kappa_2 - \kappa_1^2)\binom{\ell}{\ell\lambda}\binom{\ell}{\ell u_{\min}}}{c(\kappa_1 - u_{\min})^2 \binom{\ell}{\ell u_{\min}}\binom{\ell}{\ell\lambda} + c(\kappa_2 - \kappa_1^2)\binom{\ell}{\ell\lambda}\binom{\ell}{\ell u_{\min}}} = \frac{\kappa_2 - \kappa_1^2}{\kappa_2 - 2\kappa_1 u_{\min} + u_{\min}^2}.$$

The share of individuals with utility 1 in $\mathcal{C}'$ is

$$\frac{c'(1 - \kappa_1)^2 \binom{\ell}{\ell\lambda'}}{c'(1 - \kappa_1)^2 \binom{\ell}{\ell\lambda'} + c'(\kappa_2 - \kappa_1^2)\binom{\ell}{\ell\lambda'}} = \frac{(1 - \kappa_1)^2}{1 - 2\kappa_1 + \kappa_2}$$

and the share of individuals with utility $\lambda'$ in $\mathcal{C}'$ is

$$\frac{c'(\kappa_2 - \kappa_1^2)\binom{\ell}{\ell\lambda'}}{c'(1 - \kappa_1)^2 \binom{\ell}{\ell\lambda'} + c'(\kappa_2 - \kappa_1^2)\binom{\ell}{\ell\lambda'}} = \frac{(\kappa_2 - \kappa_1^2)}{1 - 2\kappa_1 + \kappa_2}.$$

Applying the definition of panel welfare, we find that $u_{\mathcal{C},p}(a)$ and $u_{\mathcal{C}',p}(a)$ are as specified in the lemma statement. $\qquad\square$

*Proof of Theorem 9, lower bound.* Given any $u_{\min} \in [0, 1)$ and $p < 1$, we'll show that we can construct a population $\mathcal{P}$, an alternative $a$ so that the utility of every individual in the population for $a$ is in $[u_{\min}, 1]$, and a 2-representative panel $\mathcal{C}$ in $\mathcal{P}$, such that $|u_{\mathcal{C},p}(a) - u_{\mathcal{P},p}(a)|$ gets as close to $\varepsilon$ as desired (in particular, greater than any fixed $\varepsilon' < \varepsilon$) if the size of the population and the number of features is sufficiently large. This implies that there are 2-representative panels $\mathcal{C}$ that are not $\varepsilon'$-accurate for any $\varepsilon' < \varepsilon$.

To make it clear that $\varepsilon$ depends on $p$ and $u_{\min}$, we'll write $\varepsilon(p, u_{\min})$ for the $\varepsilon$ we get from Theorem 9 for some $p$ and $u_{\min}$.

Let's first consider the case that $p \leq 0$, $u_{\min} = 0$, where we want to show that for any $\varepsilon' < 1 = \varepsilon(p, u_{\min})$ there exists a 1-representative panel that is not $\varepsilon'$-accurate. By Theorem 11, we know that for any $|\mathcal{F}|$, there exists a 2-representative panel that is not $\varepsilon$-accurate for any $\varepsilon < (|\mathcal{F}| - 2)/|\mathcal{F}|$. If we set $|\mathcal{F}|$ large enough in terms of $\varepsilon'$, for example $|\mathcal{F}| > 1 + \lceil 2/(1 - \varepsilon') \rceil$, we get that

$$\frac{|\mathcal{F}| - 2}{|\mathcal{F}|} = \frac{\lceil 2/(1 - \varepsilon') \rceil - 1}{\lceil 2/(1 - \varepsilon') \rceil + 1} > \varepsilon',$$

so there exists a 2-representative panel that is not $\varepsilon'$-accurate.

We now turn to the case $p < 1$ and $u_{\min} \in (0, 1)$ or $p \in (0, 1)$ and $u_{\min} = 0$, noting that these cases cover all of "$p > 0$", "$p < 0 \wedge u_{\min} > 0$," and "$p = 0 \wedge u_{\min} > 0$," which are all remaining cases. We fix any such $p$ and $u_{\min}$. We write $\kappa_1^*$ and $\kappa_2^*$ as the values of $\kappa_1$ and $\kappa_2$ that achieve the optimum in the optimization program from the theorem statement, i.e. being the values maximizing the difference between $\text{UMIN}(\kappa_1, \kappa_2; p, u_{\min})$ and $\text{UMAX}(\kappa_1, \kappa_2; p, u_{\min})$ so that

$$\text{UMAX}(\kappa_1^*, \kappa_2^*; p, u_{\min}) - \text{UMIN}(\kappa_1^*, \kappa_2^*; p, u_{\min}) = \varepsilon(p, u_{\min}).$$

We let $c$ be an integer and $\hat{\kappa}_1^* = \lceil c\kappa_1^* \rceil/c$, $\hat{\kappa}_2^* = \lceil c\kappa_1^* \rceil/c$, and $\hat{u}_{\min} = \lceil cu_{\min} \rceil/c$ be the smallest numbers with $c$ as denominator that are greater than $\kappa_1^*$, $\kappa_2^*$, and $u_{\min}$, respectively. Furthermore, we pick $c$ large enough so that $1 > \hat{\kappa}_1^* > \frac{\hat{\kappa}_2^* + \hat{u}_{\min}}{1 + \hat{u}_{\min}}$ and $\hat{\kappa}_2^* > (\hat{\kappa}_1^*)^2 > \hat{u}_{\min}$, which we know is possible since $1 > \kappa_1^* > \frac{\kappa_2^* + u_{\min}}{1 + u_{\min}}$ and $\kappa_2^* > (\kappa_1^*)^2 > u_{\min}$.

We now consider a set of $|\mathcal{F}| = \ell = c^2(\hat{\kappa}_1^* - \hat{u}_{\min})(1 - \hat{\kappa}_1^*) \in \mathbb{Z}$ features and let $\hat{\lambda} = \frac{\hat{\kappa}_2^* - \hat{u}_{\min}\hat{\kappa}_1^*}{\hat{\kappa}_1^* - \hat{u}_{\min}}$ and $\hat{\lambda}' = \frac{\hat{\kappa}_1^* - \hat{\kappa}_2^*}{1 - \hat{\kappa}_1^*}$. We get that $\ell\hat{\lambda}, \ell\hat{\lambda}' \in \mathbb{Z}$. Lemma 20 tells us that there exist two panels $\mathcal{C} = \{(r_1, y_1), \ldots, (r_T, y_T)\}$, $\mathcal{C}' = \{(r_1', y_1), \ldots, (r_T', y_T)\}$ with $\ell$ features, so that each tuple of up to two feature values appears in the same frequency in $\mathcal{C}$ and $\mathcal{C}'$, and that there exists an

alternative $a$ that has the maximum and minimum social welfare possible for a panel with first moment $\hat{\kappa}_1^*$, second moment $\hat{\kappa}_2^*$, and minimum population utility $\hat{u}_{\min}$, i.e., $\text{UMAX}(\hat{\kappa}_1^*, \hat{\kappa}_2^*; p, \hat{u}_{\min})$ and $\text{UMIN}(\hat{\kappa}_1^*, \hat{\kappa}_2^*; p, \hat{u}_{\min})$, respectively.

Now, note that as $c \to \infty$, we get that $\hat{\kappa}_1^* \to \kappa_1^*$, $\hat{\kappa}_1^* \to \kappa_1^*$, and $\hat{u}_{\min} \to u_{\min}$. Since $\text{UMAX}(\kappa_1, \kappa_2; p, u_{\min})$ and $\text{UMIN}(\kappa_1, \kappa_2; p, u_{\min})$ are continuous for $p < 1, u_{\min} \in (0, 1)$ or $p \in (0, 1), u_{\min} = 0$, and $\kappa_1$ and $\kappa_2$ in the feasible region, we get that as $c \to \infty$,

$$\text{UMAX}(\hat{\kappa}_1^*, \hat{\kappa}_2^*; p, \hat{u}_{\min}) \to \text{UMAX}(\kappa_1^*, \kappa_2^*; p, u_{\min}),$$
$$\text{UMIN}(\hat{\kappa}_1^*, \hat{\kappa}_2^*; p, \hat{u}_{\min}) \to \text{UMIN}(\kappa_1^*, \kappa_2^*; p, u_{\min}),$$

and therefore also

$$|u_{\mathcal{C}',p}(a) - u_{\mathcal{C},p}(a)| = \text{UMAX}(\hat{\kappa}_1^*, \hat{\kappa}_2^*; p, \hat{u}_{\min}) - \text{UMIN}(\hat{\kappa}_1^*, \hat{\kappa}_2^*; p, \hat{u}_{\min}) \to \varepsilon(p, u_{\min}).$$

We now construct a population $\mathcal{P}$ that consists of the individuals in $\mathcal{C}$ once and $n'$ times each individual in $\mathcal{C}'$. Since each tuple of up to two feature values appears in the same share of $\mathcal{C}$ and $\mathcal{C}'$, it also appears in the same share of $\mathcal{P}$, so $\mathcal{C}$ and $\mathcal{C}'$ are 2-representative panels in $\mathcal{P}$. If we let $n' \to \infty$, we see that

$$u_{\mathcal{P},p}(a) = \sum_{y_i \in \mathcal{I}} \frac{r_i + n'r_i'}{n' + 1}(u_i(a))^p \to \sum_{y_i \in \mathcal{I}} r_i'(u_i(a))^p = u_{\mathcal{C}',p}(a),$$

so $|u_{\mathcal{P},p}(a) - u_{\mathcal{C},p}(a)| \to |u_{\mathcal{C}',p}(a) - u_{\mathcal{C},p}(a)|$ as $n' \to \infty$.

Putting it all together, we get that as $c, n' \to \infty$, $|u_{\mathcal{P},p}(a) - u_{\mathcal{C},p}(a)| \to \varepsilon(p, u_{\min})$ for the 2-representative panel $\mathcal{C}$ in $\mathcal{P}$. In particular, this implies that for any $\delta > 0$, for all $c$ and $n'$ large enough, $||\varepsilon(p, u_{\min}) - |u_{\mathcal{P},p}(a) - u_{\mathcal{C},p}(a)|| < \delta$. If we set $\delta = \varepsilon(p, u_{\min}) - \varepsilon' > 0$ (and notice that always $|u_{\mathcal{P},p}(a) - u_{\mathcal{C},p}(a)| \le \varepsilon(p, u_{\min})$ as shown in the first, upper bound, part of the proof), it follows that for any $\varepsilon' < \varepsilon(p, u_{\min})$, for all $c$ and $n'$ large enough, $|u_{\mathcal{P},p}(a) - u_{\mathcal{C},p}(a)| \in (\varepsilon', \varepsilon(p, u_{\min})]$, and in particular, $|u_{\mathcal{P},p}(a) - u_{\mathcal{C},p}(a)| > \varepsilon'$. Thus, $\mathcal{C}$ is a 2-representative panel in $\mathcal{P}$ that is not $\varepsilon'$ accurate. $\qquad\square$

## A.5  Proof of Theorem 11

To keep the math in the proof concise, we use the (well-known) extension of binomial coefficients to allow for negative integers in the top part. We then apply the Chu-Vandermonde identity for generalized binomial coefficients multiple times throughout the proof.

**Definition 21** ([13, Definition 3.1.]). Let $k \in \mathbb{Z}_{\ge 0}$ and $n \in \mathbb{Z}$. We define the generalized binomial coefficient as $\binom{n}{k} = \frac{n(n-1)...(n-k+1)}{k!}$.

There is a nice identity for binomial coefficients with a negative top part, which will be useful for the proof.

**Lemma 22** ([13, Proposition 3.16.]). *Let $k \in \mathbb{Z}_{\ge 0}$ and $n \in \mathbb{Z}$. Then*

$$\binom{n}{k} = (-1)^k \binom{k - n - 1}{k}.$$

A key part of our proof is the so called Chu-Vandermon identity, first stated by mathematician Chu Shih-chieh already in 1303.

**Theorem 23** ([13, Theorem 3.29.]). *Let $k \in \mathbb{Z}_{\ge 0}$, $n, m \in \mathbb{Z}$. Then*

$$\binom{n + m}{k} = \sum_{i=0}^{k} \binom{n}{i}\binom{m}{k - i}.$$

We refer to [13] for contemporary proofs of Lemma 22 and Theorem 23.

*Proof of Theorem 11.* For this proof, we'll construct two panels, $\mathcal{C}$ and $\mathcal{C}'$, of equal size that are $m$-representative with respect to each other, i.e., for every $\bar{m} \le m$, every $\bar{m}$-tuple of feature values appears in the same number of individuals in $\mathcal{C}$ and $\mathcal{C}'$. We'll construct an alternative $a$ that gives one

person in $\mathcal{C}$ utility 0, while every person in panel $\mathcal{C}'$ has utility at least $(|\mathcal{F}| - m)/|\mathcal{F}|$. We then let $\mathcal{P}$ be the population consisting of the individuals in $\mathcal{C}$ and $\mathcal{C}'$, so $\mathcal{C}'$ is a $m$-representative panel in $\mathcal{P}$ with welfare at least $(|\mathcal{F}| - m)/|\mathcal{F}|$ for $a$, while to overall population has welfare 0.

First, note that if $|\mathcal{F}| \leq m$, the bound in the theorem follows trivially. Thus, let $\mathcal{F} = \{f_1, \ldots, f_\ell\}$ for $\ell = |\mathcal{F}| > m$. Let $V_f = \{0, 1\}$ for all $f \in \mathcal{F}$. Let $y_\ell$ be the type that has value 0 for all features, i.e. $y_\ell(f, i) = 1 - i$ for all $f \in \mathcal{F}, i = 0, 1$. For $j \in \{0, \ldots, m\}$, we let $Y_j = \{y : |f \in \mathcal{F} : y(f, 0) = 1| = j\}$ be the set of all types that have value 0 for exactly $j$ features.

We define $\mathcal{C}$ to be the panel consisting of one individual of type $y_\ell$ and $\binom{\ell - j - 1}{m - j}$ individuals of each type $y \in Y_j$, for $j \in \{0, \ldots, m\}, j \not\equiv m \mod 2$. We define $\mathcal{C}'$ to be the panel consisting of $\binom{\ell - j - 1}{m - j}$ individuals of each type $y \in Y_j$, for $j \in \{0, \ldots, m\}, j \equiv m \mod 2$. We'll first prove that the number of individuals in $\mathcal{C}$, $k$, and in $\mathcal{C}'$, denoted $k'$, is indeed the same. Let's first calculate $k$ and $k'$:

$$k = 1 + \sum_{\substack{j \in \{0, \ldots, m\} \\ j \not\equiv m \mod 2}} \binom{\ell - j - 1}{m - j} |Y_j| = 1 - \sum_{\substack{j \in \{0, \ldots, m\} \\ j \not\equiv m \mod 2}} \binom{m - \ell}{m - j}\binom{\ell}{j},$$

$$k' = \sum_{\substack{j \in \{0, \ldots, m\} \\ j \equiv m \mod 2}} \binom{\ell - j - 1}{m - j} |Y_j| = \sum_{\substack{j \in \{0, \ldots, m\} \\ j \equiv m \mod 2}} \binom{m - \ell}{m - j}\binom{\ell}{j},$$

where the two right equalities follow from Lemma 22 and the fact that $|Y_j| = \binom{\ell}{j}$. We now note that by Theorem 23,

$$\sum_{j=0}^{m} \binom{m - \ell}{m - j}\binom{\ell}{j} = \binom{m}{m} = 1,$$

which immediately implies $k = k'$.

Next, we'll argue that $\mathcal{C}$ and $\mathcal{C}'$ are $m$-representative in respect to each other, i.e., any $\bar{m}$-tuple for feature values for $\bar{m} \in [m]$ appears in $\mathcal{C}$ with the same frequency as $\mathcal{C}'$. Since $k = k'$, this means it appears in the same number of individuals in $\mathcal{C}$ as in $\mathcal{C}'$.

Let $F \subset \mathcal{F}, |F| = \bar{m}$, be any set of $\bar{m} \leq m$ features in $\mathcal{F}$ and let $\mathcal{V} = (v_f)_{f \in F} \in \bigtimes_{f \in F} V_f$ be any assignment of values to these features. Let $k_\mathcal{V}$ and $k'_\mathcal{V}$ be the number of individuals in $\mathcal{C}$ and $\mathcal{C}'$, respectively, with feature values $\mathcal{V}$. We thus want to show that $k_\mathcal{V} = k'_\mathcal{V}$ for all $\mathcal{V}$.

For any $\mathcal{V}$, let $i = |v_f \in \mathcal{V} : v_f = 0|$ be the number of zeros in this assignment. For $j = \{0, \ldots, m\}$, the number of types in $Y_j$ that have feature values $\mathcal{V}$ is $\binom{\ell - \bar{m}}{j - i}$ if $j \geq i$, since there are $\ell - \bar{m}$ feature values not defined by $\mathcal{V}$, out of which $j - i$ are zeros. If $j < i$, we know that no type in $Y_j$ has feature values $\mathcal{V}$, since types in $Y_j$ have value 0 for less features than required by $\mathcal{V}$. Type $y_0$ has feature values $\mathcal{V}$ if and only if $i = \bar{m}$ since it is all zeros..

We get that

$$k_\mathcal{V} = \mathbb{1}[i = \bar{m}] + \sum_{\substack{j \in \{i, \ldots, m\} \\ j \not\equiv m \mod 2}} \binom{\ell - j - 1}{m - j}\binom{\ell - \bar{m}}{j - i} = \mathbb{1}[i = \bar{m}] - \sum_{\substack{j \in \{i, \ldots, m\} \\ j \not\equiv m \mod 2}} \binom{m - \ell}{m - j}\binom{\ell - \bar{m}}{j - i},$$

$$k'_\mathcal{V} = \sum_{\substack{j \in \{i, \ldots, m\} \\ j \equiv m \mod 2}} \binom{\ell - j - 1}{m - j}\binom{\ell - \bar{m}}{j - i} = \sum_{\substack{j \in \{i, \ldots, m\} \\ j \equiv m \mod 2}} \binom{m - \ell}{m - j}\binom{\ell - \bar{m}}{j - i},$$

where again the two right equalities follow from Lemma 22. We now note that

$$\sum_{j=i}^{m} \binom{m - \ell}{m - j}\binom{\ell - \bar{m}}{j - i} = \sum_{j=0}^{m-i} \binom{m - \ell}{m - i - j}\binom{\ell - \bar{m}}{j} = \binom{m - \bar{m}}{m - i},$$

where the last equality follows from Theorem 23. Now, note that since $i \leq \bar{m}$, the bottom part of $\binom{m - \bar{m}}{m - i}$ is at least its top part, so by Definition 21, we get

$$\binom{m - \bar{m}}{m - i} = \begin{cases} 0 & \text{if } i < \bar{m} \\ 1 & \text{if } i = \bar{m} \end{cases} = \mathbb{1}[i = \bar{m}].$$

This immediately implies $k_{\mathcal{V}} = k'_{\mathcal{V}}$.

We let $\mathcal{P}$ be the population consisting of all individuals in $\mathcal{C}$ and $\mathcal{C}'$. Since for any $\bar{m} \leq m$, any $\bar{m}$-tuple of feature values appears in the same share of individuals in $\mathcal{C}$ and in $\mathcal{C}'$, we know they also appear in the same share in $\mathcal{P}$. Thus, $\mathcal{C}$ and $\mathcal{C}'$ are $m$-representative panels in $\mathcal{P}$.

We now define alternative $a$ such that $a(f, i) = i/\ell$ for all $f \in \mathcal{F}$, $i = 0, 1$. That is, an individual gets benefit $1/\ell$ for each feature value that is 1. Thus, individual $y_\ell$ has utility 0, and any individual $y \in Y_j$ for $j \in \{0, \ldots, m\}$ has utility $(\ell - j)/\ell$. In particular, note that the utility from $a$ for any individual in $\mathcal{P}$ is in $[0, 1]$, as required.

The $p$-mean, for $p \leq 0$, of a list of numbers including 0 is 0, so the welfare of $\mathcal{P}$ is $u_{\mathcal{P},p}(a) = 0$. However, all individuals in $\mathcal{C}'$ have at least utility $(\ell - m)/\ell$. Since for any $p$, the $p$-mean of a list of numbers is at least as big as the smallest number in the list, $u_{\mathcal{C}',p}(a) \geq \frac{\ell-m}{\ell}$. We conclude that for this alternative $a$ and $m$-representative panel $\mathcal{C}'$ in $\mathcal{P}$,

$$|u_{\mathcal{C}',p}(a) - u_{\mathcal{P},p}(a)| \geq \frac{\ell - m}{\ell}.$$

The theorem statement follows. $\qquad\square$

### A.6 Additional details and results from empirical experiments

Appendices A.6.1 to A.6.3 contain additional details on the datasets used for the empirical experiments. Appendix A.6.4 contains the plots for $\varepsilon$-approximate $m$-representative panels that are not in the main part of the paper.

#### A.6.1 European nation-wide panel

For this panel, 30,000 letters were mailed to invite individuals to participate. 1,727 individuals volunteered to be in the candidate pool; the desired panel size was 110. Seven features were being considered when selecting the panel, which we list in Table 2.

Table 2: The features considered in selecting panel EUR1

| Features | Values |
| --- | --- |
| Gender | Male, Female, ~~Other~~ |
| Age | 16–29, 30–44, 45–59, 60+ |
| Place of residence within the country | 12 distinct regions |
| Educational attainment | 3 levels |
| ~~Concern about climate change~~ | ~~Very concerned, Fairly concerned, Not very concerned, Not at all concerned, Other~~ |
| Ethnicity | White, Non-white |
| Residential density | Urban, Rural |

We obtained the underlying population data from the 2016 European Social Survey [19]. The dataset contains 1,959 respondents with post-stratification weights to account for sampling and participation bias. We dropped 44 people from the dataset: 40 because they had missing data and 4 because they were younger than 16, the minimum age set for the assembly. Furthermore, since the European Social Survey only has the fields 'Male' and 'Female' for gender, we removed the 12 individuals with gender 'Other' from the pool of candidates, reducing the size to 1,715.

One of the seven features considered was "Concern about climate change". Only 4 individuals in the pool of candidates had the value "Not at all concerned", making it impossible to even form a 1-representative panel of the desired size with this candidate pool. We thus excluded this feature from our analysis.

We note that due to its limited sample size of 1,959, the European Social Survey doesn't contain a person for all possible tuples of $m$ feature values, for $m \geq 3$. However, this is not concerning for our positive results on the existence of $m$-representative panels, since these are tuples of feature values that appear in a sufficiently small share of the population so that an $m$-representative panel would

satisfy the corresponding quota by containing 0 or 1 individuals with that tuple of feature values. In our experiments, for tuples of feature values that didn't appear in the European Social Survey, we enforced that 0 people on the panel have this tuple of feature values, tightening the constraints on the existence of $m$-representative panels.

### A.6.2 Australian state-wide panels 1 & 2

These two panels were being conducted in series with overlap in the pool of candidates and individuals on the panels. Initially, 25,000 letters were mailed to invite individuals to participate, with additional letters being sent to groups with expected low participation rates. 1,070 individuals volunteered to be in the first candidate pool, from which a panel of 50 candidates was selected. For the second panel, additional individuals were recruited for the candidate pool, then including 1,145 individuals (some of which were on the first panel). A panel of 328 individuals was selected. Four features were being considered when selecting the panels, which we list in Table 3.

Table 3: The features considered in selecting panels AUS1 and AUS2

| Features | Values |
|---|---|
| Sex | Male, Female |
| Age | 18–24, 25–34, 35–44, 45–54, 55–64, 65+ |
| Residential density | Metropolitan, Regional |
| ~~Residential status~~ | ~~Owner/Occupier, Tenant~~ |

We obtained the underlying population data from the Australian Bureau of Statistics' 2021 Census data [1]. Up to small deviations due to differential privacy noise being applied to the dataset, we obtained the exact population share of each feature value tuple including "Sex", "Age", and "Residential Density". However, the publicly available data from the Australian Census separates between individuals and dwellings, so that no data on the intersection of the "Residential status" feature with the features "Age" and "Sex" is available. We therefore dropped the residential status feature in our analysis.

### A.6.3 Australian state-wide panel 3

For this panel, individuals could self-register interest in participating. 3,500 individuals did so, and were emailed an invitation to participate. 518 of those individuals ended up volunteering to be in the candidate pool; the desired panel size was 45. Four features were being considered when selecting the panel, which we list in Table 4.

Table 4: The features considered in selecting panel AUS3

| Features | Values |
|---|---|
| Gender | Male, Female, ~~Non-binary~~, ~~I would prefer not to say~~ |
| Age | 18–24, 25–34, 35–44, 45–54, 55–64, 65–74, 75+ |
| Place of residence within the state | 3 distinct regions |
| Ability to attend all panel sessions | Yes, No, Unsure, Unsure-1 |

We obtained the underlying population data from the Australian Bureau of Statistics' 2021 Census data [1]. Since the Australian Census only has the fields 'Male' and 'Female' for sex, we removed the 9 individuals with gender 'Non-binary' or 'I would prefer not to say' from the pool of candidates, reducing the size to 509. Additionally, the feature 'Ability to attend all panel sessions' cannot be obtained for the underlying population, by itself or in intersection with other features. Thus, we only consider this feature individually, ensuring that at most $1/45$ of the individuals on the panel have each of the values 'No', 'Unsure', or 'Unsure-1', as was the requirement for the actually selected panel. For each feature value tuple including "Gender" (without "Other"), "Age", and "Place of residence",

we obtained the exact population share, up to small deviations due to differential privacy noise being applied to the dataset.

### A.6.4 Additional plots for $\varepsilon$-approximate $m$-representative panels

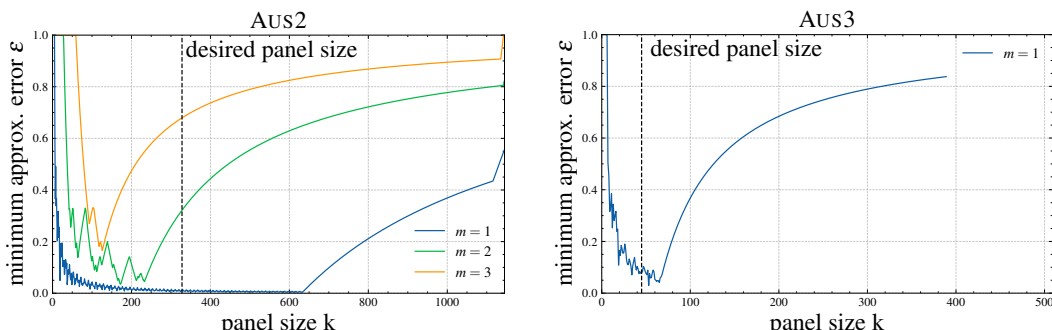

Figure 4: The smallest $\varepsilon$ for which $\varepsilon$-approximate $m$-representative panels exist as a function of the panel size $k$ for selected $m \in [|\mathcal{F}|]$. The dashed lines show the desired panel sizes.

In AUS3, it is not clear how to convert the feature 'Ability to attend all panel sessions' to a quota to use for $\varepsilon$-approximate $m$-representation. As described in Appendix A.6.3, we thus only consider panels for which at most $1/45$ of the individuals on the panel have each of the values 'No', 'Unsure', or 'Unsure-1', as was the requirement for the actually selected panel. These constraints make panels of size greater than 389 impossible, which is why the plot for AUS3 in Figure 4 does not extend beyond this value of $k$.

