# OpenReview forum: "Does Representation Guarantee Welfare?"
_NeurIPS.cc/2025/Conference — NeurIPS 2025 poster_

### Official Review · Reviewer_ibjw · 2025-06-10

**Clarity:** 3
**Significance:** 3
**Originality:** 3
**Rating:** 4
**Confidence:** 2

**Summary:**

This paper studies the role of descriptive representation in collective decision making. They show that requiring representation for pairs of demographic features (2-representation) significantly improves worse-case welfare guarantees beyond today's single-feature quotas. They demonstrate through empirical analysis that 2-representation is generally feasible in practice.

**Questions:**

As discussed in weaknesses, some questions and suggestions include:

1. How are the welfare of 2-representative panels compared with the 1-representative panels in empirical analysis? Would be good if the authors could show that.
2. Can the modeling assumptions be weaken?
3. Would be good to add discussions on more related works so that we can have a better sense of how this work stands among relevant works in this area.

**Ethical Concerns:**

["NO or VERY MINOR ethics concerns only"]

**Final Justification:**

My questions have been nicely addressed. This is a solid work with both interesting theoretical results and empirical validations. However, I'm still left with a bit of concern about whether NeurIPS would be the perfect venue for this work (which is why I gave a 4 instead of 5). Thus, I decided to maintain my rating.

**Limitations:**

yes

**Paper Formatting Concerns:**

No formatting issues.

**Quality:**

3

**Strengths And Weaknesses:**

Strengths:

1. Theoretical results are interesting and concrete. They prove that 1-representation is perfect for utilitarian social welfare but can be arbitrarily bad for $p < 1$. They then derive explicit $\epsilon$-accuracy bounds for both 1- and 2-representation via a moment problem reduction.
2. Practical feasibility of finding 2-representative panels is shown through empirical analysis.

Weaknesses:

1. In addition to the feasibility of finding 2-representative panels, I'm also curious about the welfare of 2-representative panels compared with 1-representative panels on real-world datasets. Would be good if the authors could show this.
2. Modeling assumptions are strong. For example, utilities are additive across features. I understand that this is necessary for theoretical analysis, but just not sure how much practitioners can still get out of this paper with these assumptions.
3. The related works section should be more about how this work stands among the works on computational social choice, instead of trying to persuade the reviewers that NeurIPS is a suitable venue foe this paper.

---

> ### Author Rebuttal · Authors · 2025-07-30
>
> Thank you for the review. We would be happy to add more discussion of related work, as you suggested.
>
> > How are the welfare of 2-representative panels compared with the 1-representative panels in empirical analysis? Would be good if the authors could show that.
>
> Real data from citizens' assemblies does not contain information on alternative features or utilities. While working on the paper, we considered augmenting the real data with synthetic utility data, in order to measure the empirical difference in welfare estimation between 1- and 2-representative panels. However, we eventually decided against it since we felt the results would depend too strongly on the distribution over utilities to be meaningful.
>
> > Modeling assumptions are strong. For example, utilities are additive across features. ... Can the modeling assumptions be weakened?
>
> Arguably, the only potentially restrictive assumption is the one on utilities. But we believe it is reasonable for the following reasons.
>
> First, the assumption that the utility of an agent for an alternative is given by the product of their feature vectors is common in algorithmic game theory and computational social choice. See, for example, the following papers:
>
> * Noothigattu et al. A Voting-Based System for Ethical Decision Making. AAAI 2018. (Section 5, Step II.)
> * Balseiro et al. Contextual Standard Auctions with Budgets: Revenue Equivalence and Efficiency Guarantees. Management Science 2023. (Section 2, first paragraph.)
> * Freedman et al. Adapting a Kidney Exchange Algorithm to Align with Human Values. Artificial Intelligence 2020. (Section 3.3.)
>
> Second, we note that our model is in fact more expressive than it appears at first glance. Indeed, individuals could be represented through an arbitrarily rich *feature embedding* that captures nuanced attributes. Likewise, alternatives could also be represented in the same embedding space. The model where agent utilities are the dot product of an agent feature embedding and an alternative feature embedding is ubiquitous in AI; just to give one example, it underlies the *two-tower model* of recommendation. (We are prohibited from providing a URL but you can easily Google the term.) A caveat is that the more complex the feature space, the more demanding it is to produce a representative panel.
>
> That said, a more general model would allow for non-linear utilities, e.g., the utility of an individual for an alternative could be a general concave function of the element-wise product of the individual's feature vector and the alternative's feature vector. We see this as an interesting direction for future work.

---

> > ### Comment · Reviewer_ibjw · 2025-08-03
> >
> > Thank you for the great work and explanation, and I appreciate your efforts. I decided to maintain the rating.

---

### Official Review · Reviewer_KBwz · 2025-06-30

**Clarity:** 4
**Significance:** 3
**Originality:** 3
**Rating:** 5
**Confidence:** 3

**Summary:**

The paper studies how well the welfare of representative citizen assemblies translates to the welfare of the underlying population. In particular, the paper focuses on 1- and 2-representative panels, which mirror the population with respect to the proportion of individuals possessing one or two protected attributes, such as, for example, being male and a parent.

The welfare notion considered is p-welfare, the power mean welfare: when p = 1 it captures utilitarian welfare; when p = 0 it captures Nash social welfare; and for negative p it becomes increasingly egalitarian. The question studied is how good an additive approximation the panel’s p-welfare–optimal outcome is to the p-welfare of the underlying population.

For p ≤ 1 and 1-representative assemblies, the authors prove tight bounds on the additive approximation gap ε. For 2-representative committees, they show upper bounds as well as a lower bound in the case when some alternative provides zero utility to some type. On real-world data, the authors demonstrate the feasibility of selecting 2-representative citizen assemblies.

**Questions:**

Could you elaborate on epsilon‐accuracy - for example, why you focus on additive approximations rather than multiplicative ones?

It would also be helpful if you could discuss lower bounds on the worst‐case epsilon‐accuracy when u_min > 0. Additionally, could you expand on your utility assumption, namely that individuals’ utilities for alternatives derive from the attributes those alternatives possess?

Finally, in line 123 you state “This is essentially equivalent to the notion of representation considered in prior work on panel selection.” Could you clarify to what extent these two notions are equivalent, and where they might differ?

**Ethical Concerns:**

["NO or VERY MINOR ethics concerns only"]

**Final Justification:**

Well-written, well-rounded paper.

**Limitations:**

It’s questionable whether people’s preferences for various options can truly be captured by demographic attributes like gender or age. While it’s reasonable to assume that individuals with similar characteristics with form clusters of some sort, this approach inevitably misrepresents a substantial portion of the group - particularly when we rely on the limited set of attributes commonly used in practice.

**Quality:**

4

**Strengths And Weaknesses:**

This paper is very well written and well motivated. It tackles a practically driven theoretical problem, and as a result must contend with quite involved mathematics that don’t yield neat closed-form solutions. In that light, Figure 1 is especially valuable for visualizing and understanding the resulting bounds.

Moreover, the theoretical findings are backed up by experiments demonstrating that 2-representative citizen panels often exist in practice. Taken together, this evidence suggests we should consider moving beyond single-member representation in real-world applications.

That said, the paper could be clearer in its exposition of ε-accuracy and in guiding the reader through how to interpret the additive approximations (see my questions on this point). It would also benefit from a more in-depth discussion of the challenges involved in proving a lower bound for 2-representative panels when
u_min >0.

____
Minor comments:

Page 4 Line 177: doesn't -> does not

Page 6 Line 236: or alternative -> for an alternative

Page 7 Line 276: doesn't -> does not

---

> ### Author Rebuttal · Authors · 2025-07-30
>
> Thank you for the review. We will correct the typos you pointed out in the final version of the paper.
>
> > Could you elaborate on epsilon‐accuracy - for example, why you focus on additive approximations rather than multiplicative ones?
>
> We focus on additive rather than multiplicative approximation because this mirrors standard notions of accuracy in machine learning and statistics, so it is arguably the more natural choice. For example, in the PAC learning framework, accuracy is measured using an additive approximation: for an accuracy parameter $\epsilon$ and a confidence parameter $\delta$, the probability that that the *difference* between the true error and the empirical error of a hypothesis is at most $\epsilon$ is at least $1-\delta$.
>
> That said, one can derive tight bounds for multiplicative $\epsilon$-accuracy from our proof technique by taking the ratio of the utility-maximum and utility-minimum instead of the difference in Lemma 15 and finishing the final proof steps accordingly. A matching lower bound can be proven with the techniques from the proof of the lower bound in Theorem 8.
>
> > It would also be helpful if you could discuss lower bounds on the worst‐case epsilon‐accuracy when $u_{min} > 0$.
>
> For the rebuttal, we explored this helpful question and found that *the upper bound in Theorem 9 on the accuracy of 2-representative panels is tight!* One can prove this very similarly to the lower bound on the accuracy of 1-representative panels in Theorem 8. In fact, we believe that this proof technique can also be used for $k>2$, showing that the accuracy bounds for $k$-representation one can obtain with our upper bounding technique are tight. In the revised paper, we will include this matching lower bound and provide a proof in the appendix.
>
>
> > Additionally, could you expand on your utility assumption, namely that individuals’ utilities for alternatives derive from the attributes those alternatives possess?
>
> This is a fair question; our answer is twofold.
>
> First, the assumption that the utility of an agent for an alternative is given by the product of their feature vectors is common in algorithmic game theory and computational social choice. See, for example, the following papers:
>
> * Noothigattu et al. A Voting-Based System for Ethical Decision Making. AAAI 2018. (Section 5, Step II.)
> * Balseiro et al. Contextual Standard Auctions with Budgets: Revenue Equivalence and Efficiency Guarantees. Management Science 2023. (Section 2, first paragraph.)
> * Freedman et al. Adapting a Kidney Exchange Algorithm to Align with Human Values. Artificial Intelligence 2020. (Section 3.3.)
>
> Second, we note that our model is in fact more expressive than it appears at first glance. Indeed, individuals could be represented through an arbitrarily rich *feature embedding* that captures nuanced attributes. Likewise, alternatives could also be represented in the same embedding space. The model where agent utilities are the dot product of an agent feature embedding and an alternative feature embedding is ubiquitous in AI; just to give one example, it underlies the *two-tower model* of recommendation. (We are prohibited from providing a URL but you can easily Google the term.) A caveat is that the more complex the feature space, the more demanding it is to produce a representative panel.
>
>
> > Finally, in line 123 you state “This is essentially equivalent to the notion of representation considered in prior work on panel selection.” Could you clarify to what extent these two notions are equivalent, and where they might differ?
>
> The definitions of 1-representation in prior work vary in how much 'tolerance' they allow, similar to our relaxation of $k$-representativeness in the empirical part. Specifically, previous papers assume that there is a lower and upper quota on the number of individuals with a certain feature value in the panel, with the corresponding proportion of people in the population with this feature value being within these quotas. See, for example:
>
> Flanigan et al. Fair Sortition Made Transparent. NeurIPS 2022. (Section 2, second paragraph.)

---

> > ### Comment · Reviewer_KBwz · 2025-08-06
> >
> > Thank you for addressing my questions! Well done on additional results - please include the tight upper bound for Theorem 9 as well as the result for k>2 in the revised paper.
> >
> > Regarding the utility assumption:
> > My criticism was not that there are no features where a dot product of feature vectors would be defendable (this is a wormhole i did not want to open). I meant more that gender, age, etc., seem like a very suboptimal embedding space, and these attributes are used as ongoing example in the paper. In the two-tower model, the key is that the embedding space is learned from data and features can be very rich, including complicated superpositions), while here we have a human-imposed embedding space. It would be very interesting to see the “bitter lesson” implemented in citizen assemblies, e.g., by learning feature embeddings instead of imposing them (perhaps like with medical symptom checkers that are learned with help of AI and human supervision).

---

### Official Review · Reviewer_4ra7 · 2025-07-01

**Clarity:** 3
**Significance:** 3
**Originality:** 2
**Rating:** 4
**Confidence:** 3

**Summary:**

For citizen assemblies a panel of individuals is selected from the population in a way that proportionally reflects the population’s composition across multiple features, such as age, gender, and ethnicity. However, even when an individuals preferences are only determined by their features, an optimal decision for a (1-)representative panel must not be optimal for the population. In this paper, the authors explore how close the social welfare of a representative panel aligns with the social welfare of the entire population. They show that the welfare aligns significantly better when every combination of two features is proportionally represented in the panel, where welfare is measured using the p norm of individual utilities with $p < 1$. Depending on the size and diversity of the volunteer pool, it may not be possible to select such a panel. To address this, the authors include an experimental section in which they assess the feasibility using data from four past citizen assemblies.

**Questions:**

* In line 123, you state that that your definition of 1-representation is "essentially equivalent" to notions defined in the literature. Could you clarify what the differences are?

* In the theoretical part of the paper, you assume panels to be perfectly m-proportional. In the experimental section, this requirement is relaxed to proportionality rounded up or down. How would this relaxation affect the theoretical guarantees?

* What was the rationale behind the selection of the four specific citizen assemblies used in the experimental evaluation?

**Ethical Concerns:**

["NO or VERY MINOR ethics concerns only"]

**Final Justification:**

The authors have addressed my questions. However, the weaknesses and limitations identified in my original review remain. Therefore, I have decided to keep my score unchanged.

**Limitations:**

yes

**Paper Formatting Concerns:**

/

**Quality:**

3

**Strengths And Weaknesses:**

The paper is well-written and addresses a relevant question in the context of selecting citizen assemblies. The long introduction effectively motivates the research and clearly presents the main questions and results. However, some parts are difficult to follow without prior knowledge of the model, which is introduced in the subsequent section.

The related work section feels rushed and lacks a structured overview of relevant literature. A proper discussion of existing approaches and how this work relates to them is necessary in my opinion.

The main theoretical results seem significant, their proofs are deferred to the appendix. The paper does provide an outline of the proof for one of the key results (Theorem 9). However, an important concept—the 'moment problem'—is never properly introduced or explained, despite its apparent centrality to the argument.

In the experimental section, the authors check whether a 2-proportional panel of desired size can be found for four real-world instances. While this provides some insight, the small number of instances limits the ability to draw broader conclusions about feasibility. This limitation may be due to the inavailability of data, though this is not explicitly addressed in the paper.

---

> ### Author Rebuttal · Authors · 2025-07-30
>
> Thank you for the review. We will implement your suggestions, including expanding the related work introducing the moment problem in more detail.
>
> > In line 123, you state that that your definition of 1-representation is "essentially equivalent" to notions defined in the literature. Could you clarify what the differences are?
>
> The definitions of 1-representation in prior work vary in how much 'tolerance' they allow, similar to our relaxation of $k$-representativeness in the empirical part. Specifically, previous papers assume that there is a lower and upper quota on the number of individuals with a certain feature value in the panel, with the corresponding proportion of people in the population with this feature value being within these quotas. See, for example:
>
> Flanigan et al. Fair Sortition Made Transparent. NeurIPS 2022. (Section 2, second paragraph.)
>
> > In the theoretical part of the paper, you assume panels to be perfectly m-proportional. In the experimental section, this requirement is relaxed to proportionality rounded up or down. How would this relaxation affect the theoretical guarantees?
>
> We believe that the qualitative results don't change when considering small errors with respect to the $m$-representativeness quotas, but a a precise theoretical analysis of this question seemed intractable when we considered it. A larger panel size comes with smaller error with respect to exact $m$-representation, which presents a way to alleviate this concern in practice.
>
> > What was the rationale behind the selection of the four specific citizen assemblies used in the experimental evaluation?
>
> We chose these four citizens' assemblies in the paper because the data on the underlying populations — in particular, intersections between features — are publicly available and didn't require additional permission to be used. In the geographical areas where this underlying population data was available, we asked practitioners for the datasets on the individuals in the pool of candidates. We present all citizens' assemblies for which we obtained both candidate and population data in the paper.

---

> > ### Comment · Reviewer_4ra7 · 2025-08-02
> >
> > I thank the authors for their answer. At this point, I have no further questions.

---

### Official Review · Reviewer_jovN · 2025-07-01

**Clarity:** 4
**Significance:** 3
**Originality:** 3
**Rating:** 5
**Confidence:** 3

**Summary:**

The paper studies citizens’ assemblies. Its goal is to determine whether descriptively representative panels make welfare-optimal decisions for the broader population.

The authors use the notion of m-representatives and show that 1-representatives are accurate for the arithmetic mean, but not necessarily for other means. They show that for 2-representatives, the worst-case accuracy is better than in the 1-representative case.

Finally, the paper presents an empirical analysis, showing the relationship between the representation level and panel size in practice.

**Questions:**

Why is there an assumption of the uniformity among voters? (That is, all individuals with the same values for the features used in the model (e.g., age and gender) are assumed to have the same utility for any given alternative.)

**Ethical Concerns:**

["NO or VERY MINOR ethics concerns only"]

**Final Justification:**

My evaluation remains unchanged.
I'm still in favor of accepting the paper.

**Limitations:**

Yes.

**Paper Formatting Concerns:**

None.

**Quality:**

4

**Strengths And Weaknesses:**

My overall evaluation of this paper is very positive.

+Assemblies are one of the most interesting recent topics within computational social choice.

+The paper is extremely clearly written and well-positioned within the literature.

+The concept of *m*-representation is intuitive and elegant, and the paper provides bounds on the accuracy of such representation.

+The paper includes an empirical analysis, showing, among other things, that it is indeed possible to select 2-representative panels in real-world instances.

Side notes:

-Minor issue regarding the plots: the text size is too small to read comfortably.

-I'm not sure the title accurately reflects the paper's content. It is catchy but not very descriptive.

---

> ### Author Rebuttal · Authors · 2025-07-30
>
> Thank you for the review. We will increase the text size in the plots in the final version.
>
> > Why is there an assumption of the uniformity among voters? (That is, all individuals with the same values for the features used in the model (e.g., age and gender) are assumed to have the same utility for any given alternative.)
>
> This is a fair question; our answer is twofold.
>
> First, the assumption that the utility of an agent for an alternative is given by the product of their feature vectors is common in algorithmic game theory and computational social choice. See, for example, the following papers:
>
> * Noothigattu et al. A Voting-Based System for Ethical Decision Making. AAAI 2018. (Section 5, Step II.)
> * Balseiro et al. Contextual Standard Auctions with Budgets: Revenue Equivalence and Efficiency Guarantees. Management Science 2023. (Section 2, first paragraph.)
> * Freedman et al. Adapting a Kidney Exchange Algorithm to Align with Human Values. Artificial Intelligence 2020. (Section 3.3.)
>
> Second, we note that our model is in fact more expressive than it appears at first glance. Indeed, individuals could be represented through an arbitrarily rich *feature embedding* that captures nuanced attributes. Likewise, alternatives could also be represented in the same embedding space. The model where agent utilities are the dot product of an agent feature embedding and an alternative feature embedding is ubiquitous in AI; just to give one example, it underlies the *two-tower model* of recommendation. (We are prohibited from providing a URL but you can easily Google the term.) A caveat is that the more complex the feature space, the more demanding it is to produce a representative panel.

---

> > ### Comment · Reviewer_jovN · 2025-08-04
> >
> > I thank the authors for addressing my comments.

---

### Official Review · Reviewer_o2qy · 2025-07-13

**Clarity:** 4
**Significance:** 3
**Originality:** 3
**Rating:** 4
**Confidence:** 4

**Summary:**

The paper explores descriptive representation in the context citizens' assembly, focusing on the worst-case guarantees of 2-representative panels (i.e., representative of the underlying population over intersections of features). The paper quantitatively shows that 2-representative panels can more accurately estimate $p$-mean social welfare, and presents theoretical and experimental results.

**Questions:**

The authors are encouraged but not required to answer the questions mentioned in “Strengths & Weaknesses” above.

**Ethical Concerns:**

["NO or VERY MINOR ethics concerns only"]

**Final Justification:**

Most (if not all) of my comments are constructive suggestions. The authors responded that they would revise the paper accordingly. A few short and clear points were provided under "Strengths and Weaknesses."

**Limitations:**

yes

**Quality:**

3

**Strengths And Weaknesses:**

Strengths
+ The paper provides a strong motivation for studying 1- and 2-representative panels with clear practical implications. The paper could be strengthened by discussing limitations of current methods (a real-world situation like Example 7?) and plans for trying out 2-representative assemblies in practice.
+ The authors give tight bounds for the worst-case guarantee of 1-representative panels (as a function of $p$ and the minimum utility $u_{\mathrm{min}}$).
+ Overall the paper is well written and the presentation is clear.

Weaknesses
- The paper could be strengthened by highlighting its technical contributions, e.g., which results are the most challenging to prove, and how the authors overcome the challenges. Is the upper bound in Theorem 9 tight, and is there any concrete evidence that the optimization program in Theorem 9 does not admit a closed-form solution?
- The authors could consider exploring a weighted version of descriptive representation (where each person on the panel is assigned a nonnegative weight), which may help address the issue that $q_i k$ is not an integer. The authors may want to look into characteristic functions, Fourier transforms, and their connections to low-degree moments. The reviewer wonders if the notions in the paper have any connections to proportional fairness (i.e., no subpopulation wants to leave and form a proportional smaller panel).
- The experimental setup is clearly described, but the paper could better justify the design choices. It feels a bit unnatural to use the maximum possible panel size as a measure of how easy it is to satisfy the representative constraints (i.e. it is harder to round the estimates to the true value when the true value is large). Would it make sense to study a multiplicative notion (e.g., the desired and actual number of people are within a factor of $1 \pm 0.1$)? And following this line of thought, does randomly sampling $O(k \log d / \epsilon^2)$ times from the population gives a $1 \pm \epsilon$ multiplicative tolerance for $k$-representativeness (which seems to follow from standard application of Chernoff bound and the union bound).

Typos
- Line 87: "foundational results ... has".
- Page 7: the printed version of Figure 1 is hard to read.
- Line 309: missing a period.

---

> ### Author Rebuttal · Authors · 2025-07-30
>
> Thank you for the review. We will highlight the technical contributions and correct the typos you pointed out in the final version of the paper. Below we briefly respond to some of your comments.
>
> > Is the upper bound in Theorem 9 tight?
>
> For the rebuttal, we explored this helpful question and found that *the upper bound in Theorem 9 on the accuracy of 2-representative panels is tight!* One can prove this very similarly to the lower bound on the accuracy of 1-representative panels in Theorem 8. In fact, we believe that this proof technique can also be used for $k>2$, showing that the accuracy bounds for $k$-representation one can obtain with our upper bounding technique are tight. In the revised paper, we will include this matching lower bound and provide a proof in the appendix.
>
> > ... and is there any concrete evidence that the optimization program in Theorem 9 does not admit a closed-form solution?
>
> We strongly believe that the optimization program in Theorem 9 doesn't admit a closed-form solution. The best evidence for this is the case $p=0$ (this optimization program can be found in Appendix A.3), where the variables appear both in the exponent and in the base of the first-order conditions. For a general $p$, the variables appear in the first-order conditions with high and non-integer powers in both the denominator and numerator, which is generally a good indicator that no closed-form solution exists.
>
> > The authors could consider exploring a weighted version of descriptive representation (where each person on the panel is assigned a nonnegative weight), which may help address the issue that is not an integer. The authors may want to look into characteristic functions, Fourier transforms, and their connections to low-degree moments. The reviewer wonders if the notions in the paper have any connections to proportional fairness (i.e., no subpopulation wants to leave and form a proportional smaller panel).
>
> Thank you for the interesting suggestions. While idea of a weighted panel is theoretically interesting, we believe it would not be seen as legitimate in practice. Regarding notions of proportionality (such as proportionally fair clustering, justified representation, etc.), we have worked extensively on such notions and we do not believe they can be fruitfully connected to the paper under consideration.
>
> > The experimental setup is clearly described, but the paper could better justify the design choices. It feels a bit unnatural to use the maximum possible panel size as a measure of how easy it is to satisfy the representative constraints (i.e. it is harder to round the estimates to the true value when the true value is large). Would it make sense to study a multiplicative notion (e.g., the desired and actual number of people are within a factor of $1 \pm 0.1$)?
>
> Thank you for the interesting suggestion. The definition of $k$-representativeness we used in the empirical part of this work comes from wanting to be as close to being perfectly $k$-representative as possible for a given panel size. In this setting, practitioners would generally want to choose a larger panel size to get more accurate $k$-representativeness, which may require a larger candidate pool size.
>
> We re-ran our experiments to find (for different panel sizes and different $k$) the smallest possible $\varepsilon$ such that there exists a panel that is $(i)$ $k$-representative up to a multiplicative factor of $1\pm\varepsilon$, and $(ii)$ has the desired size, as you suggested. When the number of feature-value tuples considered is greater than the panel size, then $\varepsilon=1$, since for at least one feature-value tuple, no person will be in the panel, but the required quota is positive. If the panel size is greater than this lower bound, the achievable $\varepsilon$ decreases with the panel size (approximately) inversely linearly, as expected, up to the point where the pool is 'too skewed', roughly when for some feature-value tuple the quota is higher than the number of candidates in the pool with this feature-value tuple. This point is in all cases a bit before the maximum possible panel size of $k$-representative panels according to the notion used in the paper. After this 'maximum representative panel size,' $\varepsilon$ increases with the panel size, again as expected. To illustrate this, we list the results for the Aus2 dataset from the paper below:
>
> | $k$ | too small panel sizes, $\varepsilon=1$ | range of good panel sizes, $\varepsilon \sim \Theta(1/n)$ | critical panel size $c$ | panel sizes for which pool is 'too skewed', $\varepsilon\sim\Theta(1-\frac{c}{n})$ | largest possible $k$-rep. panel size in additive definition |
> | - | ----------------- | ----------- | --- | ------------- | --------------------- |
> | **1** | 1-5               | 6-632    | 632   | 633+          | 640                   |
> | **2** | 1-27              | 28-233   | 233   | 234+          | 278                   |
> | **3** | 1-59              | 60-126   | 126   | 127+          | 210                   |
>
> We believe that this additional experiment is a valuable addition to our understanding of $k$-representative panels in practice. First, it confirms that increasing the panel size comes with more accurate $k$-representation up to some 'maximum panel size': up to the critical size $c$, the panel size is inversely proportional to $\varepsilon$. More importantly, it suggests that the maximum panel size we obtain from our additive approximation may be too high: there is a small gap between the critical size defined above, after which the pool is effectively too skewed, and the maximum possible size (in the additive sense). In this gap, the 'quality' of the $k$-representation already decreases with panel size, so the critical size can be interpreted as the largest reasonable panel size up to which adding people to the panel comes with increased representation. Fortunately, our new experiment shows that in the real instances we analyzed, the critical size for $2$- (or even $3$-) representative panels is frequently above the desired panel size and never drastically below.
>
> In practice, a mix of the two approximations may do best: A downside of the multiplicative approximation is that if quotas for some feature-value tuples are small (say, 2.5 people of a certain type on the panel), even the 'closest' integer will induce a large epsilon (0.2 in this example), which will then in turn give feature-value tuples with larger quotas more wiggle room than necessary. An obvious downside of the additive approximation is that it doesn't 'incentivize', say, having 3 people of a certain feature-value tuple on the panel instead of 2 if the quota is 2.9. Furthermore, there is a tradeoff between tighter approximation guarantees and more equal selection probabilities for individuals in the pool. Which notion should be employed will depend on the goals of the practitioners.
>
> We will add a discussion of these new insights and the additional experimental results in the revised paper.
>
>
> > And following this line of thought, does randomly sampling $O(k \log d/\epsilon^2)$ times from the population gives a $1 \pm \epsilon$ multiplicative tolerance for $k$-representativeness (which seems to follow from standard application of Chernoff bound and the union bound).
>
> We considered selecting panels by randomly sampling from the population in the setting of welfare estimation but saw that the number of samples (=people on the panel) needed to get accuracy guarantees similar to those for (worst-case) 1- or 2-representative panels exceeds the size of real citizens' assemblies.

---

> > ### Comment · Reviewer_o2qy · 2025-08-05
> >
> > I thank the authors for their detailed rebuttal and for running additional experiments in response to my comments.
> >
> > The plot for the new experiments (e.g., $\epsilon$ decreases as the panel size increases) and the discussion (e.g., the notion of "critical size") will strengthen the paper. I agree with the authors' observation that a mix of additive/multiplicative notions could be more useful in practice.

---

### Decision · Program_Chairs · 2025-09-17

**Decision:**

Accept (poster)

**Comment:**

The paper studies citizens' assemblies and shows that intersectionality improves social welfare compared to the traditional approach of proportional representation by individual features.

The reviewers found the paper to be nicely written and the topic of the paper to be interesting. The reviewers found the concept of m-representation to be intuitive and appreciated the results of the paper. I think this paper would be a nice addition to NeurIPS.

The reviewers had several suggestions that I hope the authors can incorporate for the camera-ready version, given the additional page. (1) Reviewer o2qy had an interesting suggestion about the multiplicative notion. The authors have provided additional interesting experimental results in the rebuttal. I think this is a nice result that should be added to the paper. (2) The text size in some of the plots is too small. Please consider editing these plots to make the text bigger. (3) Please expand the discussion of the related work. This paper is relevant to NeurIPS as many prior works are published here, so please expand the related work without focusing on showing how this paper is relevant to the community. (4) Please include your answer to Reviewers KBwz and jovN about uniformity in the final version.